



# Understanding cryogenic frost point hygrometer measurements after contamination by mixed-phase clouds

Teresa Jorge[1], Simone Brunamonti[1], Yann Poltera[1], Frank G. Wienhold[1], Bei P. Luo[1], Peter Oelsner[2], Sreeharsha Hanumanthu[3], Bhupendra B. Singh[4,5], Susanne Körner[2], Ruud Dirksen[2], Manish Naja[6], Suvarna Fadnavis[4], and Thomas Peter[1]

[1]Institute of Atmospheric and Climate Science, ETH Zürich
[2]Deutscher Wetterdienst (DWD)/ GCOS Reference Upper Air Network (GRUAN) Lead Center, Lindenberg, Germany
[3]Forschungzentrum Jülich (FZJ), Institute of Energy and Climate Research, Stratosphere (IEK-7), Jülich, Germany
[4]Indian Institute of Tropical Meteorology (IITM), Pune, India
[5]Department of Geophysics, Banaras Hindu University, Varanasi, India
[6]Aryabhatta Research Institute of Observational Sciences (ARIES), Nainital, India

**Correspondence:** Teresa Jorge (teresa.jorge@env.ethz.ch)

**Abstract.** Balloon-borne water vapour measurements in the (sub)tropical upper troposphere and lower stratosphere (UTLS) by means of frost point hygrometers provide important information on air chemistry and climate. However, the risk of contamination from sublimating hydrometeors collected by the intake tube may render these measurements difficult, particularly after crossing low clouds containing supercooled droplets. A large set of measurements during the 2016-2017 StratoClim balloon

5  campaigns at the southern slopes of the Himalayas allows us to perform an in-depth analysis of this type of contamination. We investigate the efficiency of wall-contact and freezing of supercooled droplets in the intake tube and the subsequent sublimation in the UTLS using Computational Fluid Dynamics (CFD). We find that the airflow can enter the intake tube with impingement angles up to 60°, owing to the pendulum motion of the payload. Supercooled droplets with radii > 70 µm, as they frequently occur in mid-tropospheric clouds, typically undergo contact freezing when entering the intake tube, whereas only about 50%

10  of droplets with 10 µm radius freeze, and droplets < 5 µm radius mostly avoid contact. According to CFD, sublimation of water from an icy intake can account for the occasionally observed high water vapour mixing ratios ($\chi_{H_2O} > 100$ ppmv) in the stratosphere. Furthermore, we use CFD to differentiate between stratospheric water vapour contamination by an icy intake tube and contamination caused by outgassing from the balloon and payload, revealing that the latter starts playing a role only at high altitudes ($p < 20$ hPa).

15  *Copyright statement.* The authors declare this is an original work





## 1 Introduction

Sources of contamination for cryogenic frost point hygrometers are water vapour outgassing from the balloon envelope, the parachute, the nylon cord, or sublimation of hydrometeors collected in the intake tube of the instrument (Hall et al., 2016; Vömel et al., 2016). These are contamination sources common to all balloon-borne water vapour measurement techniques

(Goodman and Chleck, 1971; Vömel et al., 2007c; Khaykin et al., 2013). Contamination can be severe in the stratosphere where the environmental water vapour mixing ratios are 2 - 3 orders of magnitude smaller than in the troposphere. Over time this type of contamination has been reduced by increasing the length of the cord by means of an unwinder and by giving preference to descent over ascent data (Mastenbrook and Dinger, 1961; Mastenbrook, 1965, 1968; Mastenbrook and Oltmans, 1983; Oltmans and Hofmann, 1995; Vömel et al., 1995; Oltmans et al., 2000; Khaykin et al., 2013; Hall et al., 2016). Standard

lengths presently used are of the order of 50 to 60 m (Vömel et al., 2016; Brunamonti et al., 2018). The World Meteorological Organization recommends ropes longer than 40 m (WMO, 2008; Immler et al., 2010). Nevertheless, it has been shown (Kräuchi et al., 2016) that the balloon wake in combination with the swinging motion of the payload leaves a quasi-periodic signal even in the temperature measurements. Descent data is not always an option because some instrument intakes and control systems are optimized for ascent (Kämpfer, 2013).

The first water vapour measurements in the stratosphere by balloon borne instruments reported a frost point temperature of about -70 °C at 15 hPa (Barret et al., 1949, 1950; Suomi and Barrett, 1952), corresponding to unrealistically high $H_2O$ mixing ratios (> 100 ppmv). Later, Mastenbrook and Dinger (1961) used a new light-weight dew point instrument for frost point measurements in the stratosphere. For the first time, measures to minimize contamination of the air sample with moisture carried aloft by the balloon were mentioned. The instrument was carried about 275 m below the balloon assembly and the

ascent data was accepted only if validated by the descent data. The descent was achieved by two methods: the use of a big parachute or the use of a tandem balloon assembly. Nowadays, controlled descent profiles are obtained by using a valve in the balloon neck (Hall et al., 2016; Kräuchi et al., 2016) that slowly releases gas from the balloon. Nearly all frost point hygrometers (FPH) operated by NOAA use it.

Mastenbrook (1965, 1968) identified contamination of the instrument package as a source of the higher and more variable

concentrations of water vapour at stratospheric levels. The surfaces of the sensing cavities and intake ducts were considered as a potential contamination source and redesigned using stainless steel and allowing higher flow rates. These improvements enabled to measure typical stratospheric $H_2O$ mixing ratios of about 4 ppmv. Mastenbrook (1966) started building fully symmetric instruments for ascent and descent. Mastenbrook and Oltmans (1983) paid particular attention to the intake tubes of the frost point hygrometer. These tubes, until today, are 2.5 cm in diameter, made of 25 µm thick stainless steel and need to

be thoroughly cleaned before flight. They extend above and below the instrument package by more than 15 cm, shielding the air against the contamination by water outgassing from the instrument's Styrofoam containment. The frost point temperature is measured at a surface extruding 1.25 cm from the intake tube wall and placed at the centre of the tube, 17 cm from the opening of the intake tube.





New designs of frost point hygrometers such as the SnowWhite sonde from Meteolabor, steered away from the intake tube design (Fujiwara et al., 2003; Vömel et al., 2003). However, the SnowWhite design was shown to be susceptible to the ingress of hydrometeors in the intake (Cirisan et al., 2014). Under supersaturated conditions, the intake duct was actively heated, with the intention to measure the total water content (TWC), i.e. gaseous plus particulate $H_2O$, instead of just gaseous $H_2O$ mixing

ratio, as claimed by the manufacturer (Vaughan et al., 2005). The SnowWhite sonde was also reported to measure saturation over ice in the troposphere of 120-140%, which could not be modelled irrespective of the assumed scenario, leading Cirisan et al. (2014) to conclude the measurement was erroneous and likely created by contamination.

With the increasing miniaturization and ease of use, balloon-borne frost point hygrometers started to be employed more systematically at an increasing number of locations and under a wide range of meteorological conditions (Vömel et al., 2002,

2007b; Bian et al., 2012; Hall et al., 2016; Brunamonti et al., 2018), creating new challenges for the instrument. When passing through mixed-phase clouds with supercooled liquid droplets, the balloon and payload surfaces can accumulate ice, which will sublimate in the subsaturated environment of the stratosphere. Intake tubes might represent a preferential surface for this type of contamination (Vömel et al., 2016).

During the 2016-2017 StratoClim balloon campaigns on the southern slopes of the Himalayas, 43 out of a total of 63 sound-

ings carried water vapour measurements by means of the Cryogenic Frost point Hygrometer (CFH; see Vömel et al. (2007b, 2016)) and of these 9 showed strongly contaminated water vapour measurements in the stratosphere. These 9 soundings are shown in Figure 1 (see also grey points in Fig. 2 of Brunamonti et al. (2019)). This is a common feature when the troposphere is very moist, such as during tropical deep convection [Holger Vömel, personal communication, 2016]. Contamination requires a careful quality check of the CFH data, representing a source of uncertainty, especially in the lower stratosphere. This arti-

fact can also lead to systematic biases, as it makes the operator prefer dryer launching conditions, which can affect satellite validation procedures and climatological records (Vömel et al., 2007a).

Here, we perform a thorough analysis of this type of contamination. In Section 2, we present the data of the 9 contaminated cases found during StratoClim, select cases with suitable cloud information from the COmpact Backscatter AerosoL Detector (COBALD), and analyse mixed-phase cloud conditions. In Section 3, we describe balloon trajectories and estimate

the impingement angles of supercooled droplets onto the top of the intake tubes. Section 4 introduces the Computational Fluid Dynamic (CFD) tool FLUENT by ANSYS (2012). In Section 5, we present the results of the different CFD studies, namely: Section 5.1 for the freezing efficiency of supercooled droplets; Section 5.2 for the CFD-based description of the sublimation process and the evolution of the ice layer; Section 5.3 for the implications for the measurements in the upper troposphere; and Section 5.4 for the simulation of the contamination stemming from the balloon envelope and instrument packaging. Finally,

Section 6 provides design and operation recommendations to decrease the effect of contamination.

## 2   StratoClim Balloon Campaigns

Brunamonti et al. (2018) offers an overview of the instrumentation and dataset collected during the 2016 - 2017 StratoClim balloon campaigns at the southern slopes of the Himalayas, deriving a comprehensive understanding of the morphology and





large-scale dynamics of the Asian Summer Monsoon Anticyclone (ASMA). Here, we focus on humidity measurements in the upper troposphere and lower stratosphere region, including the measurement of mixed-phase and ice clouds, and provide brief instrument descriptions.

## 2.1 CFH and RS41 Water Vapour Measurements

The two instruments measuring water vapour content in this study were the radiosonde RS41-SGP (herein after referred to as 'RS41') manufactured by Vaisala, Finland (Vaisala, 2013), and the Cryogenic Frost point Hygrometer, CFH (Vömel et al., 2007c, 2016) manufactured by ENSCI (USA). The RS41 measures relative humidity (RH) by means of a thin film capacitive sensor (Jachowicz and Senturia, 1981) with a nominal uncertainty in soundings of 4% for temperature $T >$ -60 °C (Vaisala, 2013). In this study, we used corrected RH data provided by RS41, which implements an empirical time lag correc-

tion, accounting for the operation of the capacitive sensor under heated conditions by $\Delta T = 5$ K above ambient temperature and correcting for irregularities determined by the automatized ground check (Vaisala, 2013). In contrast to RS41, CFH measures the frost point temperature ($T_{\text{frost}}$). It controls the thickness of a dew or frost layer on a mirror by heating the mirror against a cold reservoir of cryogenic refrigerant liquid. When the dew or frost layer is in equilibrium with the air mass, i.e. neither growing nor evaporating, it is by definition at the dew point or frost point temperature, which is a direct measure of the $H_2O$

partial pressure in the gas phase. The uncertainty of CFH has been estimated to be smaller than 10% in water vapour mixing ratio up to approximately 28 km altitude (Vömel et al., 2007c, 2016).

The performance of the two instruments during the 2016 - 2017 StratoClim balloon campaigns has been thoroughly compared and a dry bias of 3-6% (0.1-0.5 ppmv) for 80-120 hPa, and 9% (0.4 ppmv) for 60-80 hPa of the RS41 compared to the CFH was found (Brunamonti et al., 2019). These were campaign mean results, whereas flight by flight discrepancies as large

as 50% could occur. In previous publications of this dataset (Brunamonti et al., 2018, 2019), contaminated measurements in the stratosphere were discarded using an empirical threshold. In particular, all data above the cold-point tropopause (CPT) was flagged as contaminated, if $H_2O$ mixing ratios exceeded 10 ppmv at any altitude in the stratosphere. In addition, all data at pressures below 20 hPa was also discarded, due to suspected contamination by the balloon or payload train. With decreasing pressures starting above about the 60-hPa level, all RS41 measurements showed an unrealistic increase in $H_2O$ mixing ratios

up to several tens of ppmv (Brunamonti et al., 2019). We did not consider this behaviour to be due to contamination, as the capacitive sensor of RS41 is constantly heated to 5 °C warmer than ambient air preventing icing of the sensor in supercooled clouds and supersaturation conditions. Rather, the operation of the capacitive sensor is limited at very low relative humidity. In contrast to Brunamonti et al. (2018, 2019), here we did not remove CFH clearing and freezing cycles (Vömel et al., 2016), which occurred twice per flight at frost point temperature of approximately -15 °C and -53 °C, because this gave us confidence

that the phase of the deposit in the mirror after the clearing cycle was ice.

We compared the dew- and frost-related quantities (dew and frost points, corresponding RHs, mixing ratios) of CFH and RS41 as follows. The ice saturation ratio $S_{\text{ice}}$, i.e. relative humidity with respect to ice, was calculated using the frost point temperature measured by CFH, the air temperature measured by RS41, and the parameterisation for saturation vapour pressure over ice by Murphy and Koop (2005). While relative humidity with respect to liquid water ($S_{\text{liq RS41}}$, also sometimes simply





termed 'RH') was directly measured by RS41, we also present relative humidity ($S_{liq}$) computed from CFH frost point temperature, RS41 air temperature and the parameterisation for saturation vapour pressure over water by Murphy and Koop (2005). $S_{liq,d}$ considers the deposit on the CHF mirror to be dew, i.e. liquid water, and $S_{liq,f}$ considers the deposit to be frost, i.e. ice. Water vapour mixing ratio ($\chi_{CFH}$) in ppmv from the CFH was calculated from the frost (or dew) point temperature, the air

pressure from RS41 and the parameterisation for saturation vapour pressure over ice (or liquid water) by Murphy and Koop (2005). The water vapour mixing ratio in ppmv derived from the RS41 ($\chi_{RS41}$) uses the relative humidity, air temperature, and air pressure from RS41 and the parameterisation for saturation vapour pressure over water by Hardy (1998) as used by Vaisala (2013).

All data presented was taken during balloon ascent, because this was the part of the flight affected by contamination. We
averaged all data in 1 hPa intervals (bins) from the ground to the burst altitude. The downward looking intake did not get contaminated by hydrometeors during mixed-phase cloud traverses. However, we preferred ascent over descent because the instrument's descent velocity in the stratosphere was very high, up to 50 m s$^{-1}$, which might have caused controller oscillations and would allow for very little vertical resolution. We show below that it is important to consider payload pendulum oscillations to explain certain features in the humidity measurements. For their analysis we used 1-s GPS data retrieved from RS41. We
also used GPS altitude as the main vertical coordinate for all instruments. The ascent velocity ($w$) in m s$^{-1}$ and latitude and longitude are taken directly from the RS41 GPS product.

## 2.2 COBALD Backscatter measurements

In StratoClim, we performed a total of 63 balloon soundings, namely 35 in 2016 from Nainital (NT), India, and 28 in 2017 from Dhulikhel (DK). Of these soundings, 43 carried CFH, and 20 of these were performed at night also carrying COBALD, so
that liquid and ice clouds in the lower and middle troposphere could be detected. In Figure 1, the contaminated profiles of the CFH soundings are displayed by black lines, while the mean uncontaminated profile is shown by the gray line. Three of the 9 contaminated CFH soundings also carried COBALD, namely NT007, NT011 and NT029. We analysed these three soundings in more detail.

COBALD data is expressed as backscatter ratio (BSR), i.e., the ratio of the total-to-molecular backscatter coefficients. This
is calculated by dividing the total measured signal by its molecular contribution, which is computed from the atmospheric extinction according to Bucholtz (1995), and using air density derived from the measurements of temperature and pressure. The COBALD BSR uncertainty as inferred by this technique is estimated to be around 5% (Vernier et al., 2015). For the backscatter data analysis, we present also the Colour Index (CI). CI is defined as the 940-to-455 nm ratio of the aerosol component of the BSR, i.e., $CI = (BSR_{940} - 1)/(BSR_{455} - 1)$. CI is independent of the number density; therefore, it is a
useful indicator of particle size as long as particles are sufficiently small, so that Mie scattering oscillations can be avoided, namely radii smaller than 2-3 μm. From this one obtains CI < 7 for aerosol and CI > 7 for cloud particles (Cirisan et al., 2014; Brunamonti et al., 2018).

The CFH-COBALD combination is a powerful tool to investigate cirrus clouds. Although the estimation of ice water content (IWC) from the BSR measurements of COBALD with just 2 wavelengths–455 nm and 940 nm–is quite uncertain without


additional information about the ice crystal size or distribution, IWC can be constrained for thin cirrus clouds (Brabec et al., 2012). The retrieval of mean particle size is a matter of size distribution complexity: if the distribution is simple as is the case for stratospheric aerosol, the mode radius can be estimated from the color index (Rosen and Kjome, 1991). In this work, however, we were interested in relatively thick mixed-phase clouds as observed in tropical convection (Wendisch et al., 2016;

Cecchini et al., 2017). The backscatter from dense mixed-phase clouds may saturate COBALD. In this context, it was hard to retrieve more information from COBALD than the vertical thickness of these clouds and the indication if they were purely glaciated (CI $\sim$ 20) or not.

### 2.3 Flight NT011

We discuss results of the analysis for sounding NT011 in the main body of this paper and the results for NT029 and NT007 in

Appendices A and B. Figure 2 shows the vertical profile of NT011 measured on 15 August 2016 in Nainital. Figure 2a displays the air temperature from RS41 (green), $S_{\mathrm{liq\ RS41}}$ from RS41 (pink), calculated ice saturation ratio $S_{\mathrm{ice}}$ from CFH (blue) and calculated water saturation ratio $S_{\mathrm{liq,d}}$ and $S_{\mathrm{liq,f}}$ from CFH (dark and light purple, respectively); note that the condensate on the CFH mirror was forced to turn from dew to frost after the freezing cycle, at $T_{\mathrm{frost}}$ = -15 °C. Figure 2b shows the $H_2O$ mixing ratio, $\chi_{H_2O}$ ($\chi_{\mathrm{CFH}}$, red) and the Nainital campaign mean excluding the contaminated CFH measurements ($\langle\chi_{\mathrm{CFH}}\rangle$,

thin black line). Both panels (a) and (b) show 1-s data to illustrate the signal-to-noise ratio (S/N) of the CFH water vapour measurements. Figure 2c shows COBALD BSR at 940 nm (red line), BSR at 450 nm (blue) and CI (green).

The lower stratospheric water vapour mixing ratios were unrealistically large, due to contamination, becoming visible right above the CPT, but returning to reasonable stratospheric values below the balloon burst at 27 km altitude. COBALD identified two clouds, one very thin cirrus cloud directly below the CPT ($T_{\mathrm{air}}$ = -78 °C) and another geometrically and optically thick

cloud in the range 9 km to 13 km altitude and $T_{\mathrm{air}}$ = -20 °C to $T_{\mathrm{air}}$ = -50 °C. The lower cloud was sufficiently cold to make a high degree of glaciation likely (Korolev et al., 2003a). However, the CI observed between 9 km and 10 km altitude supports the existence of liquid in this cloud at these altitudes, with air temperature between -20 and -25 °C. Fully glaciated clouds are characterized by a very regular CI of about 20 with large ice particles, as evidenced in this cloud above 11 km altitude (see also Figure 10f in Brunamonti et al. (2018)). CI around 30 stems from the Mie oscillations in the transition regime and

thus from the presence of smaller and more monodisperse scatterers, most likely supercooled cloud droplets. Additionally, the BSR $\sim$ 1000 at 940 nm was about as high as can be observed with COBALD before the instrument would go into saturation. As indicated in Figure 2, only the lowermost 750 m of the cloud provided evidence for the existence of supercooled droplets at temperatures between -20 and -25 °C. Flights NT029 and NT007 showed a very similar cold mixed-phase cloud in terms of temperature, extent and altitude, while flight NT007 also showed a warmer mixed-phase cloud at lower altitude. Table 1

provides an overview of supercooled or mixed-phase cloud appearances in the three analysed soundings.

### 2.4 Modelling of mixed-phase clouds

Supercooled liquid droplets might lead to icing of the balloon, the payload and the intake tube during the passage through mixed-phase clouds, when they freeze upon contact with cold surfaces and lead to an icy surface coating. In contrast, passages





through fully glaciated cirrus are less critical, because the ice crystals likely bounce off the surface. Subsequently, we asked whether the Wegener-Bergeron-Findeisen process (Pruppacher and Klett, 1997; Korolev et al., 2017) provided enough time for the flights to encounter supercooled liquid droplets at these high altitudes and low temperatures: could the observed water and ice saturation conditions in NT011 from 9.25 km to 10 km altitude and at air temperatures of about $\sim$ -20 °C support liquid

droplets and how long would they survive?

Figure 3 shows the air temperature, the balloon ascent velocity, the saturation ratios $S_{\mathrm{ice}}$ and $S_{\mathrm{liq,f}}$ relative to ice and supercooled water from the CFH, respectively, and the $S_{\mathrm{liq}}$ from RS41, as well as the 940-nm BSR and CI for the mixed-phase cloud region of flight NT011. Similar figures for flights NT029 and NT007 can be found in the appendices (Figures A2 and B2). The lower part of the cloud (9.25 – 10 km) showed 5 – 10% ice supersaturation and 10% to 15% subsaturation over water.

This represented an unstable situation as the ice crystals grew at the expense of the liquid droplets, eventually resulting in a fully glaciated cloud with $S_{\mathrm{ice}}$ = 1 (Korolev et al., 2017). At altitudes above 10 km, the balloon encountered $S_{\mathrm{liq}} < 0.8$, i.e. liquid droplets were likely fully evaporated.

In order to estimate the glaciation time ($\tau_g$) we applied a simple evaporation model based on the solution of the diffusion equation for diffusive particle growth or evaporation

$$\frac{dr^2}{dt} = 2\,V_{\mathrm{H_2O}}\,D_g\,n_g\,(S-1),$$
(1)

where $r$ is the droplet or ice particle radius, $V_{\mathrm{H_2O}}$ is the volume of a $H_2O$ molecule in the condensed phase (liquid or ice), $D_g$ is the diffusivity of $H_2O$ molecules in air, $n_g$ is the number density of $H_2O$ molecules in the gas phase, and $S$ is the saturation ratio of water vapour over liquid water or ice. Equation 1 is a simplified form of Eq. 13-21 of Pruppacher and Klett (1997). The results of the simulations are presented in Figure 4.

We modelled the Bergeron-Findeisen process in these clouds by applying Eq. (1) to both the evaporating droplets ($S_{\mathrm{liq}} < 1$) and the growing ice crystals ($S_{\mathrm{ice}} > 1$). We chose the size distribution of the liquid droplets to be bimodal in order to approximate in-situ observations of broad droplet spectra in mixed-phase clouds (Korolev et al., 2017), with small liquid droplets $r_{\mathrm{liq,1}}$ = 10 µm, $n_{\mathrm{liq,1}}$ = 10 cm$^{-3}$ and big liquid droplets $r_{\mathrm{liq,2}}$ = 100 µm, $n_{\mathrm{liq,2}}$ = 0.003 cm$^{-3}$. We considered the number density of ice crystals to be consistent with ice nucleation particles (INP) at about 0.02 cm$^{-3}$ (DeMott et al., 2010), neglecting

secondary ice production processes, which might have enhanced ice number densities (Lawson et al., 2017), but would be highly uncertain. During the evolution of the mixed phase under the conditions characteristic for the lower end of the cloud in NT011 (9.25 – 10 km), the many small liquid droplets evaporated first, providing favourable conditions for the fewer large droplets, which would have needed about 20 minutes to finally evaporate, see Figure 4.

The low concentration of ice crystals and the bimodality of the liquid droplet distribution allowed the bigger droplets to exist

for a relative long period of time in a mildly subsaturated environment ($S_{\mathrm{liq}} \sim 0.90 - 0.85$). For the simulation, we assumed two different initial distributions: a lower and an upper estimate of liquid water content (LWC), see Table 1. The lower estimate was constrained by the amount of water required to sublimate in the stratosphere from the CFH intake tube in order to explain the observed contamination as determined by the Computational Fluid Dynamics simulations discussed in the next sections. The upper estimate was determined such that it would provide the sum of the amount of water sublimated in the stratosphere





plus the amount sublimated in the upper troposphere, the latter computed from the difference between $\chi_{RS41}$ and $\chi_{CFH}$. These estimates are discussed more thoroughly in Sections 5.2 and 5.3.

In Figure 4, we see that both simulations for lower (solid lines) and upper (dashed) estimates showed glaciation times of smaller droplet mode of $\tau_g \sim 6$ minutes and of the bigger droplet mode of $\tau_g \sim 17$ minutes. The overlap with the range of

observed $S_{liq}$ and $S_{ice}$ (shaded purple and blue, from Figure 3) lasted for about 7 minutes, demonstrating that the cloud at $9.25 - 10$ km in NT011 may have contained sufficient supercooled liquid to explain the contamination. The results of the simulation for flights NT029 and NT007 are shown in the appendices in Figures A3 and B3

These simulations make a causal relationship between the mixed-phase cloud and the CFH intake contamination plausible. In addition, the updraft cores of cold clouds observed by Lawson et al. (2017) over the Colorado and Wyoming high plains support

these assumptions, as these clouds did not experience secondary ice formation and significant concentrations of supercooled liquid in the form of small drops have survived temperatures as low as -37.5 °C. Observed ice crystal number densities were lower than 4 cm$^{-3}$ in clouds warmer than $T_{air} = $ -23 °C, increasing to 77 cm$^{-3}$ at $T_{air} = $-25 °C and to several hundred per cm$^{-3}$ at even lower temperatures. Thus, some of the clouds described by Lawson et al. (2017) contained fewer ice particles and more supercooled droplets than the example treated here.

## 3  Balloon pendulum movement

As we show below by means of computational fluid dynamics (CFD) simulations, the passage through clouds containing supercooled water leads to hardly any collisions of the droplets with the walls of the intake tube, if the airflow is parallel to the walls. Under those conditions, only the mirror extrusion causes collisions of larger droplets. Below the mirror extrusion, a recirculation cell might also cause some of the smaller droplets to collide, however this would hardly affect the humidity measurement

on the mirror. The situation changes dramatically when the air enters the intake tube at a finite angle, as would happen when pendulum oscillations of the balloon payload induce a component of the payload motion perpendicular to the tube walls. Such swinging or rotational motion has been documented in the literature (e.g., Kräuchi et al., 2016). Here, we approximated the balloon plus payload by a two body system connected by a weightless nylon cord, and quantified the oscillations in terms of the instantaneous displacement of the payload from the balloon path. We then used the displacement to calculate the tilt of

the payload relative to the flow and used the tilt angle and the associated horizontal velocity of the payload to quantitatively estimate the internal icing of the intake tube.

### 3.1  Pendulum oscillations derived from GPS data

We isolated the payload oscillations in relation to the balloon by removing the averaged trajectory of the payload. Figure 5a shows the horizontally projected trajectory of NT011, travelling first about 10 km northward in the troposphere and then about

40 km westward in the stratosphere before the balloon burst. The thick pink line shows the part of the trajectory, where the sonde flew through the cloud containing supercooled droplets, between 9.25 and 10 km altitude (see Figure 3). The contamination happened most likely in this segment of the flight.





Figure 5b zooms in on this cloudy section[1], showing the 1-s GPS data colour-coded by the ascent velocity in m s$^{-1}$. Figure 5c shows the residual payload motion relative to the balloon after 'detrending', i.e. subtracting the average trajectory of the payload (black dashed line in Figure 5b). We obtained the average payload trajectory or balloon trajectory by smoothing the payload trajectory with a moving average corresponding to the pendulum oscillation period, which we evaluated by two independent

methods. First, we considered the ideal pendulum oscillation frequency, $\omega = (g/L)^{1/2}$ where $L$ is the length of the pendulum, in our case 55 m and $g = 9.81$ m s$^{-2}$. This yielded the oscillation period $\tau = 2\pi/\omega$ =15 s. Second, we confirmed this result by means of a Fast Fourier Transform (FFT) analysis on the latitude and longitude detrended time series, see Appendix C. We concluded that independently of the moving average used to detrend the longitude and latitude used in the FFT, the oscillation period was $\tau \sim 16.6$ s. The same analysis was done for the clouds in flight NT029 and NT007 and the results are shown in

Figures A4 and B4 of the appendices.

Figure 5c also provides information on the degree to which the rubber balloon itself might contribute to the contamination. The approximate balloon sizes at launch and burst are depicted as circles with 1 m and 5 m radius, respectively. The oscillatory movement placed the payload typically far outside the balloon wake only sporadically penetrating the wake. The lack of periodic signs of contamination, rendered it unlikely that $H_2O$ collected by the rubber skin of the balloon contributed to the

observed contamination. However, this behaviour changed above $\sim 27$ km altitude, where the $H_2O$ partial pressure became sufficiently low and also the swing and rotation of the payload was weaker, so that the balloon outgassing started to dominate over the natural signal, leading to a systematic contamination in virtually every sounding (see Section 5.4).

Figure 6a shows a schematic of the balloon and payload as a two body system, and illustrates the displacement of the payload from under the balloon. From Figure 5c we see that the radial displacement $R$ of the payload in relation to the balloon position

for flight NT011 was typically larger than 5 m (only 4% of the measurements have $R < 5$ m). The corresponding tilt angle $\alpha$ is

$$\alpha(t) = \sin^{-1}\left(\frac{R(t)}{L}\right) > 5° \tag{2}$$

The maximum displacement was $R_{\max} \sim 23$ m, corresponding to a tilt angle $\alpha_{\max} \sim 25°$. On average, $\langle R \rangle \sim 15$ m and $\langle \alpha \rangle \sim 16°$, which represented a significant deviation from a flow through the tube parallel to the tube walls. The tilt angles

$\alpha$ of the payload in the mixed-phase cloud of flight NT029 were of the same order of magnitude of the ones observed in the cloud of flight NT011, while for flight NT007 these were much smaller, almost half. We believe this difference to stem form the different ascent velocities in the three flights.

Figure 6b shows how the flow through the CFH intake tube can be decomposed in flow parallel to the tube ($w_{||}$) and flow perpendicular to the tube ($w_\perp$). Figure 6b also shows how the different instruments are connected in the payload. The

impingement angle ($\beta$) of droplets onto the CFH intake tube was then partly determined by $w_\perp$ and $w_{||}$, and consequently $\alpha$.

---

[1]The coordinates were transformed from degrees lat/long to distances in km using the geographical distance equation from a spherical earth to a plane, $d = R_e \left[ (\Delta\phi)^2 + (\cos(\phi_m)\,\Delta\lambda)^2 \right]^{1/2}$ (Wikipedia, 2018), where the bottom of the cloud ($\lambda_0$, $\phi_0$) was taken as the origin (0,0) of this new coordinate system. Differences in longitude and latitude were calculated in radians as $\Delta\lambda(t) = \lambda(t) - \lambda_0$ and $\Delta\phi(t) = \phi(t) - \phi_0$, respectively. Distances $d$ were given in km, $R_e$ is the Earth radius (6371 km), and the mean latitude $\phi_m$ was taken as $\phi_0$





Moreover, the associated horizontal swinging or rotating motion led to additional sideways impingement, which we show to be even more important (see next section).

## 3.2 Impingement angles derived from payload motion

Impingement of droplets onto the walls of the intake tube was forced by two effects that caused an air flow in the 'horizontal 5 plane', i.e. the plane whose normal was the tube axis:

(i) the tube was tilted relative to the ascent flow, leading to the velocity $\boldsymbol{v}_{\perp,\,\mathbf{tilt}} = \boldsymbol{w}_{\perp}$;

(ii) the tube itself had a horizontal velocity $\boldsymbol{v}_{\perp,\,\mathbf{rot}}$ caused by the swinging or rotational motion of the payload;

The vector sum of (i) and (ii) gave the total velocity perpendicular to the tube walls $\boldsymbol{v}_{\perp} = \boldsymbol{v}_{\perp,\,\mathbf{tilt}} + \boldsymbol{v}_{\perp,\,\mathbf{rot}}$, refer to Equation (D4). Appendix D provides more details of the vector relations. In addition, we took into account the possibility of droplet 10 impingement on the mirror holder in the centre of the tube, even when the flow was perfectly aligned to the tube, but compared to (i) and (ii) this was a smaller contribution because larger droplets impinged already at the beginning of the tube and many of the smaller ones, which made it to the middle of the tube, were able to curve around the mirror holder and avoid contact.

Figure 5c shows that the residual motion of the payload resembles a circular motion with radius $R$ = 15 m. Here, we only highlight the relevant magnitudes, but we provide a full treatment in Appendix D. The perpendicular velocity associated with 15 the tube tilt ($w_{\perp} = v_{\perp,\,\mathrm{tilt}} = w\sin\alpha$) can be determined from the tilt angle $\alpha$ and the ascent velocity $w \sim 7.5$ m s$^{-1}$ (Figure 3a). Eq. 2 with $R(t)$ = 15 m and $L$ = 55 m yields $\alpha$ = 16° and $|v_{\perp,\,\mathrm{tilt}}|$ = 2.1 m s$^{-1}$. The perpendicular velocity associated with the payload rotational movement ($v_{\perp,\,\mathrm{rot}}$) can be calculated from the distance between consecutive measurements after detrending based on the GPS position received every second. Figure 5c shows that $|v_{\mathrm{h,\,rot}}|$ can be as big as 10 m s$^{-1}$ when the payload traverses the equilibrium point, straight below the balloon, or as small as 2 m s$^{-1}$ far from the equilibrium point. 20 The horizontal motion of the payload leads to generally more impingement than the tilt of the tube, also for circulation around the equilibrium point. Here, the radially directed tilt contribution (2.1 m s$^{-1}$) and the circular progression added as sum of orthogonal vectors, increases the typical 5 m s$^{-1}$ circular speed (see Figure 5c) to only 5.4 m s$^{-1}$, i.e. less than 10%.

After accounting for the direction of movement when combining tilt and rotation, the impingement angle was calculated from the perpendicular velocities sum ($\boldsymbol{v}_{\perp}$) and the parallel component of the inlet flow ($\boldsymbol{w}_{||}$) as shown in Figure 6c. As the 25 horizontal impingement speed could be as high as 10 m s$^{-1}$, this corresponded to a maximum impingement angle $\beta$ = 53°. Such large impingement angles are the reason why CFH flying through mixed-phase clouds encounters a large risk of droplet collisions and freezing, accumulating potentially thick ice layers inside the intake tube, which render further measurements in the stratosphere either impossible or possible only after a long recovery period of the instrument (i.e., after the ice sublimated). As result from the full numerical treatment of the impingement in Appendix D, Figure 7 shows the probability density functions 30 (pdf) of the perpendicular velocity ($v_{\perp}$) to the intake tube walls, parallel component of the ascent velocity ($w_{||}$) and the impingement angle ($\beta$) as derived for the intake tube in the 9.25-10.0 km cloud section in flight NT011. Similar figures are shown for flight NT029 and NT007 in appendices A and B. Perpendicular velocities were smaller for flight NT007 but, as the



ascent velocity was also smaller in this flight, the impingement angles were equivalent to those observed in flight NT011 and
NT029.

## 4   Computational fluid dynamic simulations

Computational Fluid Dynamics (CFD) tools have become commonly used in environmental studies, e.g. for error estimation
of lidar and sodar Doppler beam swinging measurements in wakes of wind turbines (Lundquist et al., 2015), in new designs
of photooxidation flow tube reactors (Huang et al., 2017), or to improve vehicle-based wind measurements (Hanlon and Risk,
2018). Here, we used CFD to estimate collision efficiencies of liquid droplets with different sizes encountering the CFH intake
tube under various impingement angles in order to understand first the ice build-up and second its sublimation from the icy
intake to the passing air flow. We used the academic version of FLUENT and ANSYS Workbench 14.5 Release (ANSYS,
10   2012).

### 4.1   Geometry and mesh

By means of ANSYS Workbench, a mesh was developed mapping the intake tube geometry and providing the optimal geomet-
ric coverage. The CFH intake tube geometry was as described by Vömel et al. (2007c): a 2.5 cm diameter cylinder that extends
for 34 cm. The walls of the intake tube have a thickness of 25 μm, but are approximated as infinitely thin. At the centre of the
tube the mirror head is mapped by a cylinder extruding from the intake tube wall. The mirror extrusion is 7 mm in diameter,
1.25 cm from the wall, oriented parallel to the flow.

The mesh is shown in Figure 8. As mesh assembly method we used 'cutcell', which provides cuboid shaped elements aligned
in the flow direction. Simulations had to cover conditions from the lower troposphere, where the liquid and mixed-phase clouds
occurred, to the lower stratosphere where the sublimation of ice from the intake walls took place. This required coping with
Reynolds numbers (Re) of the order of 5000 in the cloud (i.e. turbulent flow inside the tube) to 300 in the stratosphere (i.e.
laminar flow), accompanied by a transition around $\mathrm{Re} \sim 2300$ from turbulent to laminar regimes:

$$\mathrm{Re} = \frac{\rho \, v \, L}{\mu}, \tag{3}$$

where $\rho$ is the fluid's density in $\mathrm{kg\,m^{-3}}$ (here of air), $v$ is the fluid's velocity in $\mathrm{m\,s^{-1}}$ (relative to the intake tube), $L$ is a
characteristic linear dimension in m (here the tube diameter) and $\mu$ is the fluid's dynamic viscosity in $\mathrm{kg\,m^{-1}\,s^{-1}}$. We were
especially interested in the near wall effects, since the sublimation and the collision efficiency were evaluated near the wall. To
enhance the mesh description near the wall, the first layer thickness is 0.2 mm. The subsequent layers grow in thickness at a
rate of 1.2 for a total of 5 layers, before the scheme changes from radial to Cartesian coordinates with grid spacing of 1.5 mm.

### 4.2   FLUENT computational fluid dynamics software

We used a 3D steady state pressure-based solver. As recommended for wall-affected flow with small Reynolds numbers, where
turbulent resolution near the wall is important, we used an SST (shear stress transport) $k - \omega$ model (CFDWiki, 2011; ANSYS,





2012). The fluid material, air, was treated as a three substance mixture of $N_2$, $O_2$ and $H_2O$. We specified how FLUENT computes the material properties, namely calculating density ($\rho$) using an incompressible ideal gas law

$$\rho = \frac{p_{op}}{RT \sum_i \frac{m_i}{M_i}}, \tag{4}$$

where $p_{op}$ is the simulation-defined operating pressure in Pa, $R$ is the ideal gas constant, $T$ is the absolute temperature, $m_i$ and

$M_i$ are the mass fraction and molar mass of species $i$, respectively. Heat capacity ($c_p$) was calculated using a FLUENT-defined mixing law:

$$c_p = \sum_i m_i c_{p,i}. \tag{5}$$

In the dilute approximation scheme, the mass diffusion flux of a chemical species in a mixture was calculated according to Fick's law:

$$J_i = \rho D_i \frac{\partial m_i}{\partial x}, \tag{6}$$

where $D_i$ is the diffusion coefficient of species $i$ in the mixture. This relation is strictly valid when the mixture composition stays approximately constant and the mass fraction $m_i$ of a species is much smaller than 1. The amount of water expected in the simulations was less than 1000 ppmv, therefore the dilute approximation for the diffusion of water vapour in air, $i = H_2O$, was an accurate description.

The temperature and pressure dependencies of the diffusion coefficient of $H_2O$ in air were given by Pruppacher and Klett (1997)

$$D = 0.211 \frac{\mathrm{cm}^2}{\mathrm{s}} \left(\frac{T}{T_0}\right)^{1.96} \left(\frac{p_0}{p}\right), \tag{7}$$

where $T_0 = 273.15$ K and $p_0 = 1013.25$ hPa. For the viscosity and thermal conductivity no mixture laws were considered. The values of viscosity and thermal conductivity were derived from a linear fit to air viscosity and thermal conductivity of dry air

(EngineeringToolbox, 2005). Air viscosity was $\mu_a(T) = (0.0545 \times (T/K) + 2.203) \times 10^{-6}$ in kg m$^{-1}$ s$^{-1}$ and air thermal conductivity was $k_a = 8.06 \times 10^{-5} \times (T/K) + 2.02 \times 10^{-3}$ in W m$^{-1}$ K$^{-1}$ both for $T \in (193\ \mathrm{K}, 300\ \mathrm{K})$. According to kinetic gas theory both properties are only weakly pressure-dependent (neglected here).

A velocity-inlet and a pressure-outlet boundary condition were defined for the intake tube. For the velocity-inlet boundary conditions, it was possible to define the velocity magnitude and direction, turbulence intensity and temperature.

**4.2.1  Velocity and flow profiles**

Figure 9a-b show two examples of velocity profiles computed by FLUENT for two pairs of pressures and temperatures as they occurred in NT011, $p = 310$ hPa and $T$ = -20 °C and $p = 33$ hPa and $T$ = -58.7 °C. The two examples were done for the same inlet velocity of 5 m s$^{-1}$. In the lower pressure case, the Reynolds number was low and the flow was laminar. In the higher pressure case the Reynolds number was higher than 2300 and the flow was turbulent. As expected for the flow in a cylindrical





tube, the flow velocity decreased towards the tube walls, became zero at the wall and in return accelerated at the centre of the tube, thus conserving mass flux. For our simulations, we took the balloon ascent velocity as the velocity of the flow entering the intake tube at the top plane.

The mirror extrusion slowed the flow upstream, created a recirculation region downstream and accelerated the flow in front
of the mirror. The flow accelerated up to 150% of the fully developed flow velocity in the tube centre. In the troposphere, the medium was denser and the flow was in the turbulent regime. Turbulent flows develop faster into a fully developed regime, see Figure 9a-b.

### 4.2.2  Discrete phase model

We used FLUENT's discrete phase module to compute the collision efficiency for water droplets entering the tube together
with the air at some impingement angle. The droplets were accelerated in the same direction as the air flow when they entered the tube and either managed to avoid a collision with the wall or hit it at some distance down the tube. We injected one particle through each of the cells at the top inlet plane and repeated this procedure for droplets of different sizes. For each of the mixed-phase cloud simulations, we defined the droplet diameter, impingement angle $\beta$ and velocity magnitude. At the top of the tube, the impingement angles and velocities of the droplets were assumed to be identical to the air flow.

The simulations in Figure 9c were run with NT011 cloud conditions, $p$ = 310 hPa and $T$ = -20 °C. The velocity at the intake tube inlet surface was 7.5 m s$^{-1}$ in the parallel component (z-direction) and 6 m s$^{-1}$ in the perpendicular component (x-direction), which corresponded to an impingement angle of about 39°. For clarity, Figure 9c displays only one every sixth droplet trajectory. Only the first 7 cm of the intake tube are shown. As expected the air flow affected different size droplets differently. Smaller droplets had less inertia, hence tended to stay within the air flow, avoiding collisions with the tube's wall,
while bigger droplets (with higher inertia) could not follow the streamlines and collided with the walls. Most 10 μm radius droplets avoided collision, only the droplets entering very close to the intake tube wall collided. The bigger droplets to some extent also re-adjusted with the flow, but many of them collided within the first 5 cm of the intake tube. Above 70 μm droplet radius there was no dependence of the total collision efficiency on droplet size due to their large inertia.

In order to calculate the build-up of ice by impaction and considering how the injection of liquid droplets was set up in
FLUENT, with one droplet per cell in the top inlet plane, we had to account for the mesh cell surface density. As discussed above, the cell surface density was higher closer to the intake tube wall (see Figure 8c). Therefore, we normalized all collision efficiency results to the top inlet plane cell surface density, removing the effect of the mesh density from the results. The collision efficiency results are provided in Figures 11, A6, B6 and B7 for the different mixed-phase clouds considered.

### 4.2.3  Species transport

Figure 10 shows FLUENT simulations of the water vapour mixing ratios inside the tube resulting from different degrees of icing of the intake tube induced by the collisions calculated in Section 4.2.2. We simulated the sublimation of ice into the gas flow by assuming the cells adjacent to the icy wall to be saturated with respect to ice (using the vapour pressure parameterisation of Murphy and Koop (2005)). The tube was assumed to have the same temperature as the air flow (Vömel et al., 2007b). The





tube wall was divided in ring sections and each was controlled separately. FLUENT calculated the distribution of the water vapour through the intake tube with a combination of molecular diffusivity (Eq. 6) and eddy diffusivity.

Figure 10 displays color-coded vapour mixing ratios for different assumptions on the size of ice-covered area in the upstream part of the tube. For these simulations we took stratospheric conditions with $p = 33\,\mathrm{hPa}$ and $T = $ -58.7 $^\circ$C. The flow speed was
4.7 m s$^{-1}$ normal to the inlet surface. For cases (a)-(d), the ice covered the full inner circumference of the tube and extended for 15 cm and 5 cm from the rim into the flow direction and for 1 cm and 1.5 mm from 4 cm from the rim into the flow direction, respectively. Panel (e) shows the effect of a rotationally asymmetric patch, which covered one eight of the intake circumference and extended for 1 cm (between 4 cm and 5 cm), as an example of a case where a single larger hydrometeor hit the tube or of an ice layer, which sublimated in-homogeneously. As a general relationship, a larger icy coverage extent resulted in higher
contamination. However, the relation between ice coverage in the tube and contamination was not linear. In dry stratospheric air ($S_{\mathrm{ice}} \sim 0.01$), a 15 cm long ice cover achieved $S_{\mathrm{ice}} \sim 0.6$ on average in the tube's volume, while a 1 cm long ice cover still achieved $S_{\mathrm{ice}} \sim 0.15$ on average in the tube's volume.

Figure 10 shows how contamination diffuses from the tube walls towards the centre of the tube. Over the length of the tube (34 cm) the contamination homogenised, whereas at the position of the mirror, 17 cm from the top of the tube, the $H_2O$
flow was not yet homogeneous. This internal gradient resulted from the residence time $\tau \sim 0.07$ s of the air inside the tube (34 cm long and moving with 4.7 m s$^{-1}$), so that a molecular diffusivity 4.0 cm$^2$ s$^{-1}$ (see Eq. 7) allowed the $H_2O$ molecules to travel on average only a distance $(\tau D)^{1/2} \sim 0.5$ cm towards the tube centre. The resulting boundary layer was well visible in the upper parts of the tubes shown in Figure 10. Any further diffusion can be attributed to eddy diffusivity, which the turbulence scheme of FLUENT was designed to properly determine. In this range of the stratosphere, eddy diffusivity was
about 5000 cm$^2$ s$^{-1}$ (Massie and Hunten, 1981); however, this value applies to the large-scale stratospheric dimensions, not to the small dimensions inside the tube. The effective diffusivity was somewhere between the molecular and the free stratospheric value, as calculated by FLUENT.

In this study, we preferred mixing ratios averaged over the entire volume of the intake tube instead of area averaged water vapour mixing ratios at the mirror surface. The latter were 60% to 50% smaller than the former for the same simulation. We
did not believe the simulated area averaged water vapour mixing ratios at the mirror surface to be an accurate description of the air mass experienced by the mirror in real flight conditions. The cell closest to the mirror surface in the simulation was 0.2 mm thick and the area averaged velocity for these cells was 0 m s$^{-1}$. We expect the mirror to experience a better mixed and larger amount of the passing air flow. We investigated the influence of an inlet air flow not parallel to the tube. Although a different impingement angle than 0$^\circ$ disturbed the flow in the first centimetres of the tube, the flow recovered quickly. The uptake of
water vapour from the icy wall into the air flow in these first few centimetres became radially asymmetric. However, over the length of the tube it homogenised and on average we obtained the same level of contamination independent of the impingement angle. Therefore, for the stratospheric and upper tropospheric ice sublimation simulations in Section 5.2 and 5.3 respectively, we compared observed contaminated $H_2O$ mixing ratios to simulated volume averaged $H_2O$ mixing ratios ($\langle \chi_{H_2O} \rangle_{\mathrm{Vol}}$) and only considered flow parallel to the tube. Evidently, this was in contrast to the hydrometeor simulations, for which even small
impingement angles made a big difference.





## 5   Results

### 5.1   Hydrometeors freezing efficiency derived from impingement angles

To estimate the collision efficiency of supercooled droplets during the cloud passage in flight NT011, we performed 10 FLUENT simulations as described in Section 4.2.2, using $w_{||}$ = 7.5 m s$^{-1}$ for the velocity component parallel to the tube
(see Figure 6b and 7b). For each of the ten FLUENT simulations we took a different perpendicular velocity $v_{\perp}$ to the tube walls as shown in Figure 7a for steps of 1, 2, 3, .. to 10 m s$^{-1}$.

Figure 11 displays the computed collision efficiencies for perpendicular velocities ($v_{\perp}$) increasing from (a) to (j). The panels also list the corresponding impingement angle ($\beta$). For each $v_{\perp}$, we considered droplet sizes of 100 μm and 50 μm radius. In each panel, the first 5 horizontal bars represent the first 5 cm of the tube, the 6$^{th}$ bar represents the rest of the tube (including
the mirror holder) and the 7$^{th}$ bar is the sum of all the above, representing the probability of the droplet hitting the tube at all. Differences to 100% represent droplet percentage that escapes the intake tube. Figure 11k shows the collision efficiencies weighted sum by the occurrence probability (pdf) of each $v_{\perp}$ as calculated for the NT011 cloud and shown in Figure 7a.

For the droplet sizes and impingement angles considered, 100% of the 100 μm radius droplets and 96% of the 50 μm radius droplets collided with the intake tube wall and more than 90% of these collided within the first 4 cm. Figure 11k also lists the
calculated thicknesses of the ice layer in the first 5 cm of the intake tube after passing the cloud, assuming an even coverage of the intake tube inner surface and taking into consideration the simulated collision efficiencies and the upper and lower estimate of liquid water content (LWC) that we discussed in Section 2.4. The first value listed for each horizontal bar in Figure 11k refers to the lower LWC estimate and the second to the upper LWC estimate.

Figure 11 shows that the combination of high impingement angles and big droplet sizes caused an ice layer to accumulate at
the top of the intake tube, in the first 5 cm. Smaller impingement angles, up to 15° caused a more even coverage over the entire length of the intake tube (but occurred much less frequently, see Figure 7c). As the layer remained quite thin (in the range of 1 to 4 μm), representing less than 1‰ of the intake tube radius, the ice layer did not affect the inlet flow. However, it had a detrimental influence on the water vapour measurement in the stratosphere.

Freezing efficiencies were also simulated for the mixed-phase clouds in flights NT007 and NT029 (Figures A6, B6 and B7
in Appendices A and B). One of the clouds encountered during flight NT007 (Figure B2a-b) was warmer, allowing also smaller droplets to exist, which we considered in Figure B6. The total freezing efficiency of the 10 μm radius droplets in the entire tube was smaller than 50%. The collisions happened mainly in the first 2 cm of the intake tube and below the mirror holder.

### 5.2   Contaminated water vapour measurements in the stratosphere

#### 5.2.1   Sublimation and sublimated water estimation

For the simulations of sublimation of ice in the intake tube we used FLUENT in the configuration described in Section 4.2.3. We defined three scenarios of ice coverage of the intake tube as shown in Figure 10(a-c): coatings of 15 cm or 5 cm length starting at the rim of the intake tube, and 1 cm coating starting 4 cm into the intake tube. We ran simulations approximately





every km in the stratosphere driven by measurements of temperature, pressure, ascent velocity, and background water vapour mixing ratio averaged over 1-km intervals. In Figure 12, we show stratospheric measurements during flight NT011 and the FLUENT simulations results. The values used as input in the simulations are shown in the stratospheric part of Table 2 as well as the simulations results. Figure 12a displays the air temperature (green), the average Naintial 2016 summer campaign

air temperature (dashed black) and the ascent velocity parallel to the intake tube ($w_{||}$) (black). Due to high variability in the ascent velocity, we calculated the standard deviation for each ascent velocity averaged point, shown in the graph as grey dots and error bars and performed FLUENT simulations to investigate the influence of the ascent velocity variability. We concluded that $\pm 2\,\mathrm{m\,s^{-1}}$ had no significant impact on the simulated volume averaged water vapour mixing ratio in the intake tube.

In Figure 12b, we show $\chi_{H_2O}$ from CFH ($\chi_{CFH}$, red) and the average $\chi_{H_2O}$ for the Nainital 2016 summer campaign

($\langle\chi_{CFH}\rangle$, dashed black). We also show the saturation $\chi_{H_2O}$ for flight NT011 calculated from the air temperature ($\chi_{sat}$, dashed red). The $\langle\chi_{H_2O}\rangle_{Vol}$ for the 15 cm, 5 cm and 1 cm intake tube ice coverage are shown as different coloured triangles. $\langle\chi_{H_2O}\rangle_{Vol}$ for other ice coverage configurations, such as thinner rings and radially asymmetric patches, are shown at higher altitude as the measurement recovers from contamination.

From the comparison of the simulation results for $\chi_{H_2O}$ in Figure 12b, we concluded that the simulations with 5 cm ice

coverage of the intake tube yielded the best description of the observations. This result was consistent with the collision efficiency results of Section 5.1. When the observed $\chi_{H_2O}$ decreased, above 22 km altitude, the 5 cm simulation started to overestimate $\chi_{H_2O}$. As the ice coverage decreased, the inlet air flow was exposed to a smaller ice surface and was less hydrated, until, no ice surface was left and the instrument observed ambient $\chi_{H_2O}$. The transition from 5 cm ice wall coverage was very fast. At 22 km altitude, the 5 cm simulation still matched the observation, while one km higher at 23 km altitude,

we were able to match the observation to the 0.45 cm simulation. At 25 km altitude, we considered the measurement to be recovered. Figures A7 and B8 in the appendices show similar results for the stratosphere of flights NT029 and NT007.

Considering the water vapour to be well mixed within the intake tube, knowing the pressure, temperature and the intake tube volume, and having a reference water vapour measurement, we could estimate the total water sublimated in the stratosphere,

$$\sum_{i}^{N} \frac{p_{\mathrm{air}\,i}\,100\,V_{\mathrm{tube}}}{R\,T_{\mathrm{air}\,i}}\left(\chi_{\mathrm{CFH}\,i} - \chi_{\mathrm{ref}\,i}\right) \tag{8a}$$

$$N = \frac{h_{\mathrm{burst}} - h_{\mathrm{CPT}}}{100\,l_{\mathrm{tube}}} \tag{8b}$$

We did the integration in intervals of 100 intake tube volumes between the CPT and balloon burst, because of the measurement resolution. Since the contamination disappeared before balloon burst, the total water sublimated in the stratosphere was the total water frozen in the intake tube during the traversing of the mixed-phase cloud, depending on the conditions above the mixed-phase cloud in the troposphere.

Table 3 lists results for the stratospheric integration of water vapour for NT007, NT011 and NT029. A total of 4.35 mg of water sublimated from the intake tube in the stratosphere in flight NT011 (derived from the difference between $\chi_{CFH}$ and $\langle\chi_{CFH}\rangle$ in Figure 12b, see Formula 8). We considered this the lower estimate of water that froze in the intake tube during the ascent through the mixed-phase cloud, because a small part of the condensate might have already sublimated between the





mixed-phase cloud and the tropopause. The cloud extent was 750 m and the estimated collision and freezing efficiency of the hydrometeors in the cloud was 100%, so the lower estimate of liquid water content (LWC) in the cloud was 0.011 $\mathrm{g\,m^{-3}}$. This is very little LWC for a mixed-phase cloud, so we concluded that the mixed-phase was almost completely glaciated. We used this value as the lower estimate for LWC for the cloud simulation in Section 2.4. During the two other flights, more water
vapour sublimated in the stratosphere.

### 5.2.2  Ice layer evolution

As we saw from the collision efficiency results in Figure 11k, the thickness of the ice coverage inside the tube was not uniform in the flow direction. Subsequently we show how this non-uniformity influences the sublimation and the lifetime of the ice coverage in the intake tube. For this, we computed the potential of ice at a certain position downstream in the tube to hydrate
the passing air, thereby lowering the air subsaturation and, hence, slowing the sublimation of the ice further downstream. We ran FLUENT sublimation simulations with different ice coverage configurations of the intake tube for three altitudes ($p$ = 15 hPa and $T$ = -51.4 °C; $p$ = 25 hPa and $T$ = -53.6 °C; $p$ = 39 hPa and $T$ = -59.2 °C). The results are listed in Table 4.

For the simulations named 'isolated' in Table 4, we considered isolated 1 cm-long rings in the flow direction covering the entire inner circumference of the intake tube. These 1 cm-long rings started at different distances from the rim of the intake tube
down to 4 cm. With these simulations, we could compare the potential of an isolated ice layer to hydrate passing stratospheric dry air at different distances from the rim of the intake tube. Results of these simulations are given as extra $\mathrm{H_2O}$ mixing ratio from the reference, which was $\chi_{\mathrm{H_2O}}$ ~4 ppmv, and as extra ice saturation. The $\chi_{\mathrm{H_2O}}$ for ice saturation for each of the simulation conditions is also given in Table 4. The 'in group' type of simulations considered ice coverages of different length all starting at the rim of the intake tube. With these simulations we could estimate the added contribution of an ice ring at
a certain distance from the rim, once the passing air has already experienced a certain level of contamination caused by the ice on the tube wall above. For these simulations, extra $\chi_{\mathrm{H_2O}}$ and extra $\mathrm{S_{ice}}$ were calculated as differences from subsequent simulations. From 5 cm from the rim of the intake tube, the isolated rings become 2-cm long down to 15 cm from the intake tube rim and the ice layers extending from the intake tube rim increase length in 2 cm steps also down to 15 cm from the rim of the intake tube. The results for these simulations are shown in Table 5, but only for one of the pressure and temperature
pairs used in the simulations shown in Table 4. This analysis confirms that the first centimetre of the intake tube was the most efficient at hydrating the passing air compared to ice downstream. When the passing air had already been in contact with an icy surface, the hydration efficiency of the subsequent ice layers reduced strongly.

The lower layers more than 5 cm inside the tube had the smallest contribution to the air hydration, but they sublimated first after passing the cloud because the ice deposition from the hydrometeor collisions was also small (Figure 11k). Of the
top layers, more hit by the impinging hydrometers during the mixed-phase cloud, we expected the first layer to be the first to sublimate. After the first centimetre of the intake tube became ice-free, the strongest contamination arose from the next layer downstream. The layer between 1 cm and 2 cm was also the thickest layer (Figure 11k), i.e. it had an extended lifetime. The layers below were thinner but they also contributed with some water vapour to the hydration of the flow. Independent of which of the layers between 1 cm and 4 cm from the rim of the intake tube evaporated next, once isolated, any 1 cm long





ring in this region contributed very similar amounts of water vapour to dry incoming stratospheric air. This suggested that the contamination stayed significant as long as some ice was in the tube, but thereafter disappeared readily. This was confirmed by the ice patch and thinner layers (0.45 or 0.15 cm) simulations. Figure 12b shows these results at 23.5 km altitude.

In summary, the ice deeper inside the tube evaporated less quickly, but it nevertheless disappeared first, because only a thin layer of ice was deposited there when traversing the cloud. Thereafter, the ice layer sublimated fastest from the top of the intake tube, because hydration was more efficient when the air is at stratospheric dryness. Figure 12b reveals that the FLUENT simulations together with reasonable assumptions about the initial contamination in the mixed-phase cloud can achieve a good agreement with the measurements.

### 5.3 Considerations regarding the upper troposphere

The contamination in the stratosphere was a remarkable feature and was relatively easy to spot since the expected water vapour mixing ratio values were in a well defined range 4-8 ppmv. Sublimation may also occur in the upper troposphere after passing through mixed-phase clouds, although it might be harder to identify. For the stratospheric contamination we had a readily-available reference, namely the mean uncontaminated campaign measurements. Water vapour in the stratosphere has very limited day-by-day variability, whereas tropospheric water vapour is extremely variable. We investigated whether the relative

humidity measurement by the RS41 radio sonde could serve as reference. Brunamonti et al. (2019) found the RS41 to have in average a dry bias in comparison with CFH in the upper troposphere during StratoClim. However, in a flight by flight comparison, when CFH was contaminated, it was not clear whether RS41 had a dry bias or CFH measured a too high humidity. Therefore, we used the RS41 water vapour measurement as reference for the analysis of the CFH contamination in the upper troposphere.

Figure 2 shows that the profile of NT011 between the top of the lower cloud and the cirrus cloud at the tropopause was sub-saturated. In Figure 13, we provide a detailed view of this region of the flight (13-17.5 km altitude). Figure 13(a-b) are analogous to Figure 12(a-b) with the exception that in panel (b) we do not show $\langle \chi_{CFH} \rangle$, but $\chi_{RS41}$ of NT011 (orange). Figure 13c shows the same variables as Figure 3b. The dry bias of RS41 relative to CFH is noticeable in the region between 13.5 km and 17 km, right up to the CPT. To understand if the proposed mechanism of water vapour sublimation from an

ice layer at the top of the intake tube might explain the dry/wet bias observed, we ran FLUENT simulations at four selected altitudes (see symbols in Figure 13b). Pressure, temperature, inlet velocity and background water vapour from RS41 used for the FLUENT simulations are presented in the tropospheric part of Table 2, where the simulation results are again presented as $\langle \chi_{H_2O} \rangle_{Vol}$.

At 14.0, 14.6 and 15.9 km altitude there was a significant difference between CFH and RS41 $\chi_{H_2O}$ (see tropospheric part

of Table 2). Observed $S_{liq}$ was below 30% and $S_{ice}$ was below 70%. The difference in water vapour mixing ratio for the two instruments was about 50%-70% which can not be accounted for by the estimated 10% uncertainty of the CFH measurement (Vömel et al., 2007b), also not by the estimated 3-9% dry bias of RS41 relative to CFH (Brunamonti et al., 2019), nor by a combination of both. At 15.2 km altitude, the observed $\chi_{H_2O}$ for RS41 was within the CFH uncertainty, and $S_{liq}$ was 30% and $S_{ice}$ was 70%.





In Figure 13b, at 14 and 14.6 km altitude, the simulations for the 15 cm ice coverage of the intake tube could account for the extra water vapour measured by the CFH. The location at 15.2 km altitude showed the limit of the FLUENT simulations. The simulation considering 1 cm ice coverage matches the CFH observation and the other two ice coverages considered (5 cm and 15 cm) over-estimated the CFH observation. Although the observations did not show ice saturation, the dilute approximation

used in the FLUENT simulation (see Section 4.2) is no longer valid for $S_{ice} \geq 70\%$, and the simulations over-estimated how much water vapour sublimated into the air flow. The CFH observation at 15.9 km altitude could be due to the presence of a 5 cm ice layer at the top of the intake tube. The lower 10 cm of the ice layer cloud have evaporated between the lower observation at 14.5 km height and the observation at 15.9 km.

In flight NT007 (Figure B9 in Appendix B) RS41 measured lower water vapour mixing ratio than the CFH in the upper

troposphere as expected, but, once $S_{ice}$ approached 1 at 13.8 km altitude, CFH under-estimated the water vapour measurement in relation to the RS41. We suppose that the icy intake tube top was having the opposite effect in contaminating the CFH measurement. It was depleting the gas phase water vapour, and growing the ice coverage, reducing the supersaturation which in a clean intake tube case would have been observed.

We integrated how much extra water vapour was measured by the CFH in relation to the RS41 in the upper troposphere

between 13.5 and 17 km altitude using Formula (8). CFH measured more 1.45 mg of water in this interval than the RS41. This was about 1/3 of the ice sublimating in the stratosphere and could be additional water that accumulated during the mixed-phase cloud and sublimated in the upper-troposphere. Adding this water to the water sublimated in the stratosphere, gave a total of 5.8 mg of water sublimated in flight NT011. Table 3 shows total integrated water vapour and integrated water vapour for the stratosphere and upper troposphere for flights NT007, NT011 and NT029. In flights NT029, there was a cirrus clouds in the

upper troposphere, so the water vapour integration is done below and above the cirrus cloud. From the total integrated water vapour for the different flights, we calculated upper estimates of LWC for the cloud simulation in Section 2.4. These values are shown in Table 1 together with lower estimates of LWC.

## 5.4    Other types of contamination

Besides the intake tube, there may be other sources of contamination, such as the balloon envelope, the nylon cord, or the

instrumental payload. Here, we differentiate the various contamination sources. We did two extra related studies with FLUENT. The principle of the implemented simulations was identical to what was described in Section 4.2.

### 5.4.1    Balloon envelope

We ran FLUENT simulations for typical tropospheric and stratospheric conditions shown in Table 6. We used a new mesh, for which the balloons radius changed with atmospheric conditions. We considered an initial balloon size of 1 m radius at 800 hPa

and 25 °C, approximately corresponding to the launching conditions at Nainital during the summer season. As the payload ascends, the balloon radius increases, see Table 6. In the simulation, we placed the CFH package 55 m below the balloon centre, considered two different ascent velocities (4 and 7 m s$^{-1}$), and the entire surface of the balloon to be covered with an





ice layer implemented similarly as the ice covered intake tube. The simulation domain extended 5 m from the balloon surface in every direction and 5 m below the CFH package, see Figure 14.

In Figure 14(a-b), we see the water vapour mixing ratio color coded for the balloon and payload ascending at $7\,\mathrm{m\,s^{-1}}$ at the 50-hPa and 20-hPa levels, respectively. At the 50-hPa level, the excess water vapour due to balloon contamination was still

moderate. The water vapour mixing ratio observed 55 m below the balloon at the payload level was within the stratosphere natural variability (4-8 ppmv). At the 20-hPa level, the effect of contamination by the balloon was large. The enhanced water vapour mixing ratios in the wake of the balloon extended by 6 m in radius at the payload level and reached up to 100 ppmv. The contamination values directly below the balloon were upper limits of contamination. If we considered the payload oscillation at these altitudes, we would see that the CFH was consistently out of the balloon wake, where there was no contamination, in

the case at 50 hPa (see Figure 14c). At the 20-hPa level, the payload oscillation showed a nearly perfect circular movement around the balloon, with a displacement between 5 and 10 m (see Figure 14d), where $\chi_{\mathrm{H_2O}}$ was less than 10 ppmv (see Figure 14b). At 10 hPa, not shown, the balloon radius was larger and consequently, so was the wake of the balloon. At this pressure level, the payload showed a similar displacement from under the balloon as observed in Figure 14d. The $\chi_{\mathrm{H_2O}}$ in this region at this pressure level was contaminated. If the payload was 5 m displaced from under the balloon $\chi_{\mathrm{H_2O}}$ would be 40 ppmv, 10 m

displaced from under the balloon $\chi_{\mathrm{H_2O}}$ would be 5 ppmv

Figure 15 compares the expected magnitude of the contamination from the balloon envelope with the contaminated $\chi_{\mathrm{H_2O}}$ observations by the CFH in StratoClim 16/17. Figure 15 shows all the contaminated profiles which have been shown in Figure 1, and the campaign season averages ( $\langle\chi_{\mathrm{CFH}}\rangle_{\mathrm{NT}}$ in black and $\langle\chi_{\mathrm{CFH}}\rangle_{\mathrm{DK}}$ in grey). The light red shaded region highlights expected $\chi_{\mathrm{H_2O}}$ if there was contamination from the balloon skin and the payload travelled directly below the balloon or within

a 5 m radius centred 55 m below the balloon. The dark red region identifies $\chi_{\mathrm{H_2O}}$ for balloon contamination if the payload stayed outside this 5 m radius circular region centred 55 m directly below the balloon. The contaminated $\chi_{\mathrm{H_2O}}$ observed during StratoClim 16/17 were at least one order of magnitude higher than the balloon contamination. Hence, we could exclude the balloon envelope as responsible for the contaminated measurements.

However, the balloon envelope could be the source of the contamination observed from 20 hPa onwards in all flights, see

Figure 1b of Brunamonti et al. (2018). The average profiles of water vapour mixing ratio for StratoClim 16/17 showed 6 to 20 ppmv $\chi_{\mathrm{H_2O}}$ between 20 and 10 hPa, which are too high compared to stratospheric water vapour mixing ratios. Note how above the 20-hPa level the dark red area in Figure 15 overlaps the StratoClim campaign averages: $\langle\chi_{\mathrm{CFH}}\rangle_{\mathrm{NT}}$ and $\langle\chi_{\mathrm{CFH}}\rangle_{\mathrm{DK}}$.

The difference in contamination 55 m below the balloon due to different ascent velocities was not significant, less than 10% $\chi_{\mathrm{H_2O}}$. We also investigated the contamination observed 55 m below the balloon, when only half of the balloon surface was

covered in ice. We found the contamination to be approximately half of that observed for the full coverage of the balloon skin. We also investigated balloon contamination in the upper troposphere. We ran a simulation for 200 hPa with background water vapour of 100 ppmv. At this level and temperature of -40 °C, water vapour saturation is 600 ppmv. At the position of the payload, 55 m below the balloon, we could expect an extra 12 ppmv of water vapour mixing ratio. The contamination was not negligible, but it was comparable to the instrumental uncertainty of CFH (10%) and it would also be detected by the capacitive

humidity sensor on the RS41 radio sonde, hence it cannot be uniquely identified.





### 5.4.2 Instrument package

We examined possible contributions from the instrument housing, i.e. the Styrofoam package, to the observed contamination. To this end, we ran simulations for the atmospheric conditions summarized in Table 6 and used an instrument package geometry and mesh including intake and outlet tubes, see Figure 16. The CFH package is rectangular ($12\,\mathrm{cm} \times 17\,\mathrm{cm} \times 31\,\mathrm{cm}$) and not

symmetric, as the intake tube is not located at the centre of the package. It is centred along the shorter dimension (Figure 16b) but asymmetrically positioned along the longer dimension (Figure 16c). The simulation domain extended by $25\,\mathrm{cm}$ above the top of the intake tube, 30 to $35\,\mathrm{cm}$ beyond the CFH package sides, and by $150\,\mathrm{cm}$ below the outlet tube and allowed for a realistic development of the flow around and below the payload.

Figure 16 displays only a reduced part of the domain. The simulation referred to $20\,\mathrm{hPa}$ atmospheric pressure with $7\,\mathrm{m\,s^{-1}}$

ascent velocity, assuming the top surface of the CFH box to be ice-covered. Figure 16 shows the flow velocity field (panel a), the resulting distribution of $H_2O$ mixing ratio from the two major viewing angles (panels b and c), and the mixing ratio for an hypothetical CFH with a shorter intake tube (panel d). The intake tube normally extends by $12\,\mathrm{cm}$ above the box, but in the experimental version in panel (d) only by $6\,\mathrm{cm}$.

For the four stratospheric levels (Table 6), we observed the magnitude of the contamination in the wake of the payload

increase with decreasing pressure and increasing temperature. Recurrent in all simulations was the flow deceleration above and below the package (Figure 16a), creating a recirculation in these areas, and a flow acceleration on the sides. Note that in this simulation, the flow was also simulated inside the intake tube, and it became fully developed inside the tube. We found that the flow velocity at the inlet of the intake tube was about 70% of the balloon ascent velocity (see Figure 16a) which is a better quantification than the estimate of 50% provided by Vömel et al. (2007c).

The recirculation effect above the CFH package was able to pull water vapour from the package surface and increase the water vapour mixing ratio of the air surrounding the intake tube. Figure 16c shows the recirculation to be more intense on the shorter side of the package and $\chi_{H_2O}$ to be higher in this region. However, the intake tube prevented water vapour sublimated from the package to contaminate the sampled air. Contamination started from $\sim 2\,\mathrm{cm}$ below the tube inlet. However, the intake tube caused and enhanced the re-circulation effect above the CFH package. From the simulation with the shorter intake tube

(Figure 16d), we saw the contaminated area starting lower than for the longer intake tube (Figure 16b). Nevertheless, the longer intake tube seemed to remain the better option to prevent water vapour sublimated from the package to contaminate the measurement. As conclusion, the intake tube is effective at preventing contamination from the instrument package. A reduction of its length is not recommended.

## 6  Conclusions and Recommendations

### 6.1  Summary

We investigated contaminated water vapour measurements made by cryogenic frost point hygrometers during the 2016-2017 StratoClim balloon campaigns on the southern slopes of the Himalayas. We analysed extensively three distinct cases, where





COBALD backscatter measurements of aerosol and clouds were available and $H_2O$ contaminations were observed. In these cases, we encountered mixed-phase clouds in the troposphere and by means of observation and modelling we suggested that liquid water was likely present in all of them. By novel interpretation of the GPS data we quantified the balloon pendulum movement. By means of computational fluid dynamic (CFD) simulations we estimated the impact of the pendulum motion

on the collision efficiency of supercooled liquid droplets on the inner wall of the intake tube. We clarified that impingement angles in the intake tube are bigger than tilt angles resulting from the mere displacement of the payload below the balloon, due to horizontal velocity of the payload induced by the pendulum and rotating movements. We also compared the impact of different size droplets: big droplets had higher collision efficiency rates than smaller droplets, with some dependence on the impingement angle. For example, less than 50% of liquid droplets with $r \sim 10\ \mu m$ collided and froze in the intake tube at

impingement angles of around $50°$, while 100% of droplets with $r > 70\ \mu m$ froze already at impingement angles $> 5°$.

We showed that agreement can be established between the contaminated water vapour measurements in the stratosphere and computational fluid dynamics (CFD) simulations for an ice coverage starting at the rim of the intake tube and extending by 5-15 $cm$ into the tube. We showed that the recovery of contaminated water vapour measurements can be explained in terms of smaller ice coverages eventually leading to uncontaminated water vapour observation in the stratosphere once all ice in the

intake tube sublimated. This study provided a clear picture of the evolution of the ice layer inside of the intake tube during the sublimation process. Ice closer to the top of the intake tube sublimated more efficiently, thereby protecting the ice downstream closer to the centre. However, because the collisions of supercooled cloud droplets during in-cloud icing were less efficient closer to the centre of the tube, they generated a thinner ice layer and thus sublimated first. The last ice to sublimate was that around 1 to 5 $cm$ from the top of the intake tube. By comparison with RS41 we also showed that water vapour measurements

by the CFH in the upper troposphere, after passing through mixed-phase clouds could also be contaminated, especially under conditions with $S_{ice} < 0.7$.

The characteristics common to two of the three analysed flights, NT011 and NT029, were the presence of cold mixed-phase clouds, at air temperature lower than -20 °C, fast ascent balloon velocities of 6 to 7.5 $m\,s^{-1}$, and the total sublimation of any ice coverage of the intake tube within the flight time, i.e. before balloon burst. These characteristics contrasted to those

of the third flight, NT007, where a warm mixed-phase cloud was present at air temperatures between 0 and -5 °C, a slow balloon ascent through the entire flight between 3 and 4 $m\,s^{-1}$, and the contamination of the water vapour measurements in the stratosphere persisted until burst. It was known that liquid clouds and warm mixed-phase clouds could irreversibly contaminate water vapour measurements by CFH [Holger Vömel, personal communication, 2016], but our results showed that even cold mixed-phase clouds with very low LWC can affect the operation of the CFH.

We also showed that neither the balloon envelope nor the instrument package were likely to cause the water vapour contamination found in these cases. The intake tube successfully shielded the sampled air against contamination from the instrument package. However, frost on the balloon envelope may have caused the enhanced and contaminated water vapour values observed between the 20-$hPa$ level and balloon burst.



## 6.2 Design and operation recommendations

It is possible to reduce the pendulum oscillation of the payload by flying a two balloon tandem separated by a rigid triangle as described by Kräuchi et al. (2016). However, further investigation is required to confirm that by reducing the oscillation, the risk of contamination is reduced. The payload would fly more often in the wake of the balloon and hence be subject to contamination

by the balloon. Furthermore, the oscillatory movement would not be completely avoided. The smaller oscillations might result in a more uniform layer of ice, with an even thickness from the rim to the the centre of the tube. The sublimation would be faster as well as the recovery of the instrument in the stratosphere. However, variability in LWC is a much larger effect that cannot be controlled, e.g. causing a bigger ice thickness in the intake tube, resulting in longer sublimation times.

As seen in Figure 10, ambient air can enter the tube and remain unperturbed in terms of water vapour contamination for a

few centimetres, even if there is ice inside of the tube. The wider the tubes, higher the mass flow of atmospheric air, the longer the air can remain unperturbed. Mastenbrook (1965, 1968) already recommended similar measures. To reduce even further the effect of the contamination from ice inside the intake tube, we recommend moving the measurement location, i.e. the mirror for the CFH, closer to the start of the intake tube and as far as possible from the intake tube walls. However, we do not recommend shortening or removing the intake tubes because they are effective at protecting the measurement from contamination from the

instrument package.

Heating of the intake tubes could be an option to reduce the contamination caused by ice inside the intake tubes. However, heated tubes could evaporate liquid or ice water present in the air and contaminate the entire measurement of water vapour turning it into a total water measurement and make it more difficult to assess supersaturation in cloud (Kämpfer, 2013). We suggest performing one heating cycle of the intake tubes after the region of mixed-phase clouds, at air temperatures colder than

-38 °C, similar to what is done in the mirror for the CFH with the clearing and freezing cycle (Vömel et al., 2007b) but for the tube. The measurement would be perturbed in the upper troposphere for a few seconds or minutes, but a clean stratospheric water vapour profile might be the reward.

We did many assumptions throughout this study due to the lack of information of the observed clouds. The backscatter measurements from COBALD do not suffice to derive cloud drop sizes and physical states. One instrument that could provide

useful additional information is a hot-wire probe to measure liquid water content (LWC) and total water content (TWC) in mixed-phase clouds. The instrument is mainly used in aircraft and we are not aware of its use for balloon sounding. The principle is simple and detection limits are of the order of 0.003 to 0.005 $\mathrm{g\,m^{-3}}$ (Korolev et al., 2003a). However, the power availability could be a limitation for implementation in balloon sounding. In Serke et al. (2014), a new vibrating wire sonde based of the design of Hill and Woffinden (1980) was used to measure supercooled liquid water content (SLWP) from a balloon

platform with interesting results, but provided no information about the droplet size distribution.





*Code and data availability.* The data of flights NT007, NT011 and NT029 is provided (Jorge, 2020) as well as mesh and case initialization files for the three geometries used in the CFD simulations: intake tubes, balloon envelope ($r = 2.3$ m) and instrument package. To re-create the results of the CFD simulations use the information provided in Tables 2, 6, A1 and B1. The user defined function that implements ice and water saturation in the CFD simulations - developed by Lüönd (2009) and licensed under MIT-License - is also provided.





## Appendix A: Flight NT029

Here we analyse the contamination during flight NT029. The figures shown in this appendix are analogous to several figures shown in the main body of the paper. Figure A1 shows the full profile of flight NT029 on 30 August 2016. As for the flight NT011, there were contaminated water vapour mixing ratios in the stratosphere, and recovery of operation of the CFH still during ascent before balloon burst. COBALD (panel c) observed three clouds: one very thin cloud in the liquid phase regime, at air temperatures higher than 0 °C; a second could, in the mixed-phase regime with very interesting features in the BSR and CI; and a third cloud at air temperature lower than -38 °C, which was in the cirrus or ice cloud regime. We did not consider the liquid cloud to be the source of the contamination, because the cloud finished at $T = 0$ °C and between the end of the liquid cloud and the start of the mixed-phase cloud the payload went through a sub-saturated region. Liquid water on the intake tube wall would evaporate in the sub-saturated region.

Figure A2 shows a detail of the mixed-phase cloud of flight NT029. The mixed-phase cloud existed between the temperatures of -15 °C and -21 °C, when $S_{\mathrm{ice}}$ by the CFH was between 1.1 and 1.05, $S_{\mathrm{liq}}$ by the RS41 was between 0.95 and 0.85, and the CI was above 20. This confined the mixed-phase cloud to the interval between 8.1 and 9.1 km altitude.

Figure A3 shows the cloud modelling results for the mixed-phase cloud shown in Figure A2. From the integration of water vapour in the upper troposphere and stratosphere of flight NT029 (see Table 3), we determined the upper and lower estimates of liquid water content (LWC) of this mixed-phase cloud to be 0.160 g m$^{-3}$ and 0.032 g m$^{-3}$ respectively (see Table 1). For the simulation, we defined the initial distribution with the same ice crystal and liquid droplets sizes as for the NT011 cloud modelling: $r_{\mathrm{ice}} = 10$ μm, $r_{\mathrm{liq},1} = 10$ μm, $r_{\mathrm{liq},2} = 100$ μm, and $r_{\mathrm{liq},3} = 200$ μm. The bigger droplets extended the glaciation time and prolonged the duration of the cloud liquid phase. The initial ice crystal concentration was the same as the expected for ice nucleation particles (INP) at these temperatures: $n_{\mathrm{ice}} = 0.02$ cm$^{-3}$ (DeMott et al., 2010). The upper estimate of LWC started with $n_{\mathrm{liq},1} = 70$ cm$^{-3}$ and $n_{\mathrm{liq},2} = 0.030$ cm$^{-3}$ and the lower estimate of LWC started with $n_{\mathrm{liq},1} = 30$ cm$^{-3}$ and $n_{\mathrm{liq},2} = 0.002$ cm$^{-3}$, both simulations had $n_{\mathrm{liq},3} = 0.001$ cm$^{-3}$. Both upper and lower estimate clouds existed for about $\Delta t \sim 40$ minutes at the $S_{\mathrm{liq}}$ and $S_{\mathrm{ice}}$ conditions observed in the NT029 mixed-phase cloud. The average velocity of the payload in this part of the flight was 6 m s$^{-1}$, which means the payload was in the 1000-m-long cloud for about 3 minutes.

Figure A4 shows the pendulum analysis for the mixed-phase cloud of flight NT029. We observed payload oscillations with up to 40 m amplitude. Figure A5 summarizes the observed velocities ($v_{\perp}$) perpendicular to the intake tube, ascent velocities parallel ($w_{||}$) to the intake tube and impingement angles ($\beta$) experienced during the mixed-phase cloud of flight NT029.

Figure A6 shows the FLUENT simulations results for the collision/ freezing efficiency of hydrometeors inside the intake tube for flight NT029. We only show results for 100 μm radius droplets. The results for 200 μm droplets were very similar to the ones shown in Figure B6. As for flight NT011, all big droplets froze on the intake tube wall. Again, with higher impingement angles, the freezing efficiency was higher at the top of the intake tube. The ice layer thickness for the first 5 cm of the intake tube are shown in Figure A6k for the two estimates of LWC in the mixed-phase cloud.

Figure A7 shows the stratosphere of flight NT029, and the FLUENT simulation results (see also Table A1). The ascent velocity for NT029 was less variable, about $\pm 1$ m s$^{-1}$, than for flight NT011. From the comparison of the simulation results





for $\chi_{H_2O}$ in Figure A7b, we concluded, as for flight NT011, that the simulations with 5 cm ice coverage of the intake tube yielded the best description of the observations. The contamination was more persistent for flight NT029 than for flight NT011, i.e. it lasted longer, but the burst was also at higher altitude. The measurement started recovering at 26 km altitude. At 27.5 km altitude, the simulation for a 1 cm ice coverage of the intake tube matched the observed $\chi_{H_2O}$. At higher altitude levels,

simulations with smaller ice coverages such as thinner layers of 0.15 cm length, and radially asymmetric patches as shown in Figure 10g or smaller, extending only for 0.45 cm instead of 1 cm, matched the observations. Above 30.5 km altitude, we considered the measurement to be recovered.

Regarding contamination in the upper troposphere for flight NT029, we considered two regions where sublimation of ice inside the intake tube could have happen: from 11 km altitude to the start of the cirrus cloud, where $S_{ice} < 1$, and from above

the cirrus cloud at 15.5 km altitude to the CPT, which are shown in Figure A8. We excluded the region directly above the mixed-phase cloud, from 9 to 11 km altitude. The backscatter ratio, as can be seen in Figure A1d and A2b, was very perturbed and $S_{ice} > 1$, so, it was possible that the payload was in cloud. We also did not consider the cirrus cloud region. At $S_{ice} = 1$ there would be no sublimation of the ice inside of the intake tube. In the upper troposphere of flight NT029 (Figure A8), we noticed again the dry bias of RS41 in relation to the CFH. At the three lower altitudes in the upper troposphere (12.2 km, 12.7 km

and 15.6 km, see Table A1) for which we ran simulations, there is good agreement between the $\chi_{H_2O}$ of the simulation with 15 cm ice coverage inside the intake tube and the $\chi_{H_2O}$ measured by the CFH. At 16.5 km altitude, the simulation with 5 cm ice coverage inside the intake tube showed the best agreement with the observations.

The excess integrated water vapour in the stratosphere of NT029 was 15.7 mg. The excess integrated water vapour in the upper troposphere of NT029 was 3.5 mg for the lower sub-saturated region between the mixed-phase cloud and the cirrus

cloud, and 60.5 mg between the cirrus cloud and the tropopause. The total excess integrated water vapour in flight NT029 was 79.7 mg. All these values are in Table 3.

## Appendix B: Flight NT007

Here we analyse the contamination during flight NT007. The figures shown in this appendix are analogous to several figures shown in the main body of the paper. Figure B1 shows the full profile of flight NT007 on 11 August 2016. As was observed in

flights NT011 and NT029, there were contaminated water vapour mixing ratios in the stratosphere, and contrary to the other flights, there was no recovery of the operation of the CFH before balloon burst. COBALD (panel c) observed two clouds: one extending from $T_{air} = 0\,°C$ to $T_{air} = -38\,°C$, the entire mixed-phase cloud regime; and a second one in the cirrus or ice cloud regime, extending all the way to the tropopause. The lengthy cloud in the mixed-phase cloud regime was likely the cause of the contamination in the stratosphere, specifically two regions of the cloud, which were likely able to support liquid droplets at

air temperatures below $0\,°C$. Details of these two regions of the cloud are shown in Figure B2. Panels (a) and (b) refer to the warmer mixed-phase cloud in air temperatures between -4 and -7 $°C$, and between 6.25 and 7 km altitude. Panels (c) and (d) refer to the colder mixed-phase cloud in air temperatures between -21 and -25 $°C$, and between 9.2 and 9.85 km altitude.



We did not consider the part of the cloud between 5.5 km and 6.25 km altitude (see Figure B2b) for the cloud modelling, because we could not be sure if this cloud was liquid, mixed-phase or fully glaciated. $S_{ice}$ and $S_{liq}$ measurements are very close to 1. $S_{liq}$ from RS41 is not precisely 1, which is the expected performance of RS41 in a liquid cloud and fully glaciated clouds are uncommon at these temperatures (Korolev et al., 2003a), however, CI from COBALD is 20. From 6.25 km altitude,

the CFH was not operating properly (see Figure B2b). The deposit on the CFH mirror might have been liquid or a mixture of liquid water and ice, rendering the $S_{ice}$ measurement by the CFH senseless. The $S_{liq,d}$ CFH measurement agreed with $S_{liq}$ by the RS41 to some degree. From 7.5 km altitude, the CFH showed controller oscillations (Vömel et al., 2016) until the freezing cycle re-established normal operation by creating a stable ice layer in the mirror. To continue this analysis we calculated $S_{ice}$ from RS41 (black) in Figure B2b. At 6.25 km altitude, there was supersaturation over ice and sub-saturation over water, these

conditions likely allowed big supercooled liquid droplets to exist and impact the top of the intake tube. Above 7 km altitude, the cloud was sub-saturated in relation to ice, which is consistent with a sublimating glaciated cloud. The presence of a cloud was supported by the COBALD BSR. At no other point within the cloud was $S_{liq}$ equal to 1. However, between 9.2 and 9.85 km altitude, Figure B2c-d, there was a similar scenario to the one explored for the mixed-phase clouds of flights NT011 and NT029, with $S_{ice} = 1.2$ and $S_{liq} = 0.95$.

We modelled the two regions of the cloud likely to support big liquid droplets at air temperatures below 0 °C. The results are presented in Figure B3. Panels (a) and (b) refer to the warm mixed-phase cloud and panels (c) and (d) refer to the cold mixed-phase cloud. The lower and upper estimates of LWC (see Table 1) were defined by the water vapour sublimated in the stratosphere and upper troposphere (see Table 3). However, for flight NT007, both the upper and lower estimates of LWC are lower estimates, because the water vapour measurement in the stratosphere by the CFH did not recover. We have considered

similar droplets and ice crystal sizes for the distributions of both NT007 clouds. With the exception, that we considered bigger size droplets of $r_{liq,3} = 200$ μm to be present in cloud 1, between 6.25 and 7 km altitude, and not in cloud 2, between 9.2 and 9.85 km altitude. With the prescribed initial liquid droplet and ice crystal distributions, liquid droplets existed in cloud 1 at the observed $S_{ice}$ and $S_{liq}$ for about 1 hour, and in cloud 2 for about 12 minutes. In both cases, reasonable time for the payload to travel through them at about 3 to 5 m s$^{-1}$ ascent velocity.

Figure B4 shows the pendulum analysis for the two mixed-phase cloud regions of NT007. Panel (a) refers to the entire flight, panels (b) and (c) refer to the warmer cloud region (cloud 1), and panels (d) and (e) refer to the colder cloud region (cloud 2). The amplitude of the oscillation in these two cloud regions was smaller than the ones observed for the clouds of flights NT011 and NT029. The maximum amplitude of oscillation for the two clouds was about 25 to 30 m, while for the clouds of NT011 and NT029 it was 40 m. The smaller amplitudes were related to the slower ascent velocities of this flight. The ascent velocities

($w_{||}$) were smaller, but so were the perpendicular velocities ($v_{\perp}$) experienced at the intake tube inlet. Together they still caused big impingement angles ($\beta$), as can be seen in Figure B5. Panels (a), (b) and (c) refer to the warmer cloud region (cloud 1), and panels (d), (e) and (f) refer to the colder cloud region (cloud 2).

Figure B6 shows the FLUENT simulation results for the collision/ freezing efficiency of hydrometeors in the intake tube for the mixed-phase cloud 1 of flight NT007. For this simulation we considered the presence of small droplets ($r_{liq} = 10$ μm) inside

the mixed-phase cloud ($S_{liq} \sim 1$). As mentioned in Section 4.2.2, the small droplets were more connected with the flow and





their collision/ freezing efficiency was much smaller than for bigger droplets. At small impingement angles, e.g. 13° shown in Figure B6a, most of the droplets, which froze in the intake tube, froze in the 'rest of the tube' category, most likely below the mirror extrusion. At impingement angles of 50° less than 50% of the small liquid droplets collided with the top of the intake tube, unlike the bigger droplets (e.g. $r_{liq}$ = 100 μm droplets), 100% of which froze on collision with the first 3 cm of the intake

tube. The freezing efficiency of the bigger droplets, $r_{liq}$ = 200 μm, did not differ significantly from the freezing efficiency of 100-μm-radius-droplets. Figure B7 shows the FLUENT simulation results for collision/ freezing efficiency of hydrometeors in the intake tube for the cold cloud region (cloud 2) of flight NT007. The thickest ice layer inside the intake tube after both clouds, the layer between 1 - 2 cm, was only 25 μm thick (see Figure B6f and B7g), if we considered the upper estimate of LWC inside the cloud. This would represent a 0.4 % decrease of air flow through the intake tube.

Figure B8 shows the stratosphere of flight NT007, and the FLUENT simulation results (see also Table B1). The ascent velocity of NT007 showed a well defined oscillation with a spacial frequency of 1 km$^{-1}$ altitude and amplitude of about 1 m s$^{-1}$ ( black line panel (a)). On average in the stratosphere the ascent of flight NT007 was slower than that of flights NT011 and NT029 (3.5 m s$^{-1}$ vs 5.5 m s$^1$). The temperature in the stratosphere for flight NT007 showed a wave-like behaviour around the average temperature profile of the season. From the comparison of the simulation results for $\chi_{H_2O}$ in Figure B8b,

we concluded, as for flight NT011 and NT029, that the simulations with 5 cm ice coverage of the intake tube yielded the best description of the observations up to 24 km altitude. Above 24 km altitude, the 5 cm ice covered intake tube simulation over-estimated the observed $\chi_{H_2O}$ by CFH. This was consistent with the intake tube ice covered surface decreasing, however, the decrease and recovery was not as observed in the two other flights. The 1 cm ice covered intake tube simulation under-estimated the observed $\chi_{H_2O}$ up to balloon burst at 31 km altitude. The ice coverage of the intake tube in this flight was most

likely different than in the other flights. The warm cloud in this flight had possibly a more uniform droplet size distribution, which would translate in a more uniform coverage of the intake tube and the observed, more persistent, contamination of the $H_2O$ measurements. The integrated excess water vapour in the stratosphere for flight NT007 was 65.5 mg.

To evaluate the upper tropospheric contamination during flight NT007, we looked at the interval between the mixed-phase cloud and the cirrus cloud shown in Figure B9. From 12.5 km altitude, above the CFH freezing cycle, the dry bias between

RS41 and CFH was visible. We ran simulations for two altitudes, 13.0 and 14.1 km (see Table B1). The simulation with a 15 cm long ice coverage inside the intake tubes described the observed $\chi_{H_2O}$ by the CFH the best. In this flight there was an unique observation at 13.8 km altitude when $S_{ice}$ approached 1 under clear sky conditions (see Figure B9c) and CFH under-estimated the water vapour measurement in relation to the RS41. Under those conditions, the icy intake tube top had the opposite effect in contaminating the CFH measurement. It depleted the gas phase water vapour, and grew the ice coverage,

reducing the supersaturation which in a clean intake tube case would have been observed. The integrated water vapour for the upper troposphere of flight NT007 was 47.5 mg (see Table 3). In total more 113 mg of water was observed in flight NT007 than what was expected without contamination.





## Appendix C: FFT analysis

We performed a Fast Fourier Transform (FFT) analysis on the latitude and longitude time series of the payload's oscillatory movement. For this analysis we considered the detrended latitude and longitude GPS data for the mixed-phase cloud section of flight NT011 using different time intervals for the smoothing procedure with the moving average: 7 s, 9 s, 11 s, 13 s and 20 s.

The results from this analysis are shown in Figure C1. We concluded that independently of the time interval used, the highest power spectral density was at frequency $\nu \sim 0.06 \text{ s}^{-1}$, which corresponded to an oscillation period $\tau \sim 16.6$ s.

## Appendix D: Impingement angles

The flow caused by the balloon ascent ($w$) can be decomposed in two components according to the tilt angle $\alpha$ of the payload in relation to the balloon ascent direction: one perpendicular to the intake intake tube walls ($w_\perp$) and another parallel to the

intake tube walls ($w_{||}$) respectively. Their magnitudes are given by:

$$w_\perp = w \cos\alpha(t) \tag{D1a}$$

$$w_{||} = w \sin\alpha(t) \tag{D1b}$$

$w_\perp$ is the $v_{\perp,\,\mathbf{tilt}}$ component of the inlet flow perpendicular component ($v_\perp$). The magnitude of other component in the plane perpendicular to the flow tube axis due to the payload rotational movement ($v_{\perp,\,\mathrm{rot}}$) can be calculated as

$$v_{\perp,\,\mathrm{rot}} = \frac{\sqrt{(R(t+1)_y - R(t)_y)^2 + (R(t+1)_x - R(t)_x)^2}}{\Delta t} \tag{D2}$$

where $R(t)$ and $R(t+1)$ are consecutive de-trended trajectory points and $\Delta t = 1$ s.

The perpendicular component $w_\perp$ of the balloon ascent velocity, or $v_{\perp,\,\mathbf{tilt}}$ is projected into the horizontal GPS plane of the oscillation movement as $v_{\perp,\,\mathrm{tilt}_x}$ and $v_{\perp,\,\mathrm{tilt}_y}$. We assume $v_{\perp,\,\mathbf{tilt}}$ to be aligned towards the centre of the oscillation $(0, 0)$ as shown in Figure 6b. This direction is evaluated as $\theta(t)$:

$$\theta(t) = \tan^{-1}\left(\frac{R(t)_y}{R(t)_x}\right) \tag{D3a}$$

$$v_{\perp,\,\mathrm{tilt}_y} = \mathrm{sign}\left(R(t)_y\right) v_{\perp,\,\mathrm{tilt}} \sin\theta(t) \tag{D3b}$$

$$v_{\perp,\,\mathrm{tilt}_x} = \mathrm{sign}\left(R(t)_x\right) v_{\perp,\,\mathrm{tilt}} \cos\theta(t) \tag{D3c}$$

We then calculate the magnitude of the total perpendicular component of inlet flow velocity $v_\perp$ as

$$v_\perp = \sqrt{\left(v_{\perp,\,\mathrm{rot}_x} + v_{\perp,\,\mathrm{tilt}_x}\right)^2 + \left(v_{\perp,\,\mathrm{rot}_y} + v_{\perp,\,\mathrm{tilt}_y}\right)^2} \tag{D4}$$

The angle of $v_\perp$ on the horizontal plane is not relevant. We assume it is evenly distributed and hence the coating of the intake tube will be more or less radially homogeneous. The impingement angle can then be calculated as

$$\beta = \tan^{-1}\left(\frac{v_\perp}{w_{||}}\right) \tag{D5}$$

from the parallel component of the ascent speed $w_{||}$ and perpendicular component $v_\perp$ of the inlet flow to the intake tube walls.



*Author contributions.*  TJ wrote the paper and produced all figures. TJ, SB, PO, SH, BS and SK made the measurements. SB, TJ, FGW, BPL, YP, RD, MN, SF and TP provided technical and scientific support for the measurements. BPL provided support for the mixed phase cloud modelling. TJ, SB, YP, FGW and TP proofread the text.

*Competing interests.*  No competing interests are present

5   *Acknowledgements.*  The research leading to these results received funding from the European Community's Seventh Framework Programme (FP7/2007–2013) under grant agreement no. 603557 and from GAW-CH under the 'Development, Validation and Implementation of a GRUAN-Worthy Plug-and-Play Balloon-Borne Hygrometer' project.

The authors would like to acknowledge the contributions from: ISRO-ATCM project, K. Ravi Kumar and Sunil Sonbawne from the Indian Institute of Tropical Meteorology (IITM), Pune, India and Deepak Singh from the Aryabhatta Research Institute of Observational Sciences

10  (ARIES), Nainital, India; Hannu Jauhiainen from Vaisala Oyj, Vantaa, Finland; Rijan Kayastha from the Kathmandu University (KU), Dhulikhel, Nepal; Jagadishwor Karmacharya from the Department of Hydrology and Meteorology (DHM), Meteorological Forecasting Division, Kathmandu, Nepal and Markus Rex from Alfred Wegener Institute (AWI) for Polar and Marine Research, Potsdam, Germany.



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



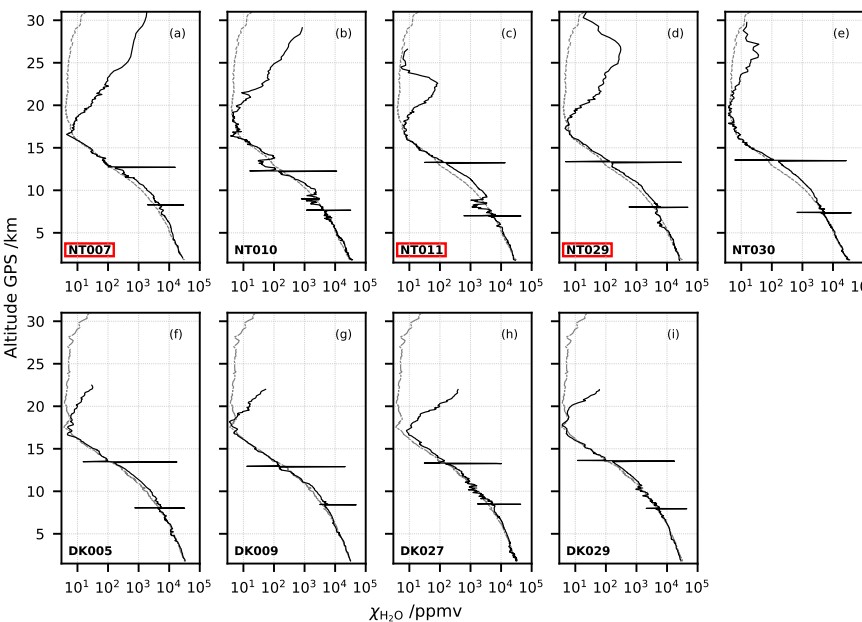

**Figure 1.** Nine water vapour mixing ratio profiles from CFH showing contaminated values in the stratosphere (out of 43 profiles taken during StratoClim 16/17). (a-e) Campaign in Naintal (NT), India, summer 2016. (f-i) Campaign in Dhulikel (DK), Nepal, summer 2017. Black lines: measured individual profiles. Grey lines: respective campaign season average (mean of 22 (NT) or 7 (DK) uncontaminated profiles) as shown in Brunamonti et al. (2018). Two spikes per profile: instrumental freezing and clearing cycles. Highlighted in red: three night time launches with CFH and COBALD, which are further investigated in this study.



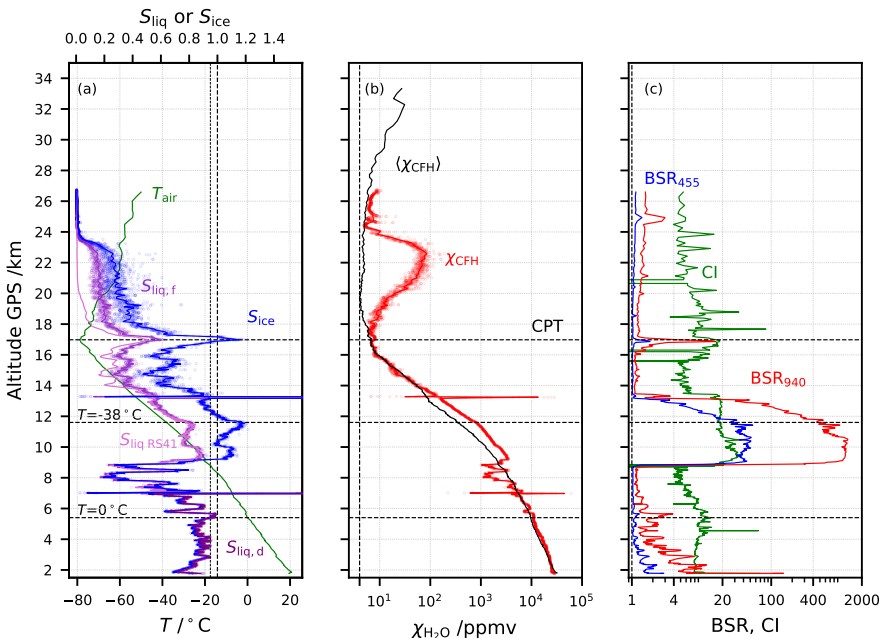

**Figure 2.** Flight NT011 in Nainital, India, on 15 August 2016. Lines: 1 hPa interval averaged values. Dots: 1 s data. (a) Green: air temperature from Vaisala RS41; pink: saturation over water ($S_{\mathrm{liq\,RS41}}$) measured by RS41; blue: ice saturation ($S_{\mathrm{ice}}$) from CFH, dark purple: saturation over water ($S_{\mathrm{liq,d}}$) from CFH considering the deposit on the mirror to be dew; light purple: saturation over water ($S_{\mathrm{liq,f}}$) from CFH considering the deposit on the mirror to be frost. (b) Red: $H_2O$ mixing ratio from CFH in ppmv; black: average $H_2O$ mixing ratio from uncontaminated CFH for the Nainital 2016 summer campaign (Brunamonti et al., 2018); 'CPT' marks the cold point tropopause. (c) Red: 940-nm backscatter ratio from COBALD; blue: same for 455 nm; green: color index (CI) from COBALD.



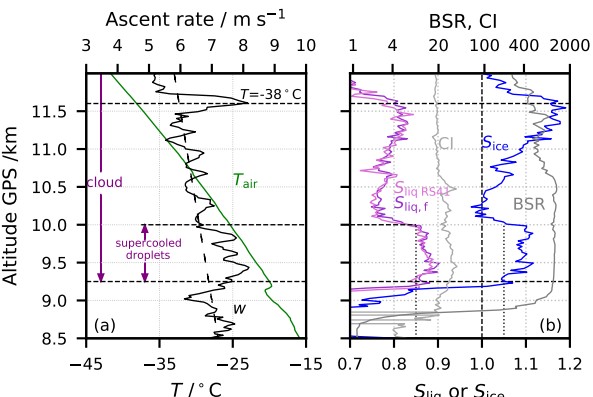

**Figure 3.** Mixed-phase cloud detail of flight NT011. Lines: 1 hPa interval averaged values. (a) Green: air temperature; black: ascent velocity measured by RS41 in $\mathrm{m\,s^{-1}}$. (b) Pink: saturation over water ($S_{\mathrm{liq\ RS41}}$) measured by RS41; light purple: saturation over water ($S_{\mathrm{liq,f}}$) from CFH considering the deposit on the mirror to be frost; blue: ice saturation ($S_{\mathrm{ice}}$) from CFH; dark grey: 940-nm backscatter ratio from COBALD; light grey: color index (CI) from COBALD. Horizontal dashed lines mark supercooled droplet region and $T_{\mathrm{air}}$= -38 °C.



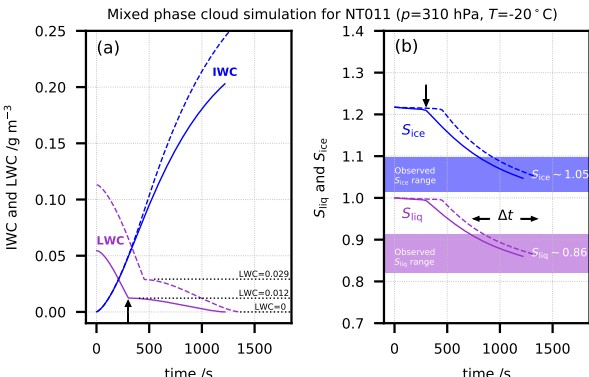

**Figure 4.** Modelling of the Wegener-Bergeron-Findeisen process in mixed-phase cloud demonstrating that flight NT011 likely encountered supercooled liquid droplets. Solid lines: lower estimate of liquid water content (LWC). Dashed lines: upper estimate (see text). Initial size distributions for lower estimate simulation: $n_{\mathrm{ice}} = 0.02$ cm$^{-3}$, $r_{\mathrm{ice}} = 10$ µm; $n_{\mathrm{liq},1} = 10$ cm$^{-3}$, $r_{\mathrm{liq},1} = 10$ µm; $n_{\mathrm{liq},2} = 0.003$ cm$^{-3}$, $r_{\mathrm{liq},2} = 100$ µm. Initial size distributions for upper estimate simulation are identical but with 50 % larger $n_{\mathrm{liq},1}$ and $n_{\mathrm{liq},2}$. (a) Blue lines: ice water content (IWC); purple lines: liquid water content (LWC); (b) Blue lines: ice saturation ratio ($S_{\mathrm{ice}}$); purple lines: liquid water saturation ratio ($S_{\mathrm{liq}}$) for lower and upper estimate. Glaciation times of small droplets $\tau_{g,1} \sim 6$ minutes, of big droplets $\tau_{g,2} \sim 17$ minutes. Shaded saturated ratios: observed ranges from Figure 3. Vertical arrows: time when smaller liquid droplets fully evaporated. The computed time interval with $S_{\mathrm{ice}}$ and $S_{\mathrm{liq}}$ matching flight observations is $\Delta t \sim 7$ minutes.

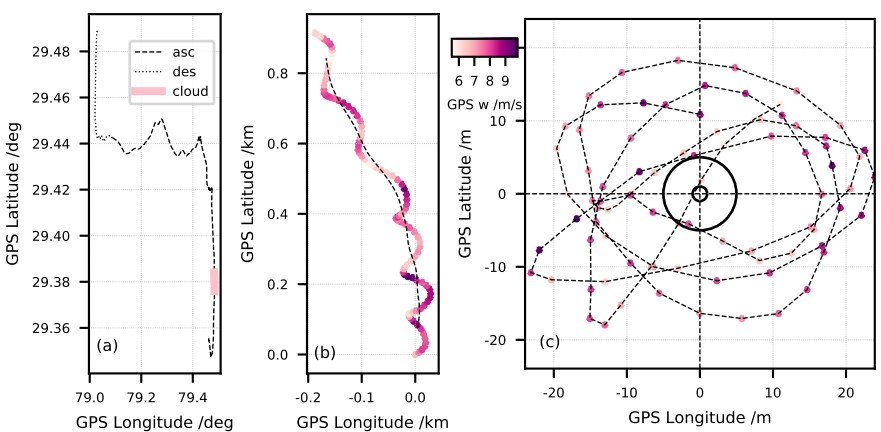

**Figure 5.** Pendulum analysis for the section of flight NT011 traversing the mixed-phase cloud. (a) Payload trajectory for entire flight: ascent (dashed), descent (dotted) and mixed-phase cloud between 9.25 and 10 km altitude (thick pink line). (b) Zoom in on the mixed-phase cloud with 1-second GPS data of payload trajectory (symbols) and derived balloon trajectory (dashed). (c) Detrended payload oscillations; approximate balloon sizes on the ground ($r = 1$ m) and at burst ($r = 5$ m) are shown by two circles. Colour code in (b) and (c): balloon ascent velocity.



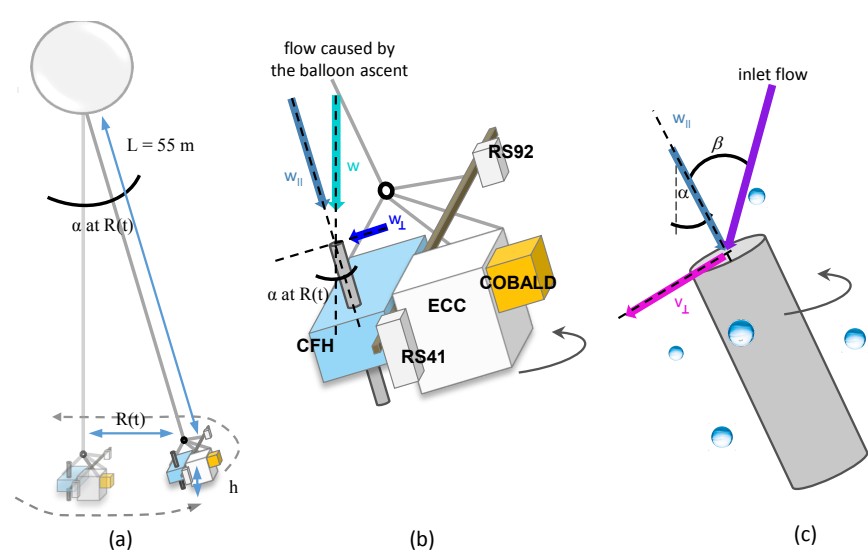

**Figure 6.** (a) Schematic of balloon and payload (not to scale). Payload is connected to the balloon by a 55 m long light-weight nylon cord. Payload oscillates with tilt angles $\alpha$ up to $25°$ during ascent. (b) Schematic of payload with the 2 radiosondes (RS41 and RS92), and the 3 instruments (CFH, ECC Ozone and COBALD). The flow caused by the balloon ascent ($w$) has a component parallel to the intake tube ($w_{||}$) and a component perpendicular to the tube walls ($w_\perp$). (c) From the rotational velocity and the tilt of the tube, the horizontal (perpendicular) velocity $v_\perp$ of the inlet flow can be determined as well as the impingement angle $\beta$.



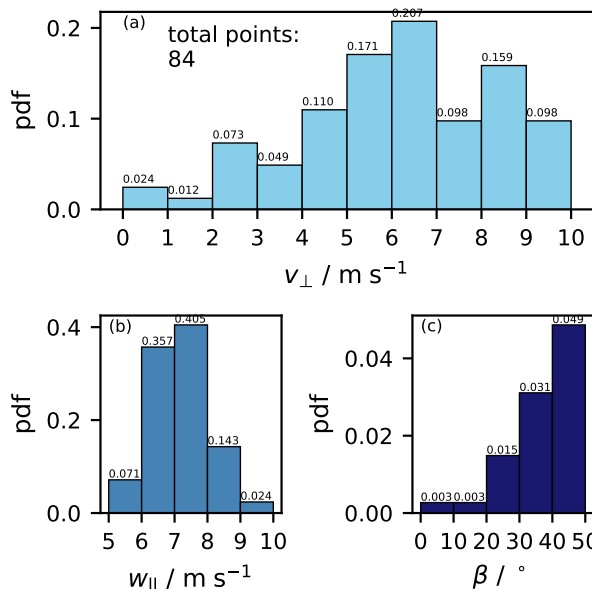

**Figure 7.** Probability density functions (pdf) of impingement parameters at the top end of the CFH intake tube during the passage through the mixed-phase cloud of flight NT011. (a) Velocity $v_\perp$ perpendicular to the tube walls; (b) velocity $w_{||}$ parallel to the axis of the tube; (c) impingement angle ($\beta$).



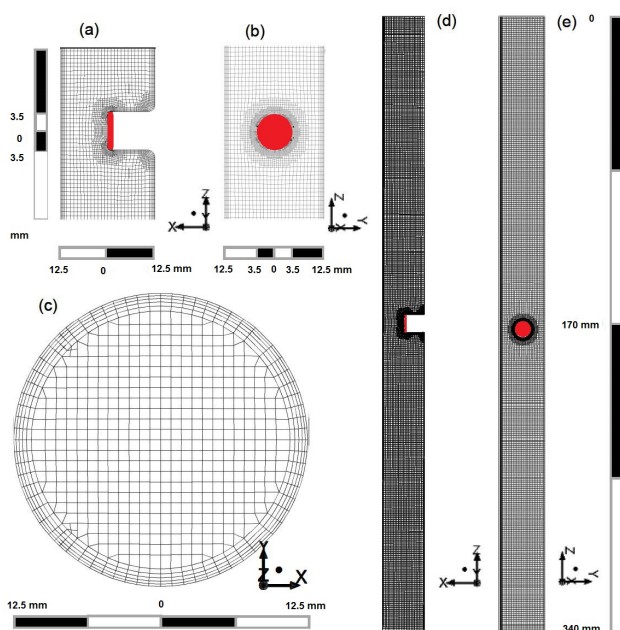

**Figure 8.** Cryogenic frost point hygrometer (CFH) intake tube mesh and geometry. The coordinate origin is located at the top centre of the intake tube. (a,b) Detailed views of mirror extrusion on y = 0 and x = 0 planes. (c) Intake tube cross-section. (d,e) Intake tube on y = 0 and x = 0 planes.



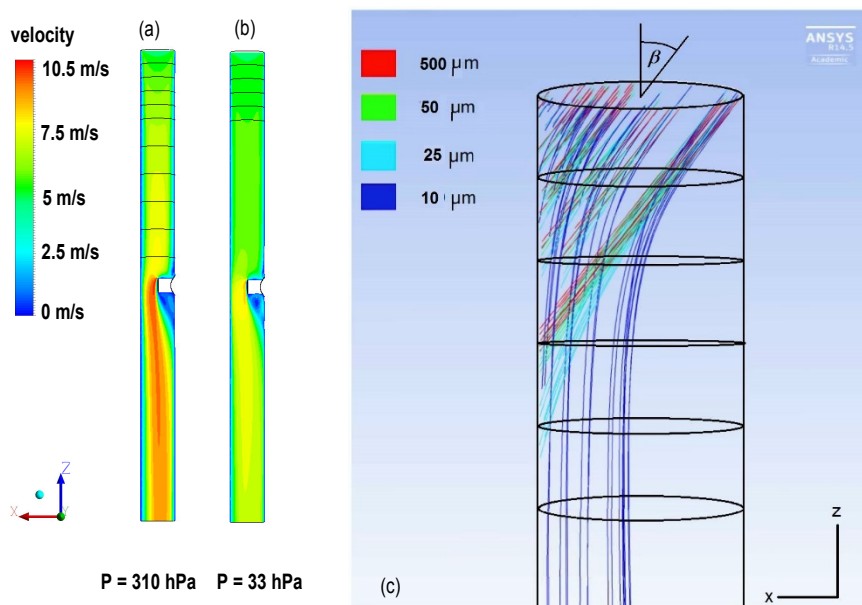

**Figure 9.** (a,b) FLUENT simulation results for air flow velocity (centre cut through mirror extrusion). (a) $p$ = 310 hPa and $T$ = -20 °C and (b) $p$ = 33 hPa and $T$ = -58.7 °C simulations with inlet velocity 5 m s$^{-1}$ normal to the inlet plane. (c) Collision efficiency analysis based on FLUENT simulation results for particle tracks of hydrometeors with radii between 10 μm and 500 μm (colour-coding). The figure shows the top 7 cm of the intake tube. Flow simulations are for the mixed-phase cloud of flight NT011, $p$ = 310 hPa and $T$ = -20 °C. Inlet velocity is $w_{||}$ = 7.5 m s$^{-1}$ parallel to the tube (largely due to the balloon's ascent velocity) and $v_\perp$ = 6 m s$^{-1}$ perpendicular onto the tube wall (largely due to the swinging motion of the payload), which results in an impingement angle $\beta$ of about 39°.

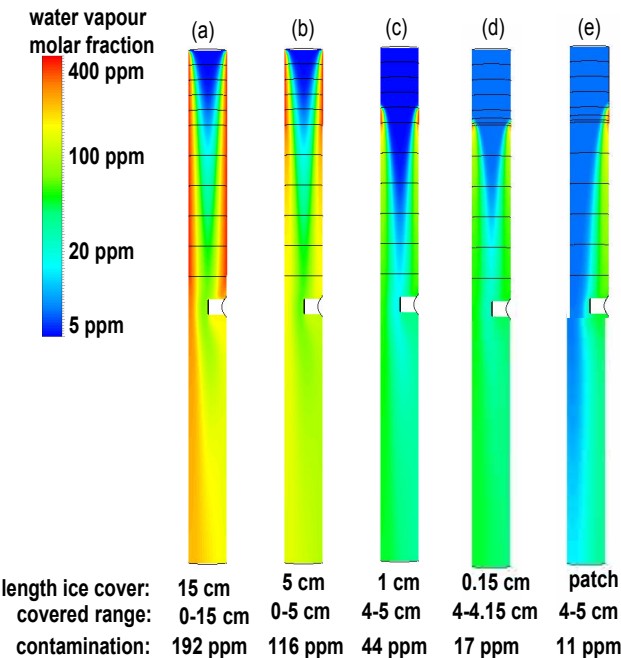

**Figure 10.** FLUENT simulation results for water vapour sublimation in the stratosphere with color coded $H_2O$ mixing ratios for $p = 33$ hPa and $T = $ -58.7 °C with different ice coverage of the intake tube: (a) 15 cm with $\langle \chi_{H_2O} \rangle_{Vol} = 192$ ppmv averaged over the tube volume, (b) 5 cm with $\langle \chi_{H_2O} \rangle_{Vol} = 116$ ppmv, (c) 1 cm with $\langle \chi_{H_2O} \rangle_{Vol} = 44$ ppmv, (d) 0.15 cm with $\langle \chi_{H_2O} \rangle_{Vol} = 17$ ppmv and (e) rotationally asymmetric patch of 1/8 intake tube circumference and 1 cm length with $\langle \chi_{H_2O} \rangle_{Vol} = 11$ ppmv. The triangular brackets indicate mixing ratios that have been averaged over the full tube volume.



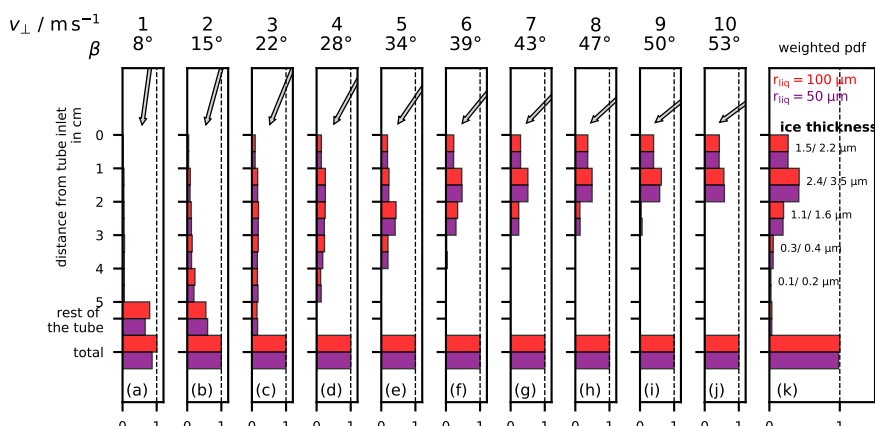

**Figure 11.** Collision/freezing efficiency of hydrometeors in the intake tube for the flight NT011 mixed-phase cloud with average vertical inlet velocity $w_{||}$ = 7.5 m s$^{-1}$. $r_{liq}$ = 100 μm (red), $r_{liq}$ = 50 μm (purple). (a-j) Freezing efficiency for various velocities ($v_\perp$) perpendicular to the tube walls and impingement angles $\beta$: (a) 1 m s$^{-1}$, 8°; (b) 2 m s$^{-1}$, 15°; (c) 3 m s$^{-1}$, 22°; (d) 4 m s$^{-1}$, 28°; (e) 5 m s$^{-1}$, 34°; (f) 6 m s$^{-1}$, 39°; (g) 7 m s$^{-1}$, 43°; (h) 8 m s$^{-1}$, 47°; (i) 9 m s$^{-1}$, 50°; (j) 10 m s$^{-1}$, 53°. The 'rest of the tube' takes account of all collisions occurring deeper than 5 cm inside the tube, including the mirror holder. (k) Sum of the efficiencies from panels (a-j) weighted by the pdf of $v_\perp$ in Figure 7a; behind each bar the thickness of the resulting ice layer is noted assuming a homogeneous ice cover inside the intake tube for the lower (left number) and upper (right number) LWC estimate for the cloud in NT011.



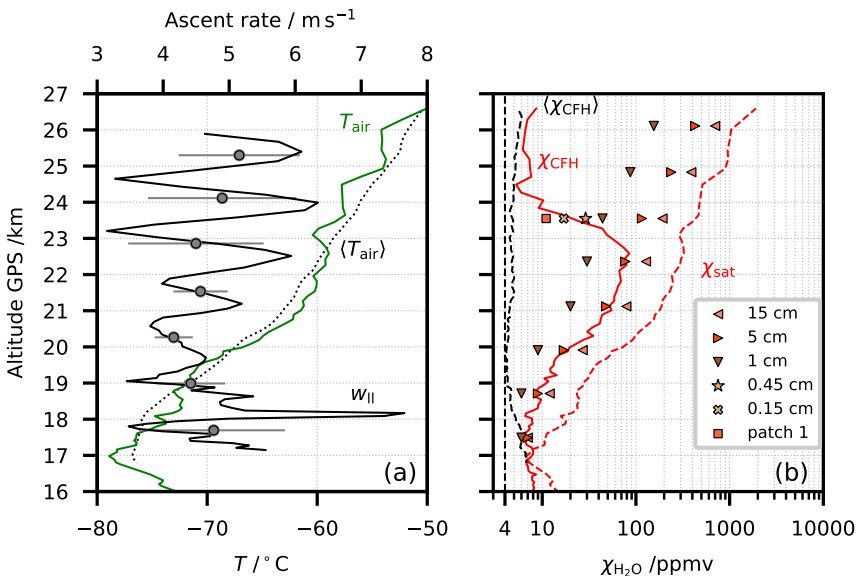

**Figure 12.** Flight NT011 and FLUENT simulation results for sublimation of ice in the intake tube in the stratosphere. (a) Green line: measured air temperature $T_{air}$; dotted black line: average air temperature for the 2016 Nainital summer campaign; solid black line: ascent velocity; circles: 1 km interval averaged ascent velocity; horizontal grey lines: standard deviation. (b) Solid red line: $H_2O$ mixing ratio measured by CFH during NT011 ($\chi_{CFH}$); dashed black line: average $H_2O$ mixing ratio for the uncontaminated soundings during the 2016 Nainital summer campaign($\langle\chi_{CFH}\rangle$); dashed red line: saturation $H_2O$ mixing ratio ($\chi_{sat}$); other symbols: FLUENT simulation results for the tube average mixing ratios $\langle\chi_{H_2O}\rangle_{Vol}$ in tubes with different ice coating depths $d$ coating the full circumference: ◄ $d = 15$ cm; ► $d = 5$ cm; ▼ $d = 1$ cm; ★ $d = 0.45$ cm; x $d = 0.15$ cm; coating only 1/8 intake tube circumference (patch): ■ $d = 1$ cm.



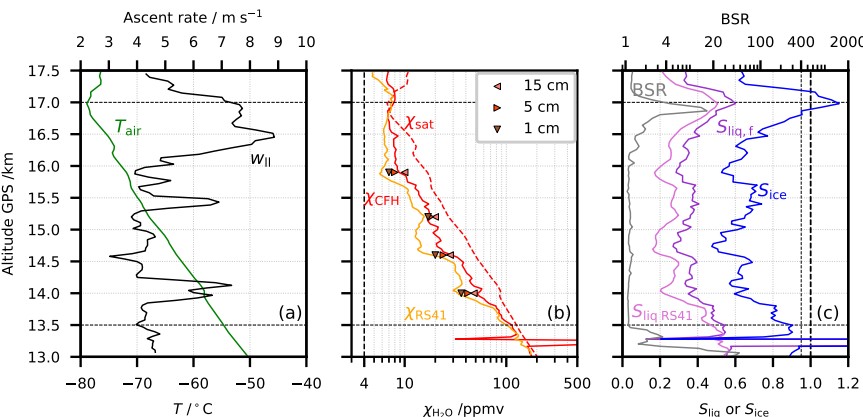

**Figure 13.** Flight NT011 and FLUENT simulation results for sublimation in the upper troposphere. (a) Green: air temperature; black: ascent velocity. (b) Red: $H_2O$ mixing ratio by the CFH; orange: $H_2O$ mixing ratio RS41; dashed red: saturation $H_2O$ mixing ratio for the air temperature; symbols: FLUENT simulation results for the tube average mixing ratios $\langle\chi_{H_2O}\rangle_{Vol}$ in tubes with different ice coating depths $d$ (full circumference): ◄ $d = 15$ cm; ► $d = 5$ cm; ▼ $d = 1$ cm; (c) Pink: saturation over water ($S_{liq\ RS41}$) by RS41; violet: saturation over water ($S_{liq\ f}$) from CFH considering the deposit on the mirror to be frost; blue: ice saturation ($S_{ice}$) from CFH; grey: 940-nm backscatter ratio from COBALD. Horizontal dashed lines limit the integration interval used for estimating the sublimated water in the upper troposphere.



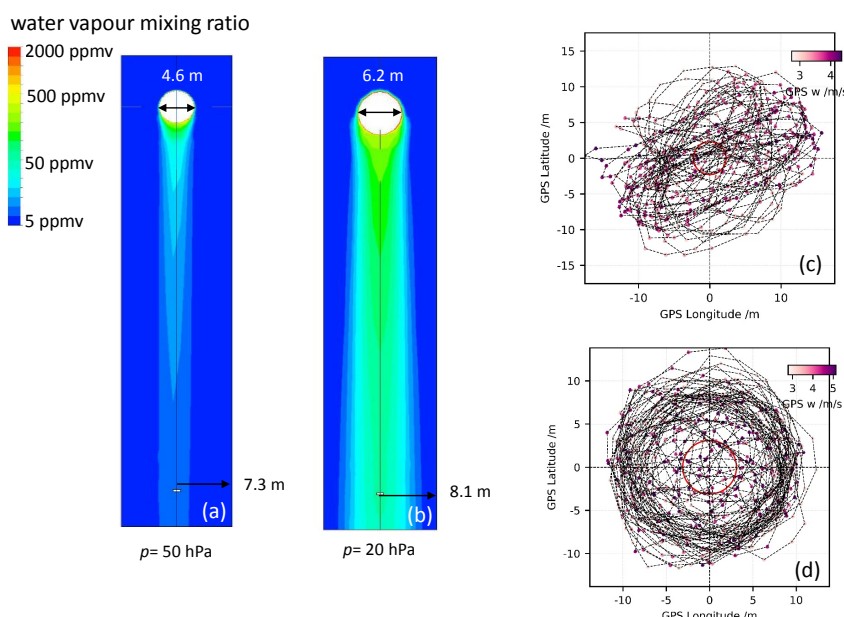

**Figure 14.** Contamination below a balloon: Central cuts through a completely icy balloon and its wake showing the $H_2O$ mixing ratio (a) at 50 hPa and (b) at 20 hPa; oscillations of the payload 55 m below the balloon measured by GPS during NT007 (a) from 20 to 22 km altitude (50-hPa-level) and (d) from 28 to 31 km altitude (20-hPa-level). Red circle: balloon cross-sections at respective pressure level.



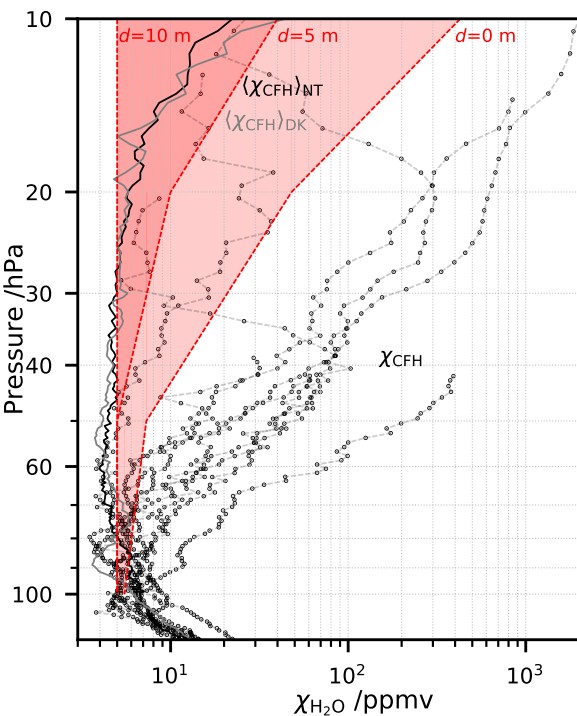

**Figure 15.** Comparison of expected contamination from the balloon envelope with contaminated $\chi_{H_2O}$ observations by the CFH in Strato-Clim 16/17. Black dots: all contaminated profiles of StratoClim 16/17 (see Figure 1), black line: Nainital 2016 season average, grey line: Dhulikhel 2017 season average, light red region: expected contamination when the payload travels directly below the balloon or within a 5 m radius 55 m below the balloon, dark red region: expected balloon contamination if the payload stays outside the 5 m radius circular region 55 m directly below the balloon.



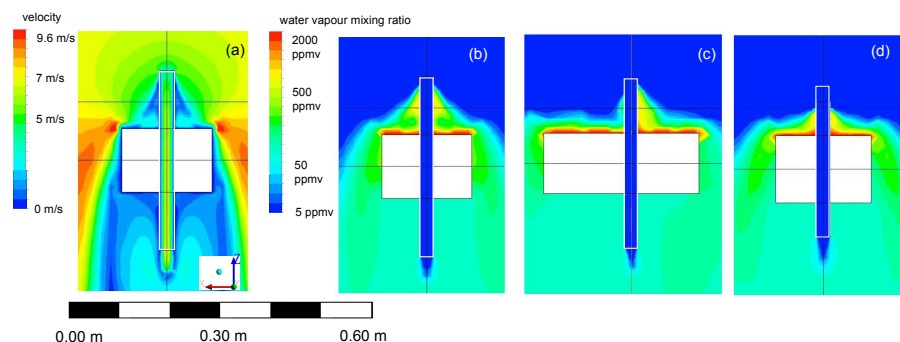

**Figure 16.** Contamination caused by an icy CFH instrument housing (Styrofoam box) at 20 hPa assuming balloon ascent velocity of $7\,\mathrm{m\,s^{-1}}$. (a) flow velocity around the CFH housing, (b-c) $H_2O$ mixing ratio for two side views of the CFH; (d) $H_2O$ mixing ratio contour for an hypothetical CFH with intake tubes extending $6\,\mathrm{cm}$ from the package instead of $11\,\mathrm{cm}$ as for CFH.



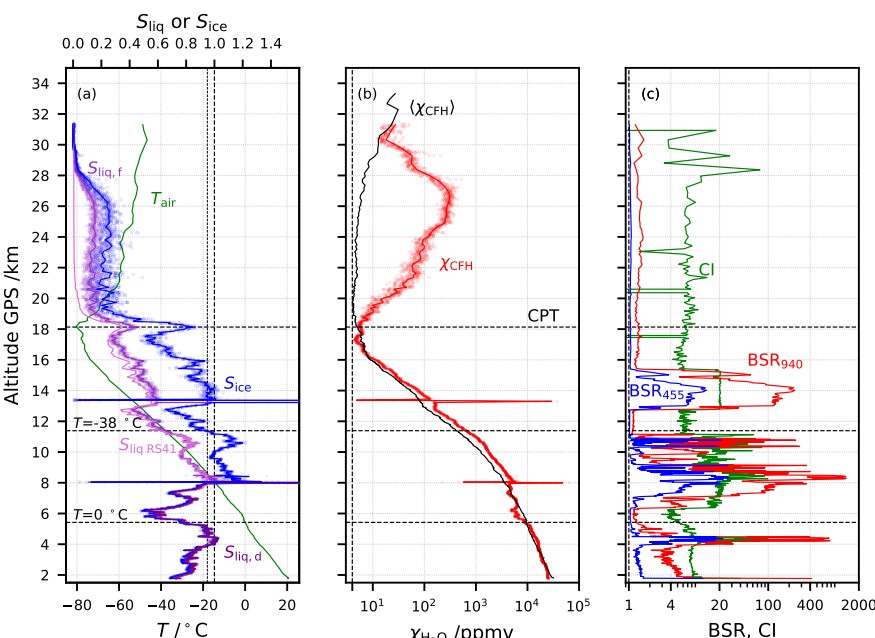

**Figure A1.** Flight NT029 in Nainital, India, on 30 August 2016. Lines: 1 hPa interval averaged values. Dots: 1 s data. (a) Green: air temperature measurement from Vaisala RS41; pink: relative humidity ($S_{\mathrm{liq\,RS41}}$) by RS41; blue: ice saturation ($S_{\mathrm{ice}}$) from CFH; dark purple: saturation over water ($S_{\mathrm{liq,d}}$) from CFH considering the deposit on the mirror to be dew; light purple: saturation over water ($S_{\mathrm{liq,f}}$) from CFH considering the deposit on the mirror to be frost. (b) Red: $H_2O$ mixing ratio from CFH in ppmv; black: average $H_2O$ mixing ratio from uncontaminated CFH for the Nainital 2016 summer campaign (Brunamonti et al., 2018); 'CPT' marks the cold point tropopause. (c) Red: 940-nm backscatter ratio from COBALD; blue: same for 455 nm; green: color index (CI) from COBALD.





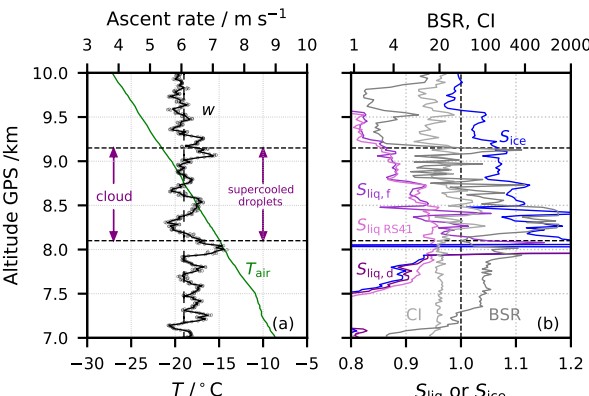

**Figure A2.** Mixed-phase cloud detail of flight NT029. Lines: 1 hPa interval averaged values. (a) Green: air temperature; black: ascent velocity measured by RS41 in $\mathrm{m\,s^{-1}}$. (b) Pink: saturation over water ($S_{\mathrm{liq\,RS41}}$) measured by RS41; dark purple: saturation over water ($S_{\mathrm{liq,d}}$) from CFH considering the deposit on the mirror to be dew; light purple: saturation over water ($S_{\mathrm{liq,f}}$) from CFH considering the deposit on the mirror to be frost; blue: ice saturation ($S_{\mathrm{ice}}$) from CFH; dark grey: 940-nm backscatter ratio from COBALD; light grey: color index (CI) from COBALD. Horizontal dashed lines mark supercooled droplet region.



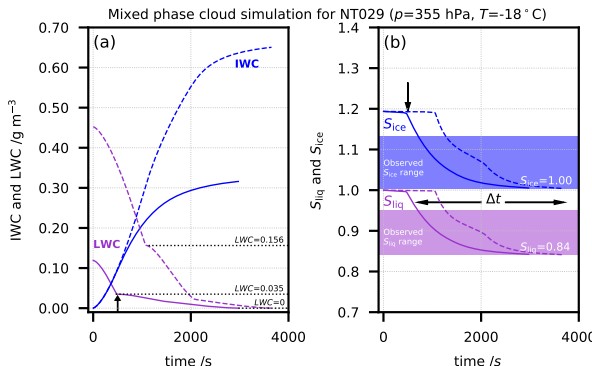

**Figure A3.** Modelling of the Wegener-Bergeron-Findeisen process in mixed-phase cloud demonstrating that flight NT029 likely encountered supercooled liquid droplets. Solid lines: lower estimate of liquid water content (LWC). Dashed lines: upper estimate (see text). Initial size distributions for lower estimate simulation: $n_{ice} = 0.02$ cm$^{-3}$, $r_{ice} = 10$ µm; $n_{liq,1} = 20$ cm$^{-3}$, $r_{liq,1} = 10$ µm; $n_{liq,2} = 0.002$ cm$^{-3}$, $r_{liq,2} = 100$ µm; $n_{liq,3} = 0.001$ cm$^{-3}$, $r_{liq,3} = 200$ µm. Initial size distributions for upper estimate simulation are identical but with 3.5× larger $n_{liq,1}$ and 15× larger $n_{liq,2}$. (a) Blue: ice water content (IWC); purple: liquid water content (LWC); vertical arrows: time when smaller liquid droplets fully evaporated. (b) Blue: ice saturation ratio ($S_{ice}$); purple: liquid water saturation ratio ($S_{liq}$) for lower and upper estimates. Glaciation times of small droplets $\tau_{g,1} \sim 8$ - 18 minutes, of big droplets $\tau_{g,2-3} \sim 45$ - 50 minutes. Shaded saturation ratios: observed ranges from Figure A2. The computed time interval with $S_{ice}$ and $S_{liq}$ matching flight observations is $\Delta t \sim 30$ - 40 minutes.

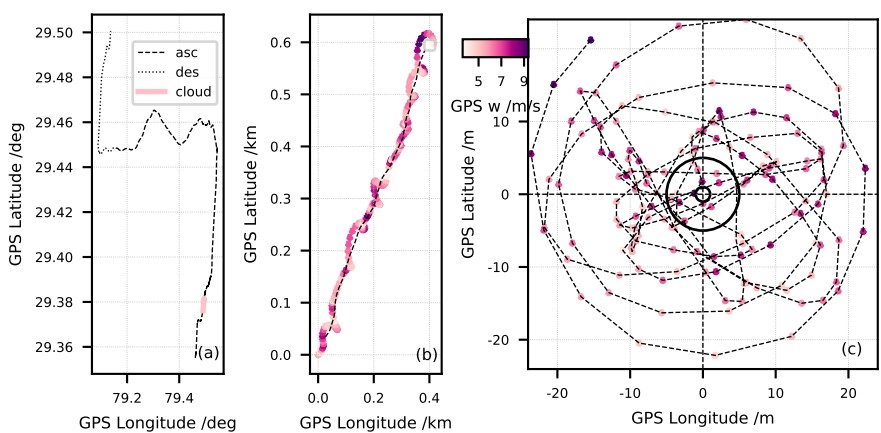

**Figure A4.** Pendulum analysis for mixed-phase cloud of flight NT029. (a) Payload trajectory: ascent (dashed), descent (dotted) and mixed-phase cloud between 8.1 and 9.15 km altitude (thick pink line). (b) Zoom in on the mixed-phase cloud with 1-second GPS data of payload trajectory (symbols) and balloon trajectory (dashed). (c) Detrended payload oscillations; approximate balloon sizes on the ground ($r = 1$ m) and at burst ($r = 5$ m) are shown by two circles. Colour code in (b) and (c): balloon ascent velocity.

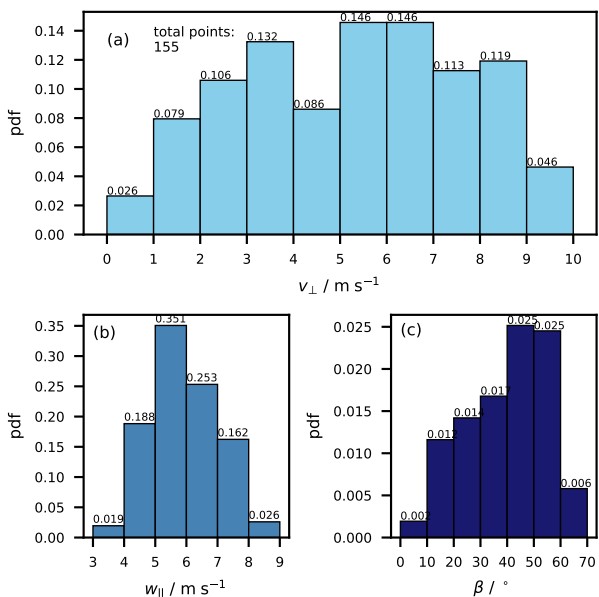

**Figure A5.** Probability density functions (pdf) of impingement parameters at the inlet plane of the CFH intake tube during the passage through the mixed-phase cloud of flight NT029. (a) Velocity $v_\perp$ perpendicular to the tube walls; (b) velocity $w_{||}$ parallel to the axis of the tube; (c) impingement angle ($\beta$).



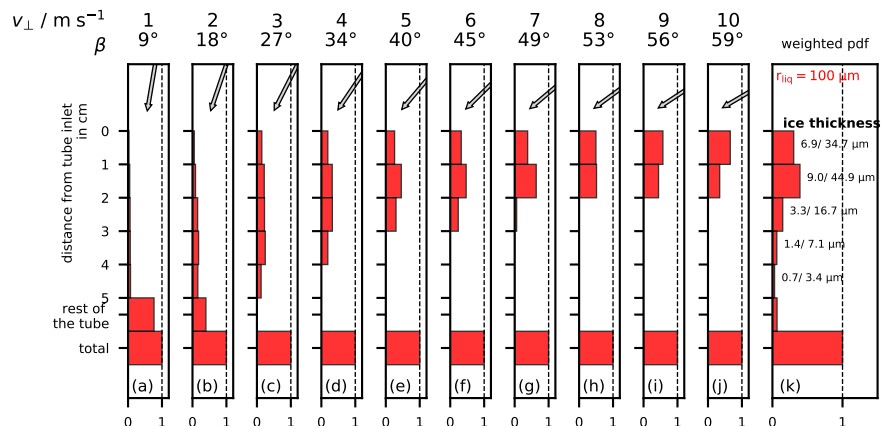

**Figure A6.** Collision/freezing efficiency of hydrometeors in the intake tube for the flight NT029 mixed-phase cloud with average vertical inlet velocity $w_{||}$ = 6.0 m s$^{-1}$. $r_{liq}$ = 100 µm (red). (a-j) Freezing efficiency for various velocities ($v_\perp$) perpendicular to the tube walls: (a) 1 m s$^{-1}$, 9°; (b) 2 m s$^{-1}$, 18°; (c) 3 m s$^{-1}$, 27°; (d) 4 m s$^{-1}$, 34°; (e) 5 m s$^{-1}$, 40°; (f) 6 m s$^{-1}$, 45°; (g) 7 m s$^{-1}$, 49°; (h) 8 m s$^{-1}$, 53°; (i) 9 m s$^{-1}$, 56°; (j) 10 m s$^{-1}$, 59°. The 'rest of the tube' takes account of all collisions occurring deeper than 5 cm inside the tube, including the mirror holder. (k) Weighted sum of the efficiencies in panels (a-j) by the horizontal velocity pdf of Figure A5a, in front of each bar we write the thickness of the subsequent ice layer considering radially homogeneous cover of the intake tube and the lower (left) and upper (right) LWC estimate for the cloud.



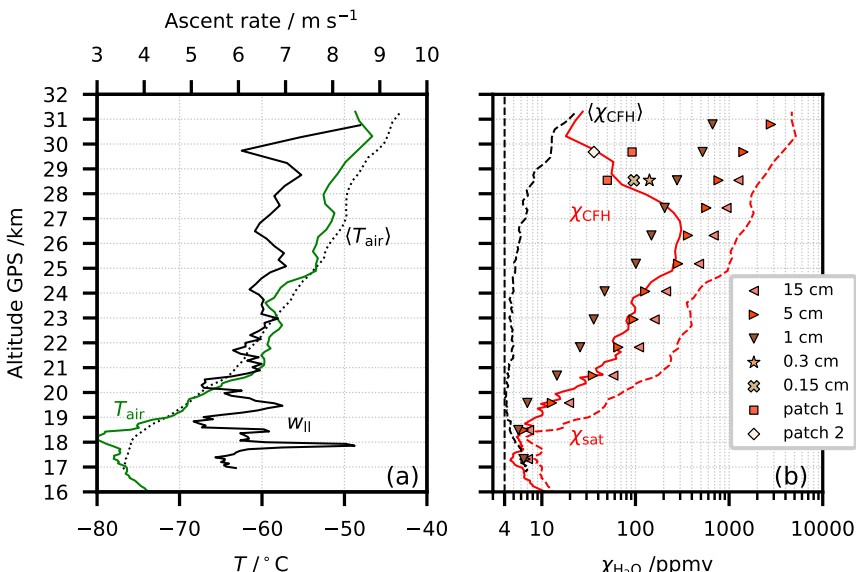

**Figure A7.** Flight NT029 and FLUENT simulation results for sublimation of ice in the intake tube in the stratosphere. (a) Green line: measured air temperature $T_{air}$; dotted black line: average air temperature for the 2016 Nainital summer campaign; solid black line: ascent velocity. (b) Solid red line: $H_2O$ mixing ratio measured by CFH during NT011 ($\chi_{CFH}$); dashed black line: average $H_2O$ mixing ratio for the uncontaminated soundings during the 2016 Nainital summer campaign($\langle\chi_{CFH}\rangle$); dashed red line: saturation $H_2O$ mixing ratio ($\chi_{sat}$); other symbols: FLUENT simulation results for the tube average mixing ratios $\langle\chi_{H_2O}\rangle_{Vol}$ in tubes with different ice coating depths $d$ coating the full circumference: ◀ $d = 15$ cm; ▶ $d = 5$ cm; ▼ $d = 1$ cm; ★ $d = 0.3$ cm, x $d = 0.15$ cm; (1/8 intake tube circumference): ■ $d$ =1 cm, ◇ $d = 0.45$ cm.



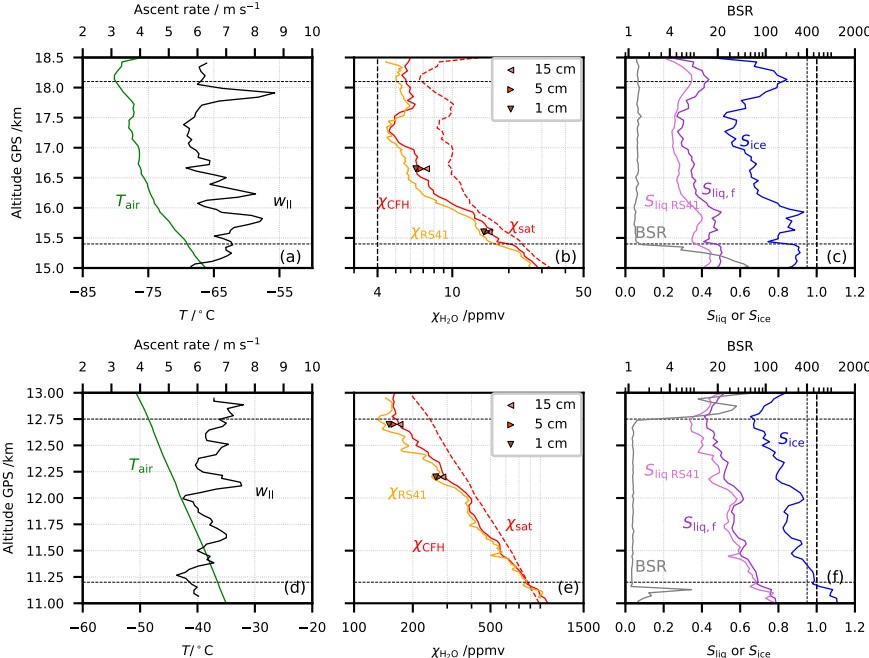

**Figure A8.** Flight NT029 and FLUENT simulation results for sublimation in the upper troposphere.: (a-c) between cirrus cloud and CPT; (d-f) between mixed-phase cloud and cirrus cloud. (a and d) Green: air temperature; black: ascent velocity. (b and e) Red: $H_2O$ mixing ratio by the CFH; orange: $H_2O$ mixing ratio RS41; dashed red: saturation $H_2O$ mixing ratio for the air temperature; symbols: FLUENT simulation results for the tube average mixing ratios $\langle \chi_{H_2O} \rangle_{Vol}$ in tubes with different ice coating depths $d$ (full circumference): ◄ $d$ = 15 cm; ► $d$ = 5 cm; ▼ $d$ = 1 cm;(c and f) Pink: saturation over water ($S_{liq\,RS41}$) by RS41; violet: saturation over water ($S_{liq\,f}$) from CFH considering the deposit on the mirror to be frost; blue: ice saturation ($S_{ice}$) from CFH; grey: 940-nm backscatter ratio from COBALD. Horizontal dashed lines limit the integration interval used for estimating the sublimated water in the upper troposphere.



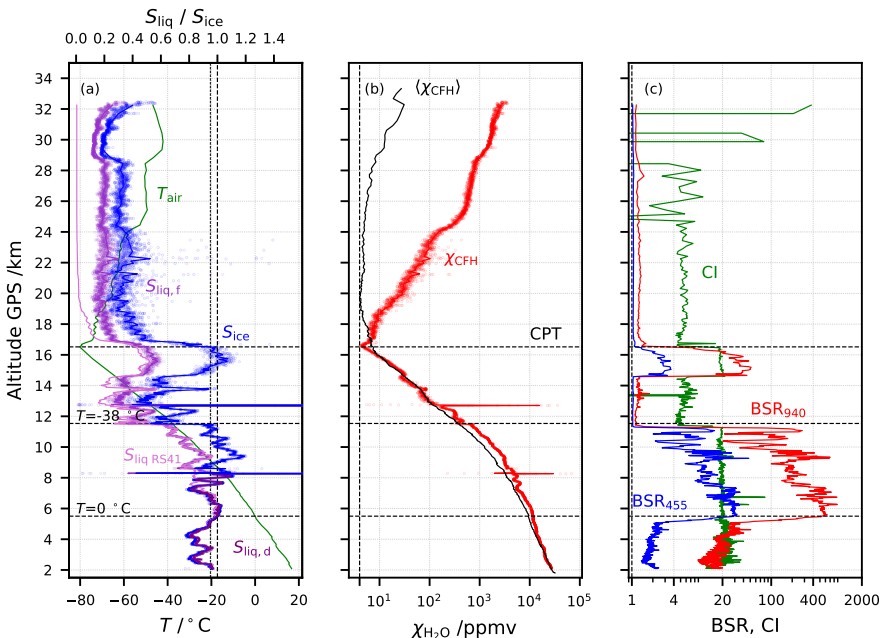

**Figure B1.** Flight NT007 in Nainital, India, on 11 August 2016. Lines: 1 hPa interval averaged values. Dots: 1 s data. (a) Green: air temperature measurement from Vaisala RS41; pink: saturation over water ($S_{\text{liq RS41}}$) measured by RS41; blue: ice saturation ($S_{\text{ice}}$) from CFH; dark purple: saturation over water ($S_{\text{liq,d}}$) from CFH considering the deposit on the mirror to be dew; light purple: saturation over water ($S_{\text{liq,f}}$) from CFH considering the deposit on the mirror to be frost. (b) Red: $H_2O$ mixing ratio from CFH in ppmv; black: average $H_2O$ mixing ratio from uncontaminated CFH for the Nainital 2016 summer campaign (Brunamonti et al., 2018); 'CPT' marks the cold point tropopause. (c) Red: 940-nm backscatter ratio from COBALD; blue: same for 455 nm; green: color index (CI) from COBALD.

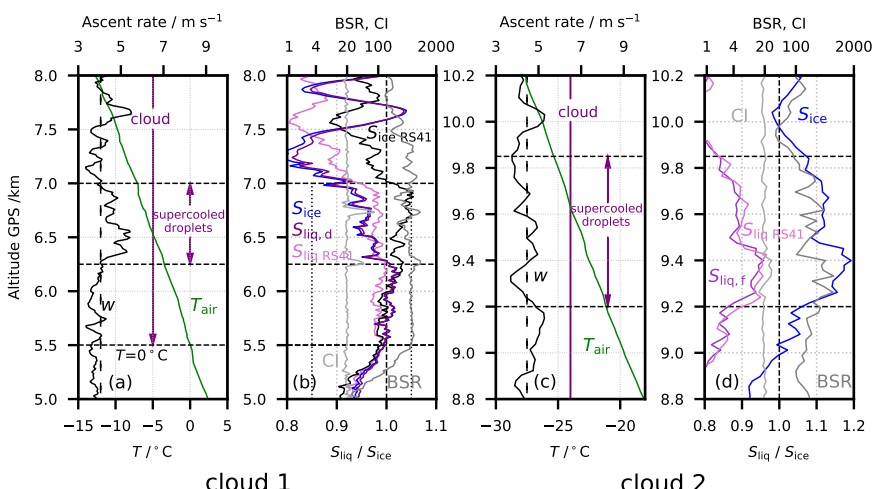

**Figure B2.** Mixed-phase cloud details of flight NT007. Lines: 1 hPa interval averaged values. (a-b) Cloud 1; (c-d) cloud 2. (a and c) Green: air temperature; black: ascent velocity by RS41 in $\mathrm{m\,s^{-1}}$. (b and d) Pink: saturation over water ($S_{\mathrm{liq\,RS41}}$) by RS41; dark purple: saturation over water ($S_{\mathrm{liq,d}}$) from CFH considering the deposit on the mirror to be dew; light purple: saturation over water ($S_{\mathrm{liq,f}}$) from CFH considering the deposit on the mirror to be frost; blue: ice saturation ($S_{\mathrm{ice}}$) from CFH; black: ice saturation ($S_{\mathrm{ice\,RS41}}$) from RS41; dark grey: 940-nm backscatter ratio from COBALD; light grey: color index (CI) from COBALD. Horizontal dashed lines mark supercooled droplet region and $T_{\mathrm{air}} = 0\,^{\circ}\mathrm{C}$.

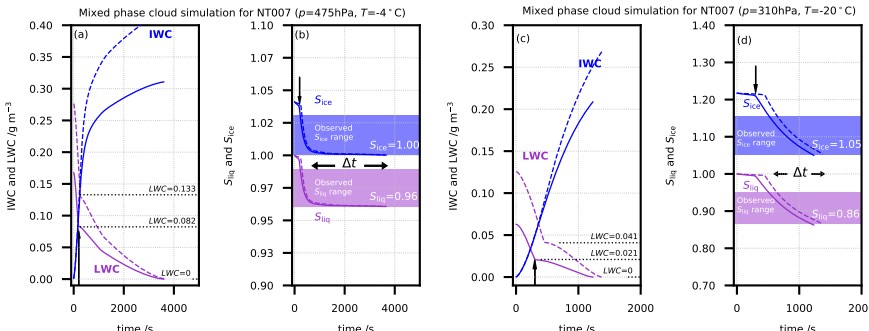

**Figure B3.** Modelling of the Wegener-Bergeron-Findeisen process in mixed-phase cloud demonstrating that flight NT007 likely encountered supercooled liquid droplets in two occasions: (a-b) refer to cloud 1 between 6.25 and 7 km altitude and (c-d) refer to cloud 2 between 9.2 and 9.85 km altitude. Solid lines: lower estimate of liquid water content (LWC). Dashed lines: upper estimate (see text). **(a-b)** Initial size distributions for lower estimate simulation: $n_{\text{ice}} = 0.02 \text{ cm}^{-3}$, $r_{\text{ice}} = 10 \text{ μm}$; $n_{\text{liq},1} = 20 \text{ cm}^{-3}$, $r_{\text{liq},1} = 10 \text{ μm}$; $n_{\text{liq},2} = 0.004 \text{ cm}^{-3}$, $r_{\text{liq},2} = 100 \text{ μm}$; $n_{\text{liq},3} = 0.002 \text{ cm}^{-3}$, $r_{\text{liq},3} = 200 \text{ μm}$. Initial size distributions for upper estimate simulation are identical but with 75% larger $n_{\text{liq},1}$ and $n_{\text{liq},2-3}$. (a) Blue: ice water content (IWC); purple: liquid water content (LWC); vertical arrows: time when smaller liquid droplets fully evaporated. (b) Blue: ice saturation ratio ($S_{\text{ice}}$); purple: liquid water saturation ratio ($S_{\text{liq}}$) for lower and upper estimates. Glaciation times of small droplets $\tau_{g,1} \sim 4$ minutes, of big droplets $\tau_{g,2-3} \sim 60$ minutes. Shaded saturated ratios: observed ranges from Figure B2b. The computed time interval with $S_{\text{ice}}$ and $S_{\text{liq}}$ matching flight observations is $\Delta t \sim 60$ minutes. **(c-d)** Initial size distributions for lower estimate simulation: $n_{\text{ice}} = 0.02 \text{ cm}^{-3}$, $r_{\text{ice}} = 10 \text{ μm}$; $n_{\text{liq},1} = 10 \text{ cm}^{-3}$, $r_{\text{liq},1} = 10 \text{ μm}$; $n_{\text{liq},2} = 0.005 \text{ cm}^{-3}$, $r_{\text{liq},2} = 100 \text{ μm}$. Initial size distributions for upper estimate simulation are identical but with $2\times$ larger $n_{\text{liq},2}$. (c) Blue: ice water content (IWC); purple: liquid water content (LWC); vertical arrows: time when smaller liquid droplets fully evaporated. (d) Blue: ice saturation ratio ($S_{\text{ice}}$); purple: liquid water saturation ratio ($S_{\text{liq}}$) for lower and upper estimates. Glaciation times of small droplets $\tau_{g,1} \sim 8$ minutes, of big droplets $\tau_{g,2} \sim 18$ minutes. Shaded saturation ratios: observed ranges from Figure B2d. The computed time interval with $S_{\text{ice}}$ and $S_{\text{liq}}$ matching flight observations is $\Delta t \sim 12$ minutes.



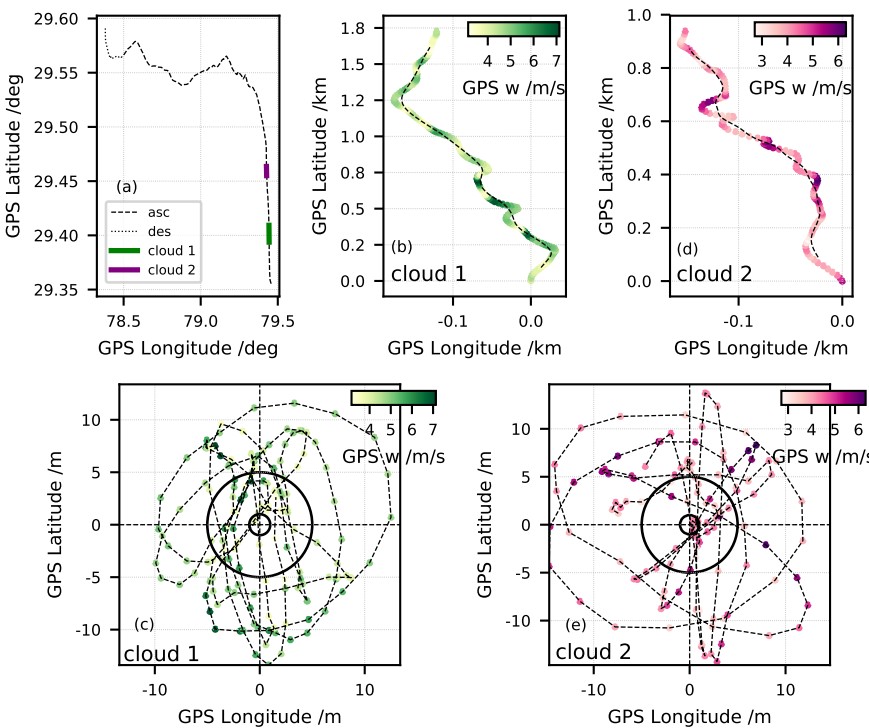

**Figure B4.** Pendulum analysis for mixed-phase clouds of flight NT007. (a) Payload trajectory: ascent (dashed), descent (dotted), mixed-phase cloud 1 between 6.25 and 7 km altitude (thick green line) and mixed-phase cloud 2 between 9.2 and 9.85 km altitude (thick purple line). (b) Zoom in on the mixed-phase cloud 1 with 1-second GPS data of payload trajectory (symbols) and balloon trajectory (dashed). (c) Detrended payload oscillations; approximate balloon sizes on the ground ($r = 1$ m) and at burst ($r = 5$ m) are shown by two circles. (d) Same as in (b) but for cloud 2. (e) Same as in (c) but for cloud 2. Colour code in (b-e): balloon ascent velocity.



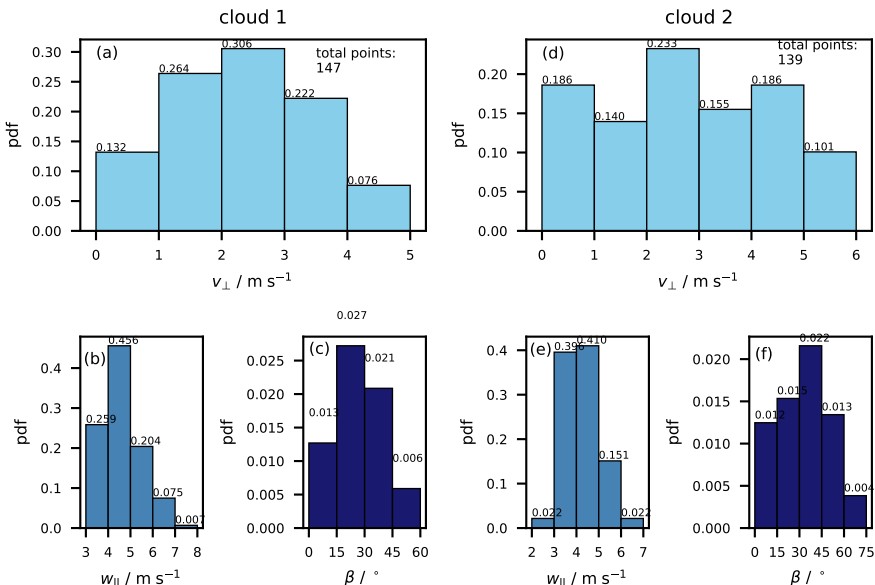

**Figure B5.** Probability density functions (pdf) of impingement parameters at the inlet plane of the CFH intake tube during the passage through the mixed-phase cloud of flight NT007. (a-c) cloud 1 between 6.25 and 7 km altitude; (d-f) refer to cloud 2 between 9.2 and 9.85 km altitude. (a and d) Velocity $v_\perp$ perpendicular to the tube walls; (b and e) velocity $w_{||}$ parallel to the axis of the tube; (c and f) impingement angle ($\beta$).



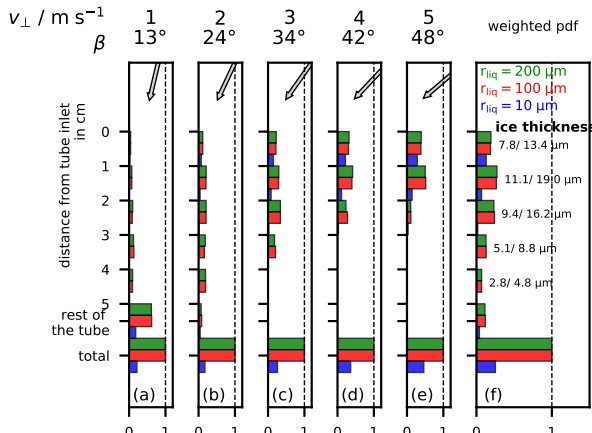

**Figure B6.** Collision/freezing efficiency of hydrometeors in the intake tube for mixed-phase cloud 1 of flight NT007 with average vertical inlet velocity $w_{||} = 4.5$ m s$^{-1}$. $r_{\text{liq}} = 200$ µm (green), $r_{\text{liq}} = 100$ µm (red), $r_{\text{liq}} = 10$ µm (blue). (a-e) Freezing efficiency for various velocities ($v_{\perp}$) perpendicular to the tube walls: (a) 1 m s$^{-1}$, 13°; (b) 2 m s$^{-1}$, 24°; (c) 3 m s$^{-1}$, 34°; (d) 4 m s$^{-1}$, 42°; (e) 5 m s$^{-1}$, 48°. The 'rest of the tube' takes account of all collisions occurring deeper than 5 cm inside the tube, including the mirror holder. (f) Weighted sum of the efficiencies in panels (a-e) by the horizontal velocity pdf of Figure B5a, in front of each bar we write the thickness of the subsequent ice layer considering radial homogeneous cover of the intake tube and the lower (left) and upper (right) LWC estimate for the cloud.



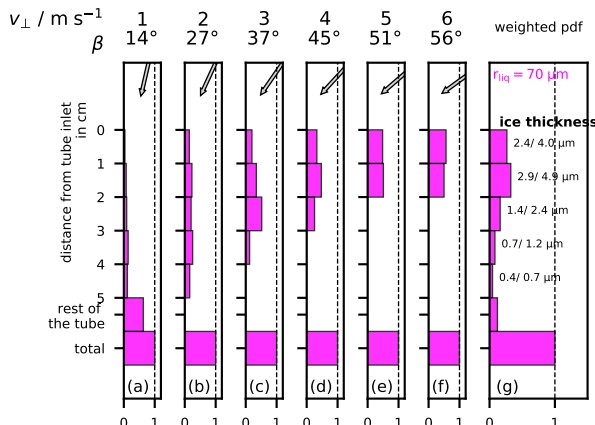

**Figure B7.** Collision/freezing efficiency of hydrometeors in the intake tube for mixed-phase cloud 2 of flight NT007 with average vertical inlet velocity $w_{||}$ = 4.0 m s$^{-1}$. $r_{\mathrm{liq}}$ = 70 μm (pink). (a-f) Freezing efficiency for various velocities ($v_\perp$) perpendicular to the tube walls: (a) 1 m s$^{-1}$, 14°; (b) 2 m s$^{-1}$, 27°; (c) 3 m s$^{-1}$, 37°; (d) 4 m s$^{-1}$, 45°; (e) 5 m s$^{-1}$, 51°; (e) 6 m s$^{-1}$, 56°. The 'rest of the tube' takes account of all collisions occurring deeper than 5 cm inside the tube, including the mirror holder. (g) Weighted sum of the efficiencies in panels (a-f) by the horizontal velocity pdf of Figure B5d, in front of each bar we write the thickness of the subsequent ice layer considering radial homogeneous cover of the intake tube and the lower (left) and upper (right) LWC estimate for the cloud.



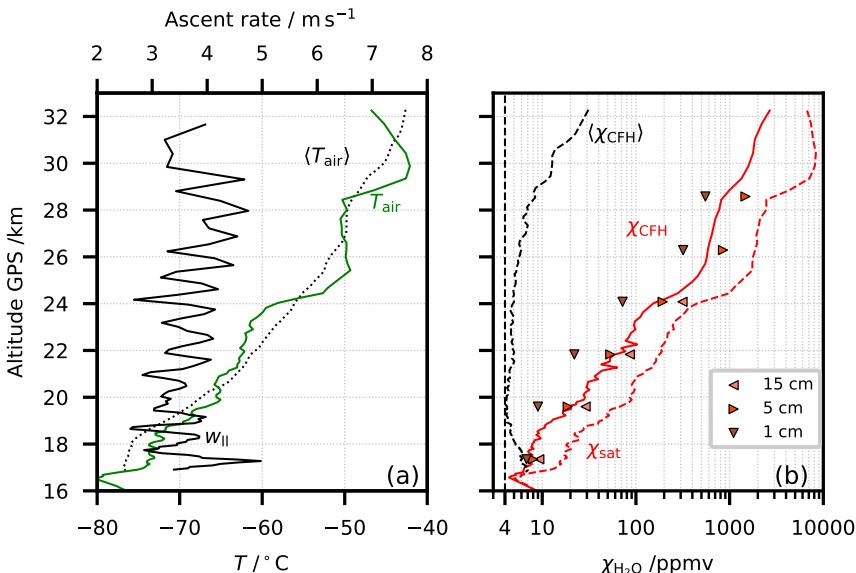

**Figure B8.** Flight NT007 and FLUENT simulation results for sublimation of ice in the intake tube in the stratosphere. (a) Green line: measured air temperature $T_{air}$; dotted black line: average air temperature for the 2016 Nainital summer campaign; solid black line: ascent velocity; (b) Solid red line: $H_2O$ mixing ratio measured by CFH during NT011 ($\chi_{CFH}$); dashed black line: average $H_2O$ mixing ratio for the uncontaminated soundings during the 2016 Nainital summer campaign($\langle\chi_{CFH}\rangle$); dashed red line: saturation $H_2O$ mixing ratio ($\chi_{sat}$); other symbols: FLUENT simulation results for the tube average mixing ratios $\langle\chi_{H_2O}\rangle_{Vol}$ in tubes with different ice coating depths $d$ coating the full circumference: ◄ $d = 15$ cm; ► $d = 5$ cm; ▼ $d = 1$ cm



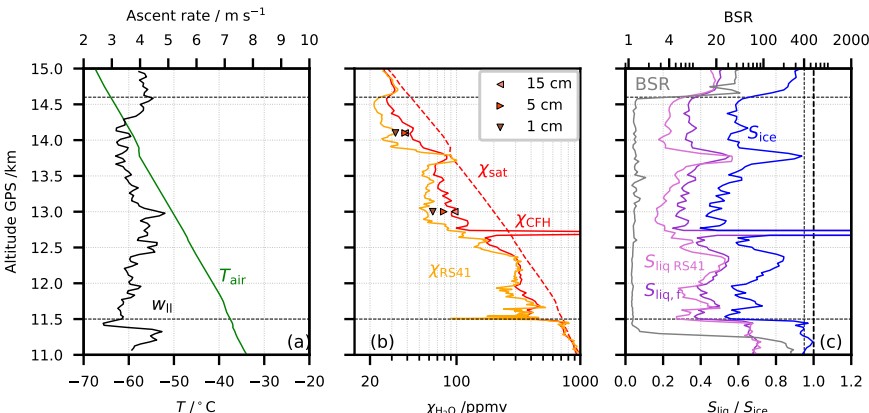

**Figure B9.** Flight NT007 and FLUENT simulation results for sublimation in the upper troposphere. (a) Green: air temperature; black: ascent velocity. (b) Red: $H_2O$ mixing ratio by the CFH; orange: $H_2O$ mixing ratio RS41; dashed red: saturation $H_2O$ mixing ratio for the air temperature; symbols: FLUENT simulation results for the tube average mixing ratios $\langle \chi_{H_2O} \rangle_{Vol}$ in tubes with different ice coating depths $d$ (full circumference): ◄ $d = 15$ cm; ► $d = 5$ cm; ▼ $d = 1$ cm; (c) Pink: saturation over water ($S_{liq\ RS41}$) by RS41; violet: saturation over water ($S_{liq\ f}$) from CFH considering the deposit on the mirror to be frost; blue: ice saturation ($S_{ice}$) from CFH; grey: 940-nm backscatter ratio from COBALD. Horizontal dashed lines limit the integration interval used for estimating the sublimated water in the upper troposphere.

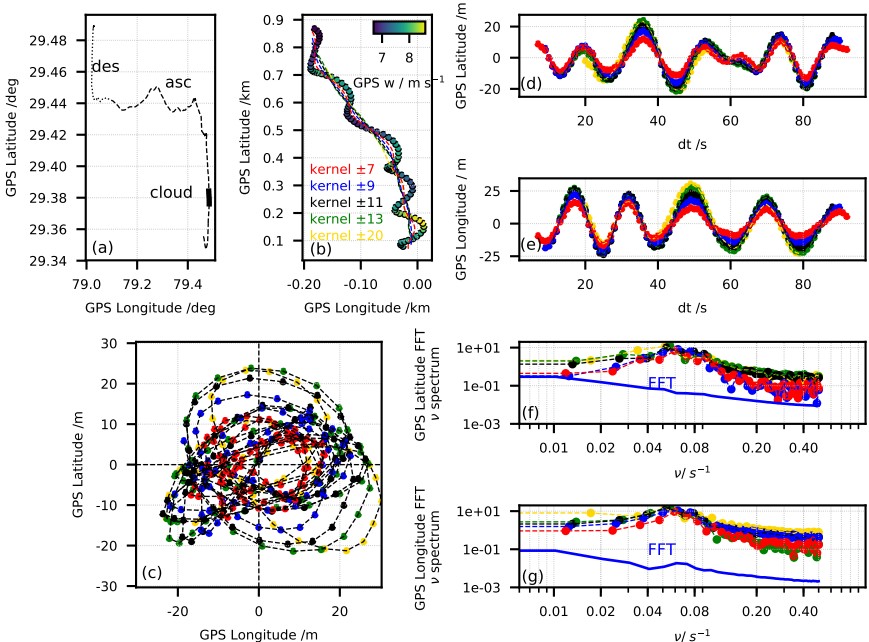

**Figure C1.** Fast Fourier Transform (FFT) analysis for the determination of the pendulum oscillation period applied to mixed-phase cloud of flight NT011. (a) Payload trajectory: ascent (dashed), descent (dotted) and mixed-phase cloud between 9.25 and 10 km altitude (thick black line). (b) Zoom-in of the mixed-phase cloud: payload trajectory (ascent velocity coloured dots) and balloon trajectories (dashed lines). The different lines have been smoothed from the payload trajectory with moving average intervals; yellow: ±20 s; green: ±13 s; black: ±11 s; blue: ±9 s; red: ±7 s. (c) Detrended payload oscillations. (d) Time series of the latitude values used in panel (c). (e) Time series of the longitude values used in panel (c). (f) Power spectral density of the latitude time series after FFT analysis. (g) Power spectral density of the longitude time series after FFT analysis.





**Table 1.** Lower cloud edge, thickness of cloud fraction containing supercooled liquid droplets, and estimated liquid water content (LWC) in mixed-phase clouds for flights NT011, NT029 and NT007.

| flights | in cloud | | | |
|---------|----------------------|----------------|----------------|----------------|
| | lower cloud edge (km) | thickness (m) | LWC (g m$^{-3}$) | |
| | | | lower estimate | upper estimate |
| NT007 | 6.25 and 9.2 | 750 + 600 | 0.080 + 0.020 | 0.137 + 0.034 |
| NT011 | 9.25 | 750 | 0.011 | 0.016 |
| NT029 | 8.1 | 1000 | 0.032 | 0.160 |



**Table 2.** FLUENT stratospheric and upper tropospheric simulations input data and results for flight NT011.

| | | | | | | | Simulations | |
| --- | --- | --- | --- | --- | --- | --- | --- | --- |
| | | Measurements | | | | 1 cm | 5 cm | 10 cm |
| $h$ | $p$ | $T$ | $w_{\parallel}$ | $\chi_{H_2O}$ | $\chi_{H_2O}$ reference | $\langle\chi_{H_2O}\rangle_{Vol}$ | $\langle\chi_{H_2O}\rangle_{Vol}$ | $\langle\chi_{H_2O}\rangle_{Vol}$ |
| (km) | (hPa) | (°C) | (m s$^{-1}$) | (ppmv) | (ppmv) | (ppmv) | (ppmv) | (ppmv) |
| Stratospheric: | | | | | | | | |
| 26.1 | 22 | -52.8 | 5.2 | 7 | 6[1] | 155 | 429 | 701 |
| 24.8 | 27 | -55.5 | 4.6 | 7 | 5[1] | 87 | 238 | 390 |
| 23.6 | 33 | -58.7 | 4.7 | 27 | 5[1] | 44 | 116 | 192 |
| 22.4 | 40 | -59.6 | 5.0 | 71 | 5[1] | 30 | 77 | 128 |
| 21.1 | 49 | -61.5 | 4.4 | 53 | 5[1] | 20 | 48 | 79 |
| 19.9 | 59 | -67.6 | 4.4 | 21 | 4[1] | 9 | 17 | 27 |
| 18.7 | 73 | -72.4 | 4.8 | 10 | 5[1] | 6 | 9 | 12 |
| 17.5 | 90 | -76.2 | 5.0 | 7 | 6[1] | 6 | 7 | 7 |
| Tropospheric: | | | | | | | | |
| 15.9 | 118 | -71.9 | 3.7 | 8 | 6[2] | 7 | 8 | 10 |
| 15.2 | 132 | -67.9 | 3.8 | 17 | 15[2] | 17 | 18 | 20 |
| 14.6 | 146 | -63.0 | 3.4 | 28 | 16[2] | 20 | 24 | 28 |
| 14.0 | 160 | -58.0 | 6.0 | 56 | 33[2] | 36 | 42 | 48 |

1: $\langle\chi_{CFH}\rangle$

2: $\chi_{RS41}$





**Table 3.** Integrated water vapour in the stratosphere and upper troposphere for flights NT011, NT029 and NT007.

| | Excess integrated Water Vapour (mg) | | |
|---|---|---|---|
| flights | upper troposphere | lower stratosphere | total |
| NT007 | 47.5 | 65.5 | 113 |
| NT011 | 1.45 | 4.35 | 5.80 |
| NT029 | 64 | 15.7 | 79.7 |



**Table 4.** Results for ice layer evolution in the stratosphere due to sublimation. The simulations 'isolated' refer to isolated 1 cm-long rings in the flow direction covering the entire inner circumference of the intake tube. These 1 cm-long rings start at different distances from the rim of the intake tube down to 4 cm. Results are given as extra $H_2O$ mixing ratio from the reference, which was $\chi_{H_2O} \sim 4$ ppmv, and as extra ice saturation. The mixing ratio corresponding to ice saturation for each of the simulation conditions is provided. The 'in group' simulations consider ice coverages of different length all starting at the rim of the intake tube. For these simulations, extra $\chi_{H_2O}$ and extra $S_{ice}$ are calculated as differences from subsequent simulations.

| | $p$ = 39 hPa, $T$ = -59.2 °C | | | | $p$ = 25 hPa, $T$ = -53.6 °C | | | | $p$ = 15 hPa, $T$ = -51.4 °C | | | |
|---|---|---|---|---|---|---|---|---|---|---|---|---|
| ice saturation in $\chi_{H_2O}$ at $T$ | 210 ppmv | | | | 1000 ppmv | | | | 2200 ppmv | | | |
| | isolated | | in group | | isolated | | in group | | isolated | | in group | |
| | extra | | extra | | extra | | extra | | extra | | extra | |
| | $\chi_{H_2O}$ | $S_{ice}$ | $\chi_{H_2O}$ | $S_{ice}$ | $\chi_{H_2O}$ | $S_{ice}$ | $\chi_{H_2O}$ | $S_{ice}$ | $\chi_{H_2O}$ | $S_{ice}$ | $\chi_{H_2O}$ | $S_{ice}$ |
| walls | (ppmv) | (%) | (ppmv) | (%) | (ppmv) | (%) | (ppmv) | (%) | (ppmv) | (%) | (ppmv) | (%) |
| 0 - 1 cm | 36 | 17% | – | – | 135 | 13% | – | – | 370 | 17% | – | – |
| 1 - 2 cm | 30 | 14% | 12 | 6% | 110 | 11% | 50 | 5% | 300 | 14% | 136 | 6% |
| 2 - 3 cm | 28 | 13% | 10 | 5% | 103 | 10% | 38 | 4% | 279 | 13% | 103 | 5% |
| 3 - 4 cm | 27 | 13% | 8 | 4% | 98 | 10% | 31 | 3% | 266 | 12% | 86 | 4% |
| 4 - 5 cm | 26 | 12% | 7 | 3% | 94 | 9% | 27 | 3% | 255 | 12% | 74 | 3% |





**Table 5.** Results for ice layer evolution in the stratosphere due to sublimation - continuation. From 5 cm from the rim of the intake tube, the isolated rings become 2-cm long down to 15 cm from the intake tube rim and the ice layers extending from the intake tube rim increase length in 2 cm steps also down to 15 cm from the rim of the intake tube. Only one of the pressure and temperature pairs used in the simulations shown in Table 4 is presented.

| ice saturation in $\chi_{H_2O}$ at $T$ | | | | |
|---|---|---|---|---|
| | $p = 25$ hPa, $T$ = -53.6 °C | | | |
| | 1000 ppmv | | | |
| | isolated | | in group | |
| | extra | | extra | |
| | $\chi_{H_2O}$ | $S_{ice}$ | $\chi_{H_2O}$ | $S_{ice}$ |
| walls* | (ppmv) | (%) | (ppmv) | (%) |
| 5 - 7 cm | 134 | 13% | 48 | 5 % |
| 7 - 9 cm | 124 | 12% | 39 | 4 % |
| 9 - 11 cm | 115 | 11% | 32 | 3 % |
| 11 - 13 cm | 106 | 11% | 27 | 3 % |
| 13 - 15 cm | 97 | 10% | 22 | 2 % |

* note that these walls are 2 cm long instead of 1 cm





**Table 6.** FLUENT input values (P, T and background $\chi_{H_2O}$ for typical tropospheric and stratospheric conditions) for simulations of contamination stemming from the balloon envelope and resulting $\chi_{H_2O}$ directly ($d$=0 m) and 5 to 10 m displaced from under de balloon ($d$=5-10 m), 55 m below the balloon - payload (CFH) location. . The balloon radius changes with height. Two different balloon ascent velocities ($w$) were considered.

| $p$ | $T$ | $r_{balloon}$ | background $\chi_{H_2O}$ | $w$ | $\chi_{H_2O}$ $d$=0 m | $d$=5-10 m |
|---|---|---|---|---|---|---|
| (hPa) | (°C) | (m) | (ppmv) | (m s$^{-1}$) | (ppmv) | (ppmv) |
| 10 | -40 | 4.0 | | | 428 | 40-5 |
| 20 | -50 | 3.1 | | | 48 | 10-5 |
| 50 | -60 | 2.3 | 5 | | 7.3 | 5 |
| 100 | -70 | 1.8 | | 7 | 5.5 | 5 |
| 200 | -40 | 1.5 | 100 | | 111 | |
| 800 | 25 | 1.0 | 20000 | | - | |



**Table A1.** FLUENT stratospheric and upper tropospheric simulations input data and results for flight NT029.

| | | | Measurements | | | | Simulations | |
| | | | | | | 1 cm | 5 cm | 15 cm |
| $h$ | $p$ | $T$ | $w_{\parallel}$ | $\chi_{H_2O}$ | $\chi_{H_2O}$ reference | $\langle\chi_{H_2O}\rangle_{Vol}$ | $\langle\chi_{H_2O}\rangle_{Vol}$ | $\langle\chi_{H_2O}\rangle_{Vol}$ |
| (km) | (hPa) | (°C) | (m s$^{-1}$) | (ppmv) | (ppmv) | (ppmv) | (ppmv) | (ppmv) |
|---|---|---|---|---|---|---|---|---|
| Stratospheric: | | | | | | | | |
| 30.8 | 11 | -47.7 | 8.1 | 21 | 17[1] | 666 | 2792 | NC |
| 29.7 | 13 | -48.5 | 6.6 | 40 | 12[1] | 521 | 1422 | NC |
| 28.5 | 15 | -51.4 | 7.2 | 71 | 8[1] | 278 | 771 | 1256 |
| 27.4 | 18 | -51.9 | 6.8 | 224 | 7[1] | 205 | 571 | 938 |
| 26.3 | 21 | -52.8 | 6.6 | 291 | 6[1] | 148 | 364 | 684 |
| 25.2 | 25 | -53.6 | 6.9 | 250 | 5[1] | 101 | 285 | 477 |
| 24.1 | 30 | -58.0 | 6.5 | 121 | 5[1] | 47 | 127 | 212 |
| 22.9 | 36 | -58.2 | 6.6 | 88 | 5[1] | 36 | 95 | 160 |
| 21.8 | 43 | -59.4 | 6.2 | 61 | 5[1] | 26 | 66 | 109 |
| 20.7 | 52 | -62.3 | 5.9 | 35 | 4[1] | 15 | 35 | 58 |
| 19.6 | 62 | -68.7 | 6.2 | 13 | 4[1] | 7 | 13 | 20 |
| 18.5 | 75 | -76.7 | 6.1 | 7 | 5[1] | 6 | 6 | 7 |
| 17.3 | 92 | -77.2 | 6.0 | 6 | 6[1] | 7 | 7 | 7 |
| Tropospheric: | | | | | | | | |
| 16.7 | 103 | -76.2 | 5.7 | 6 | 6[2] | 6 | 7 | 7 |
| 15.6 | 123 | -70.7 | 7.0 | 16 | 14[2] | 15 | 15 | 16 |
| 12.7 | 196 | -48.1 | 6.9 | 170 | 142[2] | 151 | 160 | 172 |
| 12.2 | 211 | -44.3 | 7.3 | 289 | 249[2] | 262 | 272 | 285 |

1: $\langle\chi_{CFH}\rangle$

2: $\chi_{RS41}$

NC: no convergence



**Table B1.** FLUENT stratospheric and upper tropospheric simulations input data and results for flight NT007.

| | | | | | | Simulations | | |
| | Measurements | | | | | 1 cm | 5 cm | 15 cm |
| $h$ | $p$ | $T$ | $w_{\parallel}$ | $\chi_{H_2O}$ | $\chi_{H_2O}$ reference | $\langle\chi_{H_2O}\rangle_{Vol}$ | $\langle\chi_{H_2O}\rangle_{Vol}$ | $\langle\chi_{H_2O}\rangle_{Vol}$ |
| (km) | (hPa) | (°C) | (m s$^{-1}$) | (ppmv) | (ppmv) | (ppmv) | (ppmv) | (ppmv) |
|---|---|---|---|---|---|---|---|---|
| Stratospheric: | | | | | | | | |
| 30.7 | 11 | -43.4 | 3.3 | 1831 | 16[1] | NC | NC | NC |
| 28.6 | 15 | -48.3 | 4.0 | 897 | 8[1] | 551 | 1477 | NC |
| 26.3 | 22 | -49.9 | 3.9 | 590 | 6[1] | 319 | 849 | NC |
| 24.1 | 30 | -56.8 | 3.5 | 190 | 5[1] | 72 | 193 | 314 |
| 21.8 | 43 | -62.6 | 3.7 | 66 | 5[1] | 22 | 53 | 86 |
| 19.6 | 63 | -67.0 | 3.4 | 21 | 4[1] | 9 | 19 | 29 |
| 17.4 | 92 | -74.0 | 3.8 | 7 | 6[1] | 7 | 8 | 9 |
| Tropospheric: | | | | | | | | |
| 14.1 | 158 | -59.6 | 3.1 | 44 | 31[2] | 32 | 38 | 38 |
| 13.0 | 188 | -50.4 | 4.5 | 93 | 56[2] | 64 | 79 | 97 |

1: $\langle\chi_{CFH}\rangle$

2: $\chi_{RS41}$

NC: no convergence