# Peer review of "Understanding cryogenic frost point hygrometer measurements after contamination by mixed-phase clouds"

_Atmospheric Measurement Techniques, 2020_

## Referee Comment (RC1) · Anonymous Referee #2 · 25 Jun 2020

This manuscript, with 34 figures and 3 appendices, is quite lengthy and, at times, highly technical, making it a very large meal to digest. In general, the paper is well-conceived and it should be of interest to AMT readers.

General Comments: I am not convinced that the appendices A and B add anything extraordinarily different from the main body of the paper. They do serve to greatly increase the paper's length by 4 text pages plus 17 figures (on top of the main body's 16 figures). I have never reviewed a standard (non-review) journal article with more than 20 figures, but here I will let the authors and editor decide about the necessary length for this paper.

[Figure]

I find that section 2.4, although interesting, is not really needed in this manuscript because the conclusions of this paper are not at all dependent on the modeling of mixed-phase clouds. Frost point hygrometer profiles showing the observed degree of contamination are undoubtedly afflicted by ice attached to the inside of the intake tube. This section could easily be removed to reduce the manuscript's length.

In all honesty, I was hoping that the very technical fluid dynamics modeling presented in great detail in this paper was going to result in a way to remove the effects of contamination from the measured profiles. I am guessing that the assumptions involved with such a procedure would cause the resulting corrected profiles to have very large uncertainties.

There are many grammar, clarity and language issues that I will try to help fix with the suggested changes below.

Specific Comments: Page 1 Line 1 (P1 L1): why do measurements only in the "(sub)tropical" UTLS provide "important information on air chemistry and climate"? Don't similar measurements in the mid-latitudes (where this contamination can also occur) also provide important information?

P1 L3: are the measurements rendered "difficult" or "unusable" by the contamination?

P1 L8: isn't the $60°$ maximum impingement angle somewhat determined by the length of the tether used to suspend the instrument below the balloon?

P1 L11: add "and unrealistically" before "high"

P1 L14: add "during ascent" after "only". This does not happen during descent.

P2 L2: The flight train outgassing contamination will affect all balloon-borne hygrometers, not just cryogenic FPs. Does hydrometeor contamination really affect hygrometers with a short or heated air intake?

P2 L5: "severe" implies worse than elsewhere, but the absolute contamination may be

worse at or below the tropopause than above it. I think you instead mean to say the "relative impact of contamination on the measurements is severe in the stratosphere".

P2 L7: Does increasing the tether length or preferential use of descent data help reduce the hydrometeor contamination, or just the flight train contamination? This statement sounds like both contamination types are influenced.

P2 L11: replace "ropes" with "thin hydrophobic tethers"

P2 L13: add "by radiosondes" after "temperature measurements"

P2 L15: I'm surprised the pioneering FP work by Brewer et al. (1948) is not mentioned here, although they used aircraft for their novel measurements, not balloons. I believe these were the first upper atmospheric water vapor measurements using FP hygrometry.

Brewer, A. W., Cwilong, B., and Dobson, G. M. B.: Measurement of Absolute Humidity in Extremely Dry Air, Proc. Phys. Soc., 60, 52–70, 1948.

P2 L22: Change to "Nearly all balloon-borne frost point hygrometer (FPH) soundings performed by NOAA's Global Monitoring Laboratory (Hall et al., 2016) use this valve."

P2 L24: change "of the instrument" to "by the instrument"

P2 L26: change to "using larger diameter stainless steel intake tubes that allow higher flow rates." Also, insert "the instrument" between "enabled" and "to"

P2 L29: "until today" makes it sound like their use has been discontinued. Please change to "These tubes are currently 2.5 cm"

P2 L30-32: change to "shielding the air flowing into the instrument from the contamination" and "containment" to "insulating container", add "mirror" before "surface" and change "extruding" to "extending"

P3 L12: do you mean "preferential" or "susceptible"?

P3 L18: it is also a common feature of soundings in mid-latitude convective regions like the Asian and North American monsoons

P4 L9: RH corrections for RS41 measurements are provided by the Vaisala MW41 software, not the sonde itself

P4 L11: "automated ground check" of what? A single point check? 0% RH or 100% RH?

P4 L13: "cold", "cryogenic" and "refrigerant" all imply the same thing. How about "against continuous cooling of the mirror by a cryogenic liquid."

P4 L13: change "air mass" to "air flowing past the mirror"

P4 L18: I presume these biases are for the Vaisala-corrected RS41 RH measurements. This should be stated here.

P4 L20: "could"? or "did"?

P4 L21: how is the 10 ppmv limit "empirical"? Isn't this instead a "realistic threshold"?

P4 L22,23: change "was" to "were" (data a is plural noun). Here and throughout the paper.

P4 L27: "the operation" is vague. Instead, describe the poor sensitivity at low RH values in a cold environment.

P4 L28: "clearing and freezing cycles" will not be understood by many readers. Please briefly describe why this is done.

P4 L30: and not just "ice", but "hexagonal ice" (rather than cubic ice)

P5 L5: How do the potential biases in RS41 temperature and pressure measurements increase the uncertainties of these comparisons?

P5 L13: change "would allow for very little vertical resolution" to "yields measurements at much lower vertical resolution than during ascent"

P5 L21: why do you presume that the mean (gray) profile is completely "uncontaminated"? There must be some proof. Very low flight-to-flight variability in these "uncontaminated" profiles? Comparisons to satellite profiles in the region? I think the term "uncontaminated" is not warranted here unless you provide some sort of evidence.

P5 L29: Please briefly state why the COBALD must only be flown at night

P6 L11: Here and throughout the paper. Please restrict the labeling of Figure markers, lines and curves in the body text, e.g., "air temperature from RS41 (green)", to Figure captions, otherwise you are simply repeating in the body text what is stated in the captions. When viewing the figures it is much easier for readers to consult the captions for this information than the body text.

P6 L13: Does the "freezing cycle at Tfrost= -15°C" include a "burn-off" of the existing condensate on the mirror followed by a re-growth of ice, or just a forced freezing of any liquid present on the mirror? If the former, why is the existing condensate first evaporated/sublimated?

P6 L18: How were "reasonable values" determined? Climatologies? Satellite profiles? Could high mixing ratios (>10 ppmv) actually be present in the LS due to overshooting convection? Jim Anderson and his group claim they measured >12 ppmv in the LS over the North American monsoon.

P6 L21: Some readers may not know what "glaciation" means in this context. Please briefly explain.

P6 L32: only "icing"? How about liquid water depositing on the warmer-than-ambient skin of the balloon? The balloon fill gas typically cools down at a slower rate than the ambient air temperature and keeps the balloon skin at super-ambient temperatures.

P7 L2: Some readers may not know what the Wegener-Bergeron-Findeisen process is. Please briefly explain.

P7 L13: I find this section, "Modelling of mixed-phase clouds", to be interesting, but

not really an essential part of this paper about the contamination of FP measurements. See my general comment above.

P8 L18: again, "mirror extrusion" doesn't make sense. The mirror is not extruded in manufacture nor an extrusion of any type. It is the mirror itself that extends into the flow of air.

P8 L20: "finite" is not needed here since angles cannot be "infinite". "Non-zero" is better terminology.

P8 L22: "rotational motion" may need explanation here, since it is more of a 3-dimensional motion than a 2-D "pendulum" motion. The payload does not rotate around itself (tumble), but around the vertical axis like a helicopter rotor. A quick explanation will clear up any possible misconceptions.

P9 L14: change "rubber" to "latex", since the balloon skin is synthetic, not natural

P9 L26" change "to stem form" to "stems from"

P9 L28: change "decomposed" to "separated"

P10 L4: I'm ok with the term "impingement angle", but not the use of "impingement" as a verb in this situation. I think "impact", which is both a noun and verb, is a better choice. "Droplets impacting the walls" or "Droplets that impacted the walls" is much clearer. Please change throughout the paper.

P10 L20: I do not understand what you mean by "also for the circulation around the equilibrium point". Are you addressing payload rotation (helicoptering) around the equilibrium point? Please clarify.

P10 L28: change "after the ice sublimated" to "until the ice sublimates"

P11 L14: "extends for 34 cm" from what? Presumably the insulating container?

P11 L15: more "extruding" and "extrusion" problems here. Change "extruding" to "that

extends", while the word "extrusion" can be omitted.

P13 L21: replace the comma with "while"

P14 L11: remove the "-" from in-homogeneously

P14 L26: it isn't clear what the phrase "air mass experienced by the mirror in real flight conditions" means here. Are you asserting that the entire flow of air through the instrument influences the frost point temperature, and not just the air flowing right next to the mirror? If so, I agree.

P16 L28: change to "during the traverse through the mixed-phase cloud"

P16 L31: change "water" to "ice" since liquids don't sublimate. Same for "condensate" in line 33

P17 L4: only solids sublimate, so the phrase "more water vapor sublimated" makes no sense. Similar problem P18 L24

P17 L30: "more hit" is awkward. How about "hit most frequently"? And "during the mixed-phase cloud" is also awkward, so please change to "within the mixed-phase cloud"

P17 L33: remove "with some water vapor"

P18 L2: Why is the range "4-8 ppmv" expected? 8 ppmv seems excessive for the altitude limits of balloons. However, in the LS, 8 ppmv might be possible from overshooting convection, but that would be very infrequently sampled.

P18 L14: change to "day-to-day"

P18 L15: change "have in average a dry bias" to "have, on average, a dry bias" and change to "flight-by-flight" (add hyphens)

P18 L17-19: "it was not clear whether RS41 had a dry bias or if CFH measured a too high humidity" sounds like a sold argument for NOT using the RS41 RH measurements

to check if the CFH measurements were contaminated. Then you emphatically state that this is what you did. This is somewhat confusing and needs to be re-written with greater clarity.

P19 L10: "CFH under-estimated the water vapour measurement in relation to the RS41" is awkward. The instrument doesn't "estimate" anything, it measures the frost point temperature. Please fix this sentence.

P19 L14-15: Combine these two sentences by including "1.45 mg" in the first sentence.

P19 L25: By "instrument payload" you are referring to the insulating container surrounding the CFH, correct? Something like a radiosonde on the other side of the container could not possibly contaminate the air flow into the CFH, correct?

P19 L30: add "(2084 masl)" after Nainital. This explains the surface pressure of 800 hPa.

P20 L10: in this instance, is "circular movement" what was earlier referred to "rotational motion"? If so, I prefer "circular movement" throughout the manuscript because it's meaning is perfectly clear, unlike "rotational motion".

P20 L13: "in this region at this pressure level" seems redundant

P20 L22: I don't think "exclude" is justified here, but identifying the balloon as "a minor contributor to contamination" is.

P20 L33: omit "mixing ratio" since it is clear what 12 ppmv is.

P21 L27: "As conclusion" is awkward. "In conclusion" is better.

P21 L21: change to: "We investigated the potential contamination of water vapor measurements . . ."

P22 L2: I'm pretty sure you didn't encounter mixed-phase clouds, but the balloon and payload certainly did.

[Figure]

P22 L3: Pardon my ignorance, but doesn't "mixed-phase" imply the presence of both liquid water and ice? Otherwise, what two phases are mixed in the cloud? So why is it even necessary to say that "liquid water was likely present in all of them"?

P22 L10: omit "already"

P22 L16: do you really mean "protecting" here? Or is "preserving" a better way to describe this?

P22 L22: "fast ascent balloon velocities" is awkward. I would remove "balloon".

P22 L25: replace "a slow balloon ascent through the entire flight between 3 and 4 m/s" with "the ascent rate was slow (3-4 m/s) for the entire flight"

P22 L29: the contamination does not "affect the operation of the CFH", it affects what is being measured.

P22 L31: replace "found in these cases" with "below 20 hPa during these three flights."

P22 L32: replace "the enhanced and contaminated water vapor values" with "the contamination"

P23 L1: what is a "two balloon tandem"? "flying two balloons separated" is clearer.

P23 L4: you showed (above) that the contamination from the balloon skin was nearly negligible, but now are concerned that a payload spending more time in the balloon wake would be more prone to contamination. Is more of something negligible necessarily a problem?

P23 L10: "atmospheric air" is redundant. Omit "atmospheric"

P23 L19: Just for your information, the older Vaisala RS92 did this with its dual RH sensors, deicing one while the other made measurements, then switching.

P23 L23: change to "We made many assumptions"

---

## Referee Comment (RC2) · Anonymous Referee #1 · 29 Jun 2020

Review of "Understanding cryogenic frost point hygrometer measurements after contamination by mixed-phase clouds" by Jorge et al.

It has been well known that stratospheric water vapor measurements may be heavily contaminated if the balloon payload passes through low and mid tropospheric clouds; however, the details of this mechanism have not yet been investigated. The manuscript by Jorge et al. investigates the parts of the balloon train that may generate this contamination and the physical processes that take place in the collection and release of the excess water.

The manuscript identifies that supercooled liquid water droplets may impinge on the

insides of the inlet tubes of the instrument driven by a significant radial velocity due to the pendulum motion of the payload and the off-vertical orientation of the payload. Depending on the pendulum amplitude and amount of water drops, the upper parts of the inlet tubes may receive large ice coatings than farther down the tube.

Contamination by the balloon wake is more likely to become significant near the burst altitude and may not be as significant in the lower stratosphere.

The paper uses a fluid dynamical model to study the freezing and sublimation processes as well as the mixing processes inside the tube and compare these with a series of in situ observations that triggered this study.

The manuscript is overall well written and strongly suggests processes inside the inlet tubes to be dominant. I can recommend publication of this manuscript after a few mostly technical corrections.

Detailed comments:

Page 13, line 33: How important is the assumption that the inlet tube is at the same temperature? The recommendation at the end may point towards a heated inlet tube. However, is heating of a few degrees sufficient, or will heating of many 10s of degrees be required to be effective? How might a colder inlet tube (possibly through infrared cooling at night) make the problem worse? A little bit of discussion about this assumption may be useful for the reader.

A few more words about ANSYS/FLUENT might be useful for readers, who are not familiar with CFD. ANSYS seems to be the manufacturer of the FLUENT software.

The appendix expands the paper significantly, but supports the main arguments. I can't tell if the manuscript may be too long and I would not suggest to remove it. It could be shortened if needed.

Technical comments:

Abstract, Line 1: delete "(sub)tropical". UTLS water vapor measurements are important in all geographic regions, not just the (sub-) tropics.

Introduction, line 31 on page 2: better "instrument's Styrofoam box".

Section 2.1, line 11: "automated" instead of "automatized"

Line 12: The instrument seems to control the reflectivity, which may not directly correlate to thickness.

Page 5, line 3: CFH instead of CHF

Page 5, line 10: Why do the authors use 1 hPa instead of more obvious constant altitude or constant time interval?

Page 11, line 17; space missing before 'cutcell'

Page 11: Remove the acronym SST, since it is not used except here.

Page 16, line 18, extra comma after "until"

Page 17, line 17; what means "extra ice saturation"? Maybe just delete this phrase.

Page 18, line 12; remove the hyphen after readily.

Page 19, line 10; add "the" before "CFH" (and a few other places).

Page 19, line 10; add "than" before "1.45 mg"

Page 19, line 29; maybe clearer: "which the balloon radius changed with pressure"

Figures 2 and other similar Figures: The colors are hard to distinguish, in particular pink, light purple and dark purple. Since there is no ambiguity about the ice or liquid on the mirror, maybe one of the traces could be removed.

Figure 2 b: The very high average mixing ratio near the top of the profile seems to go well above 10 ppmv, i.e. contaminated data may be part of this average.

Figure 4: The font in the Figure is too small.

Figure 6 c: The arrow for the inlet flow does not seem to be vertical as I would have expected. I assume the difference is due to the rotational speed. Can this be indicated in the Figure?

Figure 10e: The bottom half of the flow tube seems to have been shifted a little.

Figure 13: What determines the lower and upper limit of the vertical integration interval?

Figure 14, legend: The second (a) should be (c)

Figure 3 and Figure A2: The estimated region for the supercooled mixed phase clouds seems to use different selection criteria.

Table 1 and Table 3 legend: Move NT007 into first place following the ordering in the table.

---

## Author Comment (AC1) · 28 Sep 2020

Below are the comments from the referee in black and replies from the authors in blue.

**General comments**

This manuscript, with 34 figures and 3 appendices, is quite lengthy and, at times, highly technical, making it a very large meal to digest. In general, the paper is well-conceived and it should be of interest to AMT readers.

I am not convinced that the appendices A and B add anything extraordinarily different from the main body of the paper. They do serve to greatly increase the paper's length by 4 text pages plus 17 figures (on top of the main body's 16 figures). I have never reviewed a standard (non-review) journal article with more than 20 figures, but here I will let the authors and editor decide about the necessary length for this paper.

We agree with the referee that the two appendices contribute massively to the length of the manuscript. However, given there are only a very limited number of measurements offering themselves for such an analysis, the science of balloon-borne measurements gains by each additional case. The StratoClim dataset has only three cases showing this contamination phenomenon with the right combination of instruments. Therefore, we think that investigating and publishing the three cases provides significant additional support for the hypothesis developed in the manuscript. Given this dilemma we converted Appendices A and B into Supplementary Online Material (even though this does not fully satisfy the AMT rules on appendices and supplements). This will shorten the manuscript significantly.

I find that section 2.4, although interesting, is not really needed in this manuscript because the conclusions of this paper are not at all dependent on the modelling of mixed-phase clouds. Frost point hygrometer profiles showing the observed degree of contamination are undoubtedly afflicted by ice attached to the inside of the intake tube. This section could easily be removed to reduce the manuscript's length.

We admit this section was possibly not sufficiently motivated in the paper. The existence of liquid droplets in water sub-saturated clouds at these temperatures is unusual. Therefore, the micro-physical modelling of these clouds is an important prerequisite for the entire analysis, suggesting that the clouds are formed by many small and a few big droplets. The big water droplets can survive at the observed water subsaturation, after the small ones evaporate.

Furthermore, the mixed-phase cloud modelling increases the relevance of the manuscript. As mentioned in the conclusion section, "it was known that liquid clouds and warm mixed-phase clouds could irreversibly contaminate water vapour measurements by the CFH [Holger Vömel, personal communication, 2016], but our results show that even cold mixed-phase clouds with very low LWC can affect the measurement of water vapour by the CFH". It could be argued this is not a scientific breakthrough, but it complements our previous knowledge.

We improved the manuscript on page 7 lines 8 to 12: 'The existence of liquid droplets in water sub-saturated clouds at these temperatures ($T_{air}$ = -20 °C) is unusual. However, the passage through an ice cloud would not cause the observed contamination. The ice crystals likely bounce off the surfaces of the balloon, payload and intake tube. The presence of supercooled liquid droplets is necessary to form the ice layer inside the intake tube. Only supercooled liquid droplets freeze upon contact with a surface and lead to an icy surface coating of the balloon, payload and intake tube.'

In all honesty, I was hoping that the very technical fluid dynamics modelling presented in great detail in this paper was going to result in a way to remove the effects of contamination from the measured profiles. I am guessing that the assumptions involved with such a procedure would cause the resulting corrected profiles to have very large uncertainties.

Of course it would have been nice if our analysis could have been used to develop a "data correction recipe". However, there are a lot of uncertainties and assumptions involved in this study, especially concerning the properties of the tropospheric mixed-phase cloud, which make a systematic correction of the contaminated data impossible

There are many grammar, clarity and language issues that I will try to help fix with the suggested changes below.

We thank the referee for carefully reading the manuscript and all the suggestions.

**Specific comments:**

Page 1 Line 1 (P1 L1): why do measurements only in the "(sub)tropical" UTLS provide "important information on air chemistry and climate"? Don't similar measurements in the mid-latitudes (where this contamination can also occur) also provide important information?

We agree with the referee. The word (sub)tropical has been moved to line 4 to characterize the measurements used in this manuscript.

P1 L3: are the measurements rendered "difficult" or "unusable" by the contamination?

Yes, unusable. We replaced the word. However, this study identifies the cause of this contamination and suggest technical improvements. As discussed in the final section of the paper, the implementation of a heating cycle of the intake tubes will increase the technical effort, but might eventually make this type of contamination obsolete.

P1 L8: isn't the 60° maximum impingement angle somewhat determined by the length of the tether used to suspend the instrument below the balloon?

The length of the tether (55 m in our case) plays an important role determining the maximum impingement (in the new revised version of the manuscript 'impact') angle. The most significant contribution of the tether's length towards the impact angle is through the radius of the circular

movement that such tether allows. The 55 m-long tether allows the observed radius of the payload circular movements to be between r=10 m and r=20 m. If the tether was shorter or longer, we would observe a very different radius for the circular movement. However, it would have not been possible to calculate the impact angles by only considering the length of the tether. The treatment of the RS41 GPS data is essential to calculate the horizontal velocity induced by the circular movement and the angle of the intake tube relative to the ascent direction of the balloon.

P1 L11: add "and unrealistically" before "high"

Done (page 1 line 11).

P1 L14: add "during ascent" after "only". This does not happen during descent.

Done (page 1 line 14).

P2 L2: The flight train outgassing contamination will affect all balloon-borne hygrometers, not just cryogenic FPs. Does hydrometeor contamination really affect hygrometers with a short or heated air intake?

We agree with the referee. Outagssing from the flight train affects all balloon-borne hygrometers (page 2, line 4). However, we think the hydrometer contamination also affects hygrometers with short and/or heated air intake, e.g. the SnowWhite has a short heated intake duct, and still suffered from unexplained contamination as was reported in Cirisan et al. (2014). Furthermore, when the intake of the hygrometer is heated, the instrument measures total water content (TWC), instead of water vapour.

In Section 5.4.2 of this manuscript, we show by means of CFD simulation that with a shorter intake tube the sampled air might also be contaminated by water vapour outgassing from the instrument's box. If the instrument goes through a mixed phase cloud, the hydrometeors will also impact on the instruments box. From the measurement of an instrument with a shorter intake tube, it will be difficult to distinguish the source of the contamination: the hydrometeors which impacted inside the intake tube or the ones which impacted the box.

P2 L5: "severe" implies worse than elsewhere, but the absolute contamination may be worse at or below the tropopause than above it. I think you instead mean to say the "relative impact of contamination on the measurements is severe in the stratosphere".

We agree with the referee. Done (page 2 line 5 to 6).

P2 L7: Does increasing the tether length or preferential use of descent data help reduce the hydrometeor contamination, or just the flight train contamination? This statement sounds like both contamination types are influenced.

We agree with the referee. The statement has been rephrased (page 2 line 7).

P2 L11: replace "ropes" with "thin hydrophobic tethers"

Done (page 2 line 11 to 12).

P2 L13: add "by radiosondes" after "temperature measurements"

Done (page 2 line 14).

P2 L15: I'm surprised the pioneering FP work by Brewer et al. (1948) is not mentioned here, although they used aircraft for their novel measurements, not balloons. I believe these were the first upper atmospheric water vapor measurements using FP hygrometry.

Brewer, A. W., Cwilong, B., and Dobson, G. M. B.: Measurement of Absolute Humidity in Extremely Dry Air, Proc. Phys. Soc., 60, 52–70, 1948.

We did not mean to exclude this pioneering work, but found that the paper focus more on balloon borne contamination. In this sense, these measurements from an aircraft platform do not fit the message of the paper. Therefore, we changed the title of the paper to: "Understanding balloon-borne frost point hygrometer measurements after contamination by mixed-phase clouds".

P2 L22: Change to "Nearly all balloon-borne frost point hygrometer (FPH) soundings performed by NOAA's Global Monitoring Laboratory (Hall et al., 2016) use this valve."

Done (page 2 line 23 to 24).

P2 L24: change "of the instrument" to "by the instrument"

Done (page 2 line 25).

P2 L26: change to "using larger diameter stainless steel intake tubes that allow higher flow rates." Also, insert "the instrument" between "enabled" and "to"

Done (page 2 line 27). Done (page 2 line 28).

P2 L29: "until today" makes it sound like their use has been discontinued. Please change to "These tubes are currently 2.5 cm"

Done (page 2 line 30). We added a more recent reference to the sentence (page 2 line 31).

P2 L30-32: change to "shielding the air flowing into the instrument from the contamination" and "containment" to "insulating container", add "mirror" before "surface" and change "extruding" to "extending"

Done (page 2 line 32). Upon suggestion from the Anonymous Referee # 1 we replaced "containment" by "box" (page 2 line 33). Done (page 2 line 33). The term "extending" sounds as if the mirror starts at the wall and extends until 1.25 cm from the wall. The authors replaced "extruding" by "displaced" (page 2 line 33).

P3 L12: do you mean "preferential" or "susceptible"?

We want to convey that from all the surfaces of the CFH exposed to the environment, the intake tubes are the most likely surface to be subject to icing and to cause contamination of the measurement. In this sense, the word susceptible is a better fit. Done (page 3 line 12).

P3 L18: it is also a common feature of soundings in mid-latitude convective regions like the Asian and North American monsoons

The statement has been rephrased (page 3, line 15).

P4 L9: RH corrections for RS41 measurements are provided by the Vaisala MW41 software, not the sonde itself

Done (page 4 line 14).

P4 L11: "automated ground check" of what? A single point check? 0% RH or 100% RH?

Here, we mean the MW41 ground check before launching the RS41. The "automated ground check" checks the capacitive sensors $RH = 0\%$ by heating the sensor until all humidity has been blown off and it compares the pressure sensor measurement to a user inserted pressure value. For this campaign, we also did an 100% humidity chamber check, but this is not automatically accounted for by the MW41 software. The statement has been changed to "zero humidity automated ground check" in the manuscript (page 4 line 16).

P4 L13: "cold", "cryogenic" and "refrigerant" all imply the same thing. How about "against continuous cooling of the mirror by a cryogenic liquid."

Done (page 4, line 18).

P4 L13: change "air mass" to "air flowing past the mirror"

Done (page 4, line 19).

P4 L18: I presume these biases are for the Vaisala-corrected RS41 RH measurements. This should be stated here.

Brunamonti et al. (2019) uses Vaisala-corrected RS41 RH measurements. Done (page 14, line 25).

P4 L20: "could"? or "did"?

Discrepancies of 50% $H_2O$ mixing ratio between the CFH and the Vaisala-corrected RS41 RH measurements did occur. Discrepancies as high as 100% were also recorded. Done (page 4, line 26).

P4 L21: how is the 10 ppmv limit "empirical"? Isn't this instead a "realistic threshold"?

The 10 ppmv threshold is realistic, however it is also empirical. There are two schools of stratospheric water vapour measurements. One school accepts that measurements can be discarded

if they seam unrealistic, the other tries to extensively explain by physical processes how could these measurements happen. The authors of this manuscript and of Brunamonti et al. (2019) clearly belong to the second school. So, we prefer to call it an empirical threshold. (page 4 line 27)

P4 L22,23: change "was" to "were" (data a is plural noun). Here and throughout the paper.

Since the use of the word "data" is accepted in English as a singular and as a plural, we will wait for the editor's and typewriter's decision regarding this point.

P4 L27: "the operation" is vague. Instead, describe the poor sensitivity at low RH values in a cold environment.

Done (page 5, line 2).

P4 L28: "clearing and freezing cycles" will not be understood by many readers. Please briefly describe why this is done.

The following explanation was added to the manuscript: 'The clearing and freezing cycle consists of a forced heating of the CFH mirror to blow-off any deposit, followed by a forced cooling of the mirror. During the cycle at approximately -15 °C, the mirror is forced cooled to temperatures below which ice certainly forms (¡-40 °C). During the second cycle at approximately -53 °C, the mirror is forced cooled to temperatures below which hexagonal ice forms (¡-82 °C). Hexagonal ice is more stable than cubic ice. The data collected during the freezing and clearing cycles is not used for further analysis, but we do not remove it from the water vapour profiles. This feature gives us confidence that after it the phase of the deposit in the mirror was ice or hexagonal ice.' (page 5, line 4 to 9).

P4 L30: and not just "ice", but "hexagonal ice" (rather than cubic ice)

Done (page 5, line 7 and 9).

P5 L5: How do the potential biases in RS41 temperature and pressure measurements increase the uncertainties of these comparisons?

Uncertainties arising from the treatment of the basic measurements through the water vapour parameterisation of Murphy and Koop 2005 and Hardy 98 are not relevant for the goal of this paper.

P5 L13: change "would allow for very little vertical resolution" to "yields measurements at much lower vertical resolution than during ascent"

Done (page 5, line 23).

P5 L21: why do you presume that the mean (gray) profile is completely "uncontaminated"? There must be some proof. Very low flight-to-flight variability in these "uncontaminated" profiles? Comparisons to satellite profiles in the region? I think the term "uncontaminated" is not warranted here unless you provide some sort of evidence.

All the water vapour measurements by the CFH in the 16/17 StratoClim dataset have been extensively analysed in previous publication, see Brunamonti et al. (2018, 2019). The measurements have not been compared to satellite data but to the ECMWF operational and reanalysis products. The CFH water vapour measurements in the stratosphere show very little flight-to-flight variability. However, they still show higher variability than the model products. They would also show higher variability than the satellite measurement. However, this is the nature and the purpose of balloon borne water vapour measurements. They allow for better vertical and horizontal resolution of water vapour features. Nevertheless, we agree that the use of the expression "uncontaminated" is abusive without some sort of evidence. So, we have reformulated it to "excluding contaminated profiles". The change was applied throughout the manuscript.

P5 L29: Please briefly state why the COBALD must only be flown at night

The following explanation was added to the manuscript: 'The COBALD can only be flown at night because daylight saturates the photodetector (Cirisan et al., 2014).' (page 5 line 32 and 33).

P6 L11: Here and throughout the paper. Please restrict the labeling of Figure markers, lines and curves in the body text, e.g., "air temperature from RS41 (green)", to Figure captions, otherwise you are simply repeating in the body text what is stated in the captions. When viewing the figures it is much easier for readers to consult the captions for this information than the body text.

Done where applicable.

P6 L13: Does the "freezing cycle at Tfrost= -15 °C" include a "burn-off" of the existing condensate on the mirror followed by a re-growth of ice, or just a forced freezing of any liquid present on the mirror? If the former, why is the existing condensate first evaporated/sublimated?

In the existing literature the burn-off of the condensate on the mirror at about -15 °C is not mentioned. Nevertheless, it is present as can be seen in Figure 2. We think that if the ice layer is formed just with a freezing cycle, there is a risk, in case there was already a mixture of liquid water and ice on the mirror, that the "final" ice layer is very inhomogeneous - with ice crystals of different sizes instead of a smooth ice layer. The burn-off at this temperature will eliminate all condensate. The subsequent fast cooling of the mirror allows for the formation of a more homogeneous ice layer.

Vömel et al. (2016) shows images of condensate layers at different conditions and temperatures. Although the controlling of a coarse ice layer with liquid patches at relatively warm temperatures (T = -26.8 °C) is stable, the controlling of a similarly looking condensate, although totally frozen, at lower temperatures is no longer stable.

We do not think that further discussion of the clearing and freezing cycles is relevant for the manuscript. The clearing and freezing cycles of the CFH are already discussed somewhere else in the paper (page 5 line 4 to 9).

P6 L18: How were "reasonable values" determined? Climatologies? Satellite profiles? Could high mixing ratios (> 10 ppmv) actually be present in the LS due to overshooting convection? Jim Anderson and his group claim they measured > 12 ppmv in the LS over the North American monsoon.

In this context, "reasonable values" mean values similar to those observed in the season average of the water vapour mixing ratio measured by the CFH excluding the contaminated profiles - the black line in Figure 2b.

P6 L21: Some readers may not know what "glaciation" means in this context. Please briefly explain.

We have rephrased the sentence: 'The lower cloud has $S_{\mathrm{liq}} < 1$ and is sufficiently cold that the presence of liquid water is unlikely' (page 6, line 32 and 33). Glaciation time is explained later (page 7, line 25).

P6 L32: only "icing"? How about liquid water depositing on the warmer-than-ambient skin of the balloon? The balloon fill gas typically cools down at a slower rate than the ambient air temperature and keeps the balloon skin at super-ambient temperatures.

As long as the balloon is colder than 0 °C, super cooled water will freeze on impact with the balloon skin. At temperatures warmer than 0 °C, any liquid water that is not absorbed by the balloon skin, will most likely run off the skin of the balloon or evaporate once the balloon is in a sub-saturated region.

P7 L2: Some readers may not know what the Wegener-Bergeron-Findeisen process is. Please briefly explain.

We added an explanation of why we use the Wegener-Bergeron-Findeisen process here: 'Subsequently, we asked whether the balance between the different water phases described by the Wegener-Bergeron-Findeisen (Pruppacher and Klett, 1997; Korolev et al., 2017) process would provided enough time for the flights to encounter supercooled liquid droplets at these high altitudes and low temperatures' (page 7, line 13 to 15). Any further explanation is not in the scope of this paper, but can be found in the references provided.

P7 L13: I find this section, "Modelling of mixed-phase clouds", to be interesting, but not really an essential part of this paper about the contamination of FP measurements. See my general comment above.

Answered above.

P8 L18: again, "mirror extrusion" doesn't make sense. The mirror is not extruded in manufacture nor an extrusion of any type. It is the mirror itself that extends into the flow of air.

We consider the mirror to be the surface parallel to the air flow, where the ice layer forms. "Mirror extrusion" refers to the "mirror part" which is perpendicular to the flow and allows

the mirror to stay halfway inside of the intake tube. We have replace "mirror extrusion" with "mirror holder" everywhere. The "mirror holder" defention is provided in page 9 line 4.

P8 L20: "finite" is not needed here since angles cannot be "infinite". "Non-zero" is better terminology.

Done (page 9, line 7).

P8 L22: "rotational motion" may need explanation here, since it is more of a 3- dimensional motion than a 2-D "pendulum" motion. The payload does not rotate around itself (tumble), but around the vertical axis like a helicopter rotor. A quick explanation will clear up any possible misconceptions.

Replaced by "circular movement" everywhere as suggested below.

P9 L14: change "rubber" to "latex", since the balloon skin is synthetic, not natural

It is not relevant, so we removed it.

P9 L26" change "to stem form" to "stems from"

Done (page 10, line 16).

P9 L28: change "decomposed" to "separated"

Done (page 10, line 18).

P10 L4: I'm ok with the term "impingement angle", but not the use of "impingement" as a verb in this situation. I think "impact", which is both a noun and verb, is a better choice. "Droplets impacting the walls" or "Droplets that impacted the walls" is much clearer. Please change throughout the paper.

Thanks for the suggested clarification, which we implement.

P10 L20: I do not understand what you mean by "also for the circulation around the equilibrium point". Are you addressing payload rotation (helicoptering) around the equilibrium point? Please clarify.

We agree with the referee. We rephrased the sentence (page 11, line 8).

P10 L28: change "after the ice sublimated" to "until the ice sublimates"

Done (page 11, line 16 and 17).

P11 L14: "extends for 34 cm" from what? Presumably the insulating container?

The intake tube is 34 cm long from intake to outlet. We have rephrased the sentence. (page 12, line 5).

P11 L15: more "extruding" and "extrusion" problems here. Change "extruding" to "that extends", while the word "extrusion" can be omitted.

We rephrased the sentence (page 12, line 7 and 8).

P13 L21: replace the comma with "while"

Done (page 14, line 14).

P14 L11: remove the "- " from in-homogeneously

Done (page 15, line 9).

P14 L26: it isn't clear what the phrase "air mass experienced by the mirror in real flight conditions" means here. Are you asserting that the entire flow of air through the instrument influences the frost point temperature, and not just the air flowing right next to the mirror? If so, I agree.

We appreciate the comment of the referee and have rephrased the paragraph: 'We believe that the entire flow of air through the intake tube influences the frost point temperature, and not just the air flowing right next to the mirror.' (page 15, line 24 to 26).

P16 L28: change to "during the traverse through the mixed-phase cloud"

Done (page 17, line 28).

P16 L31: change "water" to "ice" since liquids don't sublimate. Same for "condensate" in line 33

Done (page 17, line 31 and 33).

P17 L4: only solids sublimate, so the phrase "more water vapor sublimated" makes no sense. Similar problem P18 L24

Done everywhere sublimation was not associated with ice.

P17 L30: "more hit" is awkward. How about "hit most frequently"? And "during the mixed-phase cloud" is also awkward, so please change to "within the mixed-phase cloud"

Done (page 18, line 29).

P17 L33: remove "with some water vapor"

Done (page 18, line 32).

P18 L11: Why is the range "4-8 ppmv" expected? 8 ppmv seems excessive for the altitude limits of balloons. However, in the LS, 8 ppmv might be possible from overshooting convection, but that would be very infrequently sampled.

We thank the reviewer and change the "4-8 ppmv" to "2-6 ppmv". (page 19 line 11)

P18 L14: change to "day-to-day"

Done (page 19, line 13).

P18 L15: change "have in average a dry bias" to "have, on average, a dry bias" and change to "flight-by-flight" (add hyphens)

Done (page 19, line 12 and 13).

P18 L17-19: "it was not clear whether RS41 had a dry bias or if CFH measured a too high humidity" sounds like a sold argument for NOT using the RS41 RH measurements to check if the CFH measurements were contaminated. Then you emphatically state that this is what you did. This is somewhat confusing and needs to be re-written with greater clarity.

We see the reviewer's point. We clarified this by rewriting as follows: "Brunamonti et al. (2019) found the RS41 to have, on average, a dry bias in comparison with the CFH in the upper troposphere during StratoClim. However, in a flight-by-flight comparison, when the CFH was contaminated, it was not clear whether the RS41 had a dry bias or the CFH measured a too high humidity. As a conservative assumption, we assumed the RS41 water vapour measurement to be correct and we used it as reference for the analysis of the CFH contamination in the upper troposphere." (page 19, lines 18 and 19)

P19 L10: "CFH under-estimated the water vapour measurement in relation to the RS41" is awkward. The instrument doesn't "estimate" anything, it measures the frost point temperature. Please fix this sentence.

Fixed (page 20, line 10 to 11).

P19 L14-15: Combine these two sentences by including "1.45 mg" in the first sentence.

We have rephrased these sentences according to the Anonymous Referee # 1 suggestion: 'To estimate an upper limit for the LWC in the mixed phase cloud, we compared the total water vapour measured by the CFH and the RS41 using Formula (8) in the interval between the top of the lower cloud and the cirrus cloud at the tropopause (from 13.5 to 17 km altitude). We concluded that the CFH measured at least 1.45 mg of water more than the RS41 in this interval.' (page 20, line 14 to 16).

P19 L25: By "instrument payload" you are referring to the insulating container surrounding the CFH, correct? Something like a radiosonde on the other side of the container could not possibly contaminate the air flow into the CFH, correct?

Yes, it is more likely that the air flow into the CFH is contaminated by ice sublimating from the CFH box than by ice sublimating from the radiosonde, but just because the CFH box is closer to the opening of the intake tube than the radiosonde.

P19 L30: add "(2084 masl)" after Nainital. This explains the surface pressure of 800 hPa.

We appreciate the referee's diligence providing the altitude of Nainital. However, the launch location was at ARIES - Aryabhatta Research Institute of Observational Sciences which stands at 1820 masl. We have added the altitude above sea level for ARIES but kept the location as Nainital for simplicity (page 20, line 30).

P20 L10: in this instance, is "circular movement" what was earlier referred to "rotational motion"? If so, I prefer "circular movement" throughout the manuscript because it's meaning is perfectly clear, unlike "rotational motion".

Changed everywhere as suggested by the referee, see above.

P20 L13: "in this region at this pressure level" seems redundant

We have removed this sentence (page 21, line 13).

P20 L22: I don't think "exclude" is justified here, but identifying the balloon as "a minor contributor to contamination" is.

Thanks, we have rephrased this sentence as suggested. (page 21, line 22 and 23).

P20 L33: omit "mixing ratio" since it is clear what 12 ppmv is.

Done (page 21, line33).

P21 L27: "As conclusion" is awkward. "In conclusion" is better.

Done (page 22, line 26).

P21 L21: change to: "We investigated the potential contamination of water vapor measurements ..."

Done (page 22, line 31).

P22 L2: I'm pretty sure you didn't encounter mixed-phase clouds, but the balloon and payload certainly did.

We rephrased the sentence (page 23, line 2).

P22 L3: Pardon my ignorance, but doesn't "mixed-phase" imply the presence of both liquid water and ice? Otherwise, what two phases are mixed in the cloud? So why is it even necessary to say that "liquid water was likely present in all of them"?

The referee is correct, "mixed-phase" imply the presence of both liquid water and ice. However, the observed $S_{\text{liq}}$ within the clouds does not allow us to immediately infer the presence of liquid in these clouds. The modelling of the cloud suggests a scenario where the presence of liquid droplets is compatible with the observed $S_{\text{liq}}$. We have rephrased the motivation of Section 2.4 to convey this message (page 6, line 32 and 33 and page 7 lines 8 to 12). We will maintain the emphasis on the presence of liquid water in these mixed-phase clouds in the conclusion (page

23, page 2).

P22 L10: omit "already"

Done (page 23, line 9).

P22 L16: do you really mean "protecting" here? Or is "preserving" a better way to describe this?

We agree with the referee. Done (page 23, line 16).

P22 L22: "fast ascent balloon velocities" is awkward. I would remove "balloon".

Done (page 23, line 22).

P22 L25: replace "a slow balloon ascent through the entire flight between 3 and 4 m/s" with "the ascent rate was slow (3-4 m/s) for the entire flight"

Done (page 23, line 24 and 25).

P22 L29: the contamination does not "affect the operation of the CFH", it affects what is being measured.

We rephrased the sentence (page 23, line 28).

P22 L31: replace "found in these cases" with "below 20 hPa during these three flights."

Done (page 23, line 30).

P22 L32: replace "the enhanced and contaminated water vapor values" with "the contamination"

We rephrased this sentence to refer to the contamination in the season average water vapour profile (page 23, line 31 and 32).

P23 L1: what is a "two balloon tandem"? "flying two balloons separated" is clearer.

Done (page 24, line 2).

P23 L4: you showed (above) that the contamination from the balloon skin was nearly negligible, but now are concerned that a payload spending more time in the balloon wake would be more prone to contamination. Is more of something negligible necessarily a problem?

Above, we showed that the contamination is negligible, if the payload oscillates outside of a certain range directly below the balloon, i.e. when the radius of the circular movement is > 5 m. If the radius of the circular movement is < 5 m, the risk of contamination by ice sublimating from the balloon skin becomes significant above the 50 hPa level, see Figure 15.

P23 L10: "atmospheric air" is redundant. Omit "atmospheric"

Done (page 24, line 10).

P23 L19: Just for your information, the older Vaisala RS92 did this with its dual RH sensors, deicing one while the other made measurements, then switching.

Noted.

P23 L23: change to "We made many assumptions"

Done (page 24, line 23).

**References**

Brunamonti, S., Jorge, T., Oelsner, P., Hanumanthu, S., Singh, B. B., Kumar, K. R., Sonbawne, S., Meier, S., Singh, D., Wienhold, F. G., Luo, B. P., Böttcher, M., Poltera, Y., Jauhiainen, H., Kayastha, R., Dirksen, R., Naja, M., Rex, M., Fadnavis, S., and Peter, T.: Balloon-borne measurements of temperature, water vapor, ozone and aerosol backscatter at the southern slopes of the Himalayas during StratoClim 2016-2017, Atmospheric Chemistry and Physics, 2018, 1–38, https://doi.org/10.5194/acp-2018-222, 2018.

Brunamonti, S., Füzér, L., Jorge, T., Poltera, Y., Oelsner, P., Meier, S., Dirksen, R., Naja, M., Fadnavis, S., Karmacharya, J., Wienhold, F. G., Luo, B. P., Wernli, H., and Peter, T.: Water Vapor in the Asian Summer Monsoon Anticyclone: Comparison of Balloon-Borne Measurements and ECMWF Data, Journal of Geophysical Research: Atmospheres, https://doi.org/10.1029/2018jd030000, 2019.

Cirisan, A., Luo, B. P., Engel, I., Wienhold, F. G., Sprenger, M., Krieger, U. K., Weers, U., Romanens, G., Levrat, G., Jeannet, P., Ruffieux, D., Philipona, R., Calpini, B., Spichtinger, P., and Peter, T.: Balloon-borne match measurements of midlatitude cirrus clouds, Atmospheric Chemistry and Physics, 14, 7341–7365, https://doi.org/10.5194/acp-14-7341-2014, 2014.

Korolev, A., McFarquhar, G., Field, P. R., Franklin, C., Lawson, P., Wang, Z., Williams, E., Abel, S. J., Axisa, D., Borrmann, S., Crosier, J., Fugal, J., Krämer, M., Lohmann, U., Schlenczek, O., Schnaiter, M., and Wendisch, M.: Mixed-Phase Clouds: Progress and Challenges, Meteorological Monographs, 58, 5.1–5.50, https://doi.org/10.1175/amsmonographs-d-17-0001.1, 2017.

Pruppacher, H. R. and Klett, J. D.: Microphysics of Clouds and Precipitation., Kluwer Academic Publishers, 1997.

Vömel, H., Naebert, T., Dirksen, R., and Sommer, M.: An update on the uncertainties of water vapor measurements using cryogenic frost point hygrometers, Atmospheric Measurement Techniques, 9, 3755–3768, https://doi.org/10.5194/amt-9-3755-2016, 2016.

---

## Author Comment (AC2) · 28 Sep 2020

Below are the comments from the referee in black and replies from the authors in blue.

**General comments**

It has been well known that stratospheric water vapor measurements may be heavily contaminated if the balloon payload passes through low and mid tropospheric clouds; however, the details of this mechanism have not yet been investigated. The manuscript by Jorge et al. investigates the parts of the balloon train that may generate this contamination and the physical processes that take place in the collection and release of the excess water.

The manuscript identifies that supercooled liquid water droplets may impinge on the insides of the inlet tubes of the instrument driven by a significant radial velocity due to the pendulum motion of the payload and the off-vertical orientation of the payload. Depending on the pendulum amplitude and amount of water drops, the upper parts of the inlet tubes may receive large ice coatings than farther down the tube.

Contamination by the balloon wake is more likely to become significant near the burst altitude and may not be as significant in the lower stratosphere.

The paper uses a fluid dynamical model to study the freezing and sublimation processes as well as the mixing processes inside the tube and compare these with a series of in situ observations that triggered this study.

The manuscript is overall well written and strongly suggests processes inside the inlet tubes to be dominant. I can recommend publication of this manuscript after a few mostly technical corrections.

We are grateful to the referee for carefully reading the paper and providing valuable suggestions.

**Detailed comments:**

Page 13, line 33: How important is the assumption that the inlet tube is at the same temperature? The recommendation at the end may point towards a heated inlet tube. However, is heating of a few degrees sufficient, or will heating of many 10s of degrees be required to be effective? How might a colder inlet tube (possibly through infrared cooling at night) make the problem worse? A little bit of discussion about this assumption may be useful for the reader.

The temperature of the intake tube and ice layer in the ice sublimation CFD simulations is very important. This temperature is a crucial part of determining how much water vapour is transferred to the incoming dry air. Hence, it determines the level of contamination observed in the intake tube. The extend of the ice layer inside the intake tube also plays a crucial role

on determining the overall contamination.

Philipona et al. (2013) determined that the combination of incoming and outgoing long- and short-wave radiation produces radiative heating of very thin thermocouples in the upper troposphere and lower stratosphere (UTLS). The effects are very similar during the day and night. Following this account, we could expect the intake tubes to be warmer in the region of interest - UTLS. We should also consider that the tube has thermal inertia and is travelling from a region of colder into warmer air, at least in the lower stratosphere. Nevertheless, the expected effect of thermal radiation is less than 1 °C for the thermocouples, we assume it would be the same for the intake tubes. This temperature difference does not cause a significant effect in the results of our contamination simulation. Especially, if we also account for the uncertainties of inlet velocity, ice layer length and internal mixing in the tube.

A few lines of discussion were added to this point (page 14, lines 24 to 27 and lines 30 to 32).

We would like to make clear that our recommendation is not to have an heated intake tube. We recommend to perform a short heating cycle of the intake tube after the region of mixed-phase clouds at air temperatures below -38 °C (the homogeneous freezing threshold). This heating cycle should not last longer than a few seconds to minutes. It is more appropriate to heat the intake tube by 10 s of degrees warmer than air, if we envision the heating cycle to be as fast as possible. We further clarified our text in Subsection 6.2 containing the recommendations. (page 24, lines 21 and 22)

A few more words about ANSYS/FLUENT might be useful for readers, who are not familiar with CFD. ANSYS seems to be the manufacturer of the FLUENT software.

We agree with the referee and a few more words regarding Fluent were added (page 11, line 29 to 32 and page 12, line 1 and 2).

The appendix expands the paper significantly, but supports the main arguments. I can't tell if the manuscript may be too long and I would not suggest to remove it. It could be shortened if needed.

Due to conflicting suggestions from the two referees, and after discussion with the editor, we decided to convert Appendices A and B into Supplement material.

**Technical comments:**

Abstract, Line 1: delete "(sub)tropical". UTLS water vapor measurements are important in all geographic regions, not just the (sub-) tropics.

We agree with the referee. The word (sub)tropical has been moved to line 4 to characterize the measurements used in this manuscript.

Introduction, line 31 on page 2: better "instrument's Styrofoam box".

Done (page 2, line 31), (page 51, label Figure 16).

Section 2.1, line 11: "automated" instead of "automatized".

Done (page 4, line 11).

Line 12: The instrument seems to control the reflectivity, which may not directly correlate to thickness.

We thank the reviewer for this good point. Indeed, our AMTD manuscript does not mention the reflection of the light by the frost-covered mirror, but immediately talks about the thickness of the frost layer, which is one of the quantities affecting the amount of reflected light, but not the only one. Following the reviewer's suggestion we clarify this in the revised text. We do not use "reflectivity", rather "reflectance", which is defined as the ratio of reflected radiant flux (optical power) to the incident flux at a reflecting object, whereas reflectivity refers only to flat, unstructured surfaces. Reflectance is more general, also referring to rough surfaces, where light is scattered, such as the frost on the mirror (Richmond, 1982). (page 4 line 17).

Page 5, line 3: CFH instead of CHF

Done (page 5 line 16).

Page 5, line 10: Why do the authors use 1 hPa instead of more obvious constant altitude or constant time interval?

The use of averaged data in 1 hPa stems from the analysis of the COBALD data. By dividing the atmosphere in 1 hPa bins, we ensure equal mass layers of backscattering material. The same reasoning applies to the water vapour and the energy balance.

Page 11, line 17; space missing before 'cutcell'

Done (page 12 line 9).

Page 11: Remove the acronym SST, since it is not used except here.

SST $k - \omega$ is the name of the model used for the computational fluid dynamics simulations. The description of the SST acronym is only provided because the target readers of the Journal AMT might not be familiar with the computational fluid dynamics terms. We prefer to keep the acronym and the full name for clarity.

Page 16, line 18, extra comma after "until"

Done (page 17 line 19).

Page 17, line 17; what means "extra ice saturation"? Maybe just delete this phrase.

Any reference to "extra ice saturation" has been removed from the text and from Tables 4 and 5. The results are now presented only as "extra $\chi_{\text{H}_2\text{O}}$".

Page 18, line 12; remove the hyphen after readily.

Done (page 19 line 12).

Page 19, line 10; add "the" before "CFH" (and a few other places).

Done for all "CFH", "RS41" and "COBALD" references in the text.

Page 19, line 10; add "than" before "1.45 mg"

We have rephrased the paragraph. (page 20, lines 13 to 15).

Page 19, line 29; maybe clearer: "which the balloon radius changed with pressure"

Done (page 20, lines 28 and 29).

Figures 2 and other similar Figures: The colors are hard to distinguish, in particular pink, light purple and dark purple. Since there is no ambiguity about the ice or liquid on the mirror, maybe one of the traces could be removed.

Indeed, there is ambiguity about the ice or liquid on the mirror up to the first clearing and freezing cycle of the CFH. After it, it is clear that the deposit on the mirror is ice. Before, the deposit can be liquid, mixed-phase or ice. The three colors of the saturation over water lines are sufficiently similar, yet color blind friendly and can be distinguished after magnification.

Figure 2 b: The very high average mixing ratio near the top of the profile seems to go well above 10 ppmv, i.e. contaminated data may be part of this average.

We agree with the referee. Above 20 hPa, the season average mixing ratio excluding the contaminated profiles of the 16/17 StratoClim campaigns goes well above 10 ppmv. Contaminated data is part of this average. This point has been discussed in Brunamonti et al. (2018) and in Section 5.4.1 of this manuscript when we address the contamination from the balloon envelope. However, this point is not clear when Figure 1 and Figure 2 are discussed. This has been changed in Figures 1 and 2b. The contaminated values in the season average mixing ratio profile above the 20-hPa level are now highlight in the figures. There is now also reference to this contamination in the main body of the manuscript in page 3, lines 24 and 25.

Figure 4: The font in the Figure is too small.

The font size in Figures 4 has been increased.

Figure 6 c: The arrow for the inlet flow does not seem to be vertical as I would have expected. I assume the difference is due to the rotational speed. Can this be indicated in the Figure?

Done (page 41). See below the new figure and legend.

Figure 10e: The bottom half of the flow tube seems to have been shifted a little.

This has been corrected, but it might be an error created by the pdf viewer.

Figure 13: What determines the lower and upper limit of the vertical integration interval?

The lower integration limit is the top of the cloud at 13.5 km altitude and the upper integration limit is the cirrus cloud at 17 km altitude at the tropopause. The explanation has been added to the paper (page 20, line 15 and 16)

Figure 14, legend: The second (a) should be (c)

Done.

Figure 3 and Figure A2: The estimated region for the supercooled mixed phase clouds seems to use different selection criteria.

The selection criteria for the supercooled mixed-phase clouds in Figures 3 and now Figure 2 of the Supplement is the same. The COBALD CI should be higher than 20. CI of 20 is an indication of an ice cloud, while CI around 30 stems from Mie oscillations in the transition regime and thus from the presence of smaller and more monodispersed scatterers, most likely super cooled cloud droplets.

In addition, the water saturation should be compatible with liquid water. Most often liquid droplets exist only when the water saturation is 1, or very close to 1 [0.99 - 1.00]. However, if we consider the process modelled in section 2.4 of the paper, we conclude that if the water droplet size distribution consists of many small droplets and a few big droplets, once the first ice crystals nucleate, the small droplets evaporate fast to feed the solid phase (up to six minutes) but the the big droplets remain in a water sub saturated environment. The process can take up to 17 minutes. However, there is a sub saturation limit to this process too. Through our simulations, we found this limit to be around $S_{liq} \sim 0.85$ for the temperature of the clouds of NT011 and NT029.

So, the selection criteria for the supercooled mixed phase clouds are $S_{liq} > 0.85$ and CI > 20. We realize that this might not be very clear in the paper, so we have rephrased the transition between section 2.3 and 2.4 (page 7, lines 9 to 13) and emphasized the selection criteria for the cold mixed-phase clouds (page 8, lines 21 to 25)

Table 1 and Table 3 legend: Move NT007 into first place following the ordering in the table.

Done.

[Figure]

Figure 1: (a) Schematic of balloon and payload (not to scale). Payload is connected to the balloon by a 55 m long light-weight nylon cord. Payload oscillates with tilt angles $\alpha$ up to 25° during ascent. (b) Schematic of payload with the 2 radiosondes (RS41 and RS92), and the 3 instruments (CFH, ECC Ozone and COBALD) and of intake flow geometry due to balloon ascent and payload rotation. The flow caused by the vertical balloon ascent ($w$) has a component parallel to the intake tube ($w_{||}$) and a component perpendicular to the tube walls ($w_\perp$). Circular motion of the payload adds an additional component ($v_{\perp\text{circ}}$) in the plane perpendicular to the intake tube. (c) The total velocity perpendicular to the tube becomes $v_\perp = v_{\perp\text{circ}} + w_\perp$. The total perpendicular velocity $v_\perp$ and the parallel component of the ascent velocity to the intake tube $w_{||}$ determine the inlet flow and the impact angle $\beta$.

**References**

Brunamonti, S., Jorge, T., Oelsner, P., Hanumanthu, S., Singh, B. B., Kumar, K. R., Sonbawne, S., Meier, S., Singh, D., Wienhold, F. G., Luo, B. P., Böttcher, M., Poltera, Y., Jauhiainen, H., Kayastha, R., Dirksen, R., Naja, M., Rex, M., Fadnavis, S., and Peter, T.: Balloon-borne measurements of temperature, water vapor, ozone and aerosol backscatter at the southern slopes of the Himalayas during StratoClim 2016-2017, Atmospheric Chemistry and Physics, 2018, 1–38, https://doi.org/10.5194/acp-2018-222, 2018.

Philipona, R., Kräuchi, A., Romanens, G., Levrat, G., Ruppert, P., Brocard, E., Jeannet, P., Ruffieux, D., and Calpini, B.: Solar and Thermal Radiation Errors on Upper-Air Radiosonde Temperature Measurements, Journal of Atmospheric and Oceanic Technology, 30, 2382–2393, https://doi.org/10.1175/JTECH-D-13-00047.1, 2013.

Richmond, J. C.: Rationale for emittance and reflectivity, Applied Optics - Symbols Units Nomenclature - Letters to the Editor, 21, 1982.

---

## Author Response (AR2)

**Author's response**

We would like to thank the editor Gabriele Stiller for accepting our paper for publication at the AMT journal. In this author's response, we would like to address some remaining technical corrections. The editor's technical corrections are presented in black and the author's response in blue.

1) Please make sure that the name "FLUENT" is always written in capital letters. I spotted a few paras in the paper (for example, page 11, around line 30) where this is not the case, and this might be confusing.

We thank the editor for spotting this conflicting use of the word "FLUENT". The three instances in page 11 have been corrected.

2) Fig. 2 and similar figures: I tend to agree with reviewer # 1 that "pink" and "light purple" lines are hard to distinguish, and I would urge you to select a different color (at least a different shade) for any of the two.

We have changed the colors of the mentioned lines in Fig 2. and similar figures. The saturation over water ($S_{\text{liq RS41}}$) measured by the RS41 is now light blue, it used to be pink. The saturation over water ($S_{\text{liq,f}}$) from the CFH considering the deposit on the mirror to be frost is now pink, it used to be light purple. Similar figures in the supplement were also altered.

3) The remark of Referee # 2 about the pioneering work by Brewer et al. and your reaction: I think that the reviewer meant that this pioneering work for measurements of water vapour in very dry air (as the Brewer et al. paper title suggests), regardless from which platform, should be cited. I understand very well that the technical details of your work are heavily related to the balloon platform and would not occur in a similar way on an aircraft platform. However, without the method as such, all your work could not be done. Changing the title of your paper is fine with me since the new title describes the work more specifically; however, I still think it would be appropriate to refer to Brewer et al. for the method as such.

We have added the following statement to the paper in page 2 lines 16 to 19: 'The first water vapour measurements in the stratosphere were performed by means of a frost point hygrometer on board an aircraft. The reported frost point temperature was about -83 °C at 12 km height (Brewer et al., 1948). Frost point hygrometers were then developed for balloon borne platforms. The first water vapour measurement from balloon borne frost point hygrometers reported a frost point temperature of about -70 °C at 15 hPa (Barret et al., 1949, 1950; Suomi and Barrett, 1952), corresponding to unrealistically high $H_2O$ mixing ratios ($> 100$ ppmv)'.

To comply with figures copyright's under the Academic License, 14.5 Release, we have added the sentence 'Images used courtesy of ANSYS, Inc' to all figures which include screen-shots of ANSYS products.

**References**

[revised manuscript text omitted]

$$v_{\perp,\,\mathrm{circ}} = \frac{\sqrt{\left(R(t+1)_\mathrm{y} - R(t)_\mathrm{y}\right)^2 + \left(R(t+1)_\mathrm{x} - R(t)_\mathrm{x}\right)^2}}{\Delta t} \tag{B2}$$

where $R(t)$ and $R(t+1)$ are consecutive de-trended trajectory points and $\Delta t = 1$ s.

The perpendicular component $\boldsymbol{w}_\perp$ of the balloon ascent velocity, or $\boldsymbol{v}_{\perp,\,\mathbf{tilt}}$ is projected into the horizontal GPS plane of the oscillation movement as $v_{\perp,\,\mathrm{tilt}_\mathrm{x}}$ and $v_{\perp,\,\mathrm{tilt}_\mathrm{y}}$. We assume $\boldsymbol{v}_{\perp,\,\mathbf{tilt}}$ to be aligned towards the centre of the oscillation $(0, 0)$ as shown in Figure 6b. This direction is evaluated as $\theta(t)$:

$$\theta(t) = \tan^{-1}\left(\frac{R(t)_\mathrm{y}}{R(t)_\mathrm{x}}\right) \tag{B3a}$$

$$v_{\perp,\,\mathrm{tilt}_\mathrm{y}} = \mathrm{sign}\left(R(t)_\mathrm{y}\right)\,v_{\perp,\,\mathrm{tilt}}\,\sin\theta(t) \tag{B3b}$$

$$v_{\perp,\,\mathrm{tilt}_\mathrm{x}} = \mathrm{sign}\left(R(t)_\mathrm{x}\right)\,v_{\perp,\,\mathrm{tilt}}\,\cos\theta(t) \tag{B3c}$$

We then calculate the magnitude of the total perpendicular component of inlet flow velocity $v_\perp$ as

$$v_\perp = \sqrt{\left(v_{\perp,\,\mathrm{circ}_\mathrm{x}} + v_{\perp,\,\mathrm{tilt}_\mathrm{x}}\right)^2 + \left(v_{\perp,\,\mathrm{circ}_\mathrm{y}} + v_{\perp,\,\mathrm{tilt}_\mathrm{
[revised manuscript text omitted]
 the RS41 in m s$^{-1}$. (b and d) Light blue: saturation over water ($S_{\text{liq RS41}}$) by the RS41; purple: saturation over water ($S_{\text{liq,d}}$) from the CFH considering the deposit on the mirror to be dew; pink: saturation over water ($S_{\text{liq,f}}$) from the CFH considering the deposit on the mirror to be frost; blue: ice saturation ($S_{\text{ice}}$) from the CFH; black: ice saturation ($S_{\text{ice RS41}}$) from the RS41; dark grey: 940-nm backscatter ratio from the COBALD; light grey: color index (CI) from the COBALD. Horizontal dashed lines mark supercooled droplet region and $T_{\text{air}}$= 0 °C.

[Figure]

Figure 11: Modelling of the Wegener-Bergeron-Findeisen process in mixed-phase cloud demonstrating that flight NT007 likely encountered supercooled liquid droplets in two occasions: (a-b) refer to cloud 1 between 6.25 and 7 km altitude and (c-d) refer to cloud 2 between 9.2 and 9.85 km altitude. Solid lines: lower estimate of liquid water content (LWC). Dashed lines: upper estimate (see text). (a-b) Initial size distributions for lower estimate simulation: $n_{ice}$ = 0.02 cm$^{-3}$, $r_{ice}$ = 10 $\mu$m; $n_{liq,1}$ = 20 cm$^{-3}$, $r_{liq,1}$ = 10 $\mu$m; $n_{liq,2}$ = 0.004 cm$^{-3}$, $r_{liq,2}$ = 100 $\mu$m; $n_{liq,3}$ = 0.002 cm$^{-3}$, $r_{liq,3}$ = 200 $\mu$
[revised manuscript text omitted]